# Force-dependent allostery of the α-catenin actin-binding domain controls adherens junction dynamics and functions

Noboru Ishiyama [1], Ritu Sarpal[2], Megan N. Wood[3], Samantha K. Barrick[4], Tadateru Nishikawa[1], Hanako Hayashi[5], Anna B. Kobb[6], Annette S. Flozak[3], Alex Yemelyanov[3], Rodrigo Fernandez-Gonzalez [2,6], Shigenobu Yonemura[5,7], Deborah E. Leckband[4,8], Cara J. Gottardi [3,9], Ulrich Tepass[2] & Mitsuhiko Ikura[1,10]

α-catenin is a key mechanosensor that forms force-dependent interactions with F-actin, thereby coupling the cadherin-catenin complex to the actin cytoskeleton at adherens junctions (AJs). However, the molecular mechanisms by which α-catenin engages F-actin under tension remained elusive. Here we show that the α1-helix of the α-catenin actin-binding domain (αcat-ABD) is a mechanosensing motif that regulates tension-dependent F-actin binding and bundling. αcat-ABD containing an α1-helix-unfolding mutation (H1) shows enhanced binding to F-actin in vitro. Although full-length α-catenin-H1 can generate epithelial monolayers that resist mechanical disruption, it fails to support normal AJ regulation in vivo. Structural and simulation analyses suggest that α1-helix allosterically controls the actin-binding residue V796 dynamics. Crystal structures of αcat-ABD-H1 homodimer suggest that α-catenin can facilitate actin bundling while it remains bound to E-cadherin. We propose that force-dependent allosteric regulation of αcat-ABD promotes dynamic interactions with F-actin involved in actin bundling, cadherin clustering, and AJ remodeling during tissue morphogenesis.

[1] Princess Margaret Cancer Centre, University Health Network, Toronto, ON M5G 1L7, Canada. [2] Department of Cell and Systems Biology, University of Toronto, Toronto, ON M5S 3G5, Canada. [3] Department of Medicine, Northwestern University Feinberg School of Medicine, Chicago, IL 60611, USA. [4] Department of Chemistry, University of Illinois, Urbana, IL 61801, USA. [5] RIKEN Center for Life Science Technologies, Kobe, Hyogo 650-0047, Japan. [6] Institute of Biomaterials and Biomedical Engineering, University of Toronto, Toronto, ON M5S 3G9, Canada. [7] Department of Cell Biology, Tokushima University Graduate School of Medical Science, Tokushima 770-8503, Japan. [8] Department of Chemical and Biomolecular Engineering, University of Illinois, Urbana, IL 61801, USA. [9] Department of Cellular and Molecular Biology, Northwestern University Feinberg School of Medicine, Chicago, IL 60611, USA. [10] Department of Medical Biophysics, University of Toronto, Toronto, ON M5G 1L7, Canada. Correspondence and requests for materials should be addressed to N.I. (email: noboru.ishiyama@uhnresearch.ca) or to M.I. (email: mitsu.ikura@uhnresearch.ca)

The mechanical coupling of intercellular adhesion proteins to the cytoskeleton plays a key role in balancing the integrity and plasticity of epithelial tissues. Mechanical tension generated by cortical actomyosin is transmitted through the epithelial sheet by adherens junctions (AJs), allowing contractile forces to change cell and tissue shape[1,2]. The cadherin-catenin cell adhesion complex is the major building block of AJs, and has a crucial function in the dynamic behaviors of epithelial cells, such as cell polarization and cell rearrangements[3,4]. The enormous versatility of cadherin-mediated cell adhesion in tissue morphogenesis and homeostasis requires catenin-dependent regulation of the dynamic cadherin-actin interface in response to variable tension.

α-catenin is an actin-binding and actin-bundling protein responsible for connecting the cadherin-catenin complex to filamentous actin (F-actin) at AJs[5–8]. It plays critical roles in development and tissue homeostasis across the metazoans[9–12], and α-catenin gene mutations have been linked to a variety of physiological abnormalities[13–15], including tumor metastasis[16]. The α-catenin family includes three paralogs expressed in amniotes, E (epithelial), N (neuronal), and T (testis and heart), as well as a single homolog expressed in invertebrates, such as *Drosophila*[17]. Monomeric α-catenin binds to cadherin-bound β-catenin and anchors the cell adhesion complex to the actin cytoskeleton[7,18,19]. α-catenin dissociated from β-catenin can homodimerize to promote actin bundling[5], but the underlying mechanism and function of α-catenin dimers in cell adhesion have been controversial[20,21] and remain to be clarified.

The structure of α-catenin (100 kDa) consists of three distinct domains. The N-terminal (N) domain (30 kDa) facilitates β-catenin binding and homodimerization in a mutually exclusive manner[22,23]. The central mechanosensitive modulatory (M) domain (40 kDa) contains a cryptic binding site for another F-actin-binding protein vinculin[6,24–27]. The C-terminal actin-binding domain (ABD) (28 kDa), which is connected to the rest by a flexible P-linker region[28] (2 kDa), directly binds to F-actin, and closely resembles the vinculin ABD (vin-ABD)[27,29]. Unlike vinculin that forms an autoinhibitory head-to-tail interaction[30], the unhindered αcat-ABD[23,27] forms a catch bond with F-actin that stabilizes the interaction under tension[8]. However, the molecular basis of this catch bond is unknown, and the physiological significance of its distinctive mechanical properties has not yet been demonstrated.

Here we reveal that a force-dependent conformational change in the αcat-ABD allosterically regulates direct F-actin binding. Several lines of evidence suggest that α1-helix unfolding changes the conformational dynamics of the actin-binding site. Furthermore, the αcat-ABD in an activated state homodimerizes to facilitate actin bundling. Our data suggest that manipulation of the ABD-dependent mechanosensory function of α-catenin severely interferes with AJ remodeling in mammalian cells and *Drosophila* embryos. Surprisingly, not only loss but also gain of F-actin binding propensity dramatically compromises α-catenin function in morphogenesis. Based on these results, we propose a new mechanism of the force-dependent, dynamic cadherin-actin linkage regulated by the ABD of α-catenin.

## Results

**Force-dependent unfolding of αcat-ABD enhances actin binding**. The direct interaction between α-catenin and F-actin was demonstrated to be a catch bond[8], an interaction that is stabilized by increased force[31,32]. Since the C-terminal tail (residues 865-906) of α-catenin is postulated to be part of the interface between the αcat-ABD and F-actin[33–35], we hypothesized that a regulatory motif resides within or near the N terminus of ABD.

We monitored the disassembly and reformation of AJs in α-catenin-deficient R2/7 epithelial cells[36,37] expressing various αE-catenin deletion mutants (Supplementary Fig. 1a; Supplementary Table 1). We found that the deletion of residues 663-696 from the ABD was associated with an unusual accumulation of cadherin-catenin-F-actin complexes in the cytoplasm after trypsinization of cell monolayers (Supplementary Fig. 1b, c), and delayed reformation of AJs with a unique square wave-like arrangement (Supplementary Fig. 2a). Cells with these deformed junctions showed diminished tight junction barrier function compared to full-length αE-catenin (αEcatFL)-expressing cells (Supplementary Fig. 2b). In addition, the αEcat-ABD residues 663-906 expressed in R2/7 cells colocalized with actin-rich regions at the cell periphery (Fig. 1a), whereas an N-terminally truncated form of ABD (ABD*; residues 697-906) prominently accumulated along stress fibers and actin rods (Fig. 1a), consisting of tightly packed actin bundles (Supplementary Fig. 2c). These results suggest the αE-catenin residues 663-696 regulate the association of αcat-ABD with different actin assemblies (Fig. 1a), and are critical for the normal function of αcat-ABD in forming AJs and, consequently, epithelial differentiation.

Comparison of crystal structures of αcat-ABDs[27,38] with the vin-ABD[30] revealed several highly conserved motifs of α-catenin potentially involved in its unique actin-binding mechanism: an N-terminal α1-helix (αE-catenin residues 669-675), a β-hairpin (βH; residues 799-810), and a C-terminal tail (Fig. 1b, c and Supplementary Fig. 3). Considering that the α1-helix is part of the ABD truncation (residues 663-696) that resulted in abnormal F-actin association and a failure to form normal AJs in R2/7 cells (Fig. 1a and Supplementary Fig. 1b, 2a–c), we sought to explore the potential role of α1-helix in the regulation of force-dependent αcat-ABD-F-actin interaction. We performed equilibrium and constant-force steered molecular dynamics (SMD) simulations of the αN-catenin ABD (αNcat-ABD) to gain insights into how α1-helix may respond to increasing mechanical tension at the cadherin-actin interface. To help discuss equivalent residues between αN-catenin and αE-catenin with different residue numbering (e.g., V795 of αN-catenin is equivalent to V796 of αE-catenin), henceforth the αN-catenin residues will be denoted by using the equivalent αE-catenin residue numbers accompanied by a subscripted 'N' (e.g., V795 as $V796_N$) for clarity. The SMD simulations showed α1-helix unfolding after a constant pulling force was applied on αNcat-ABD for 60 ns (Fig. 1d, Supplementary Fig. 4a, b and Supplementary Movie 1). Interestingly, shortly before α1-helix unfolded (at ~45 ns), the side chain of $V796_N$ turned over from a cryptic position to an exposed position (Fig. 1e and Supplementary Movie 2). αN-catenin residues $V796_N$ and $I792_N$ are equivalent to the vinculin actin-binding site residues, V1001 and I997[39] (Fig. 1c). These results suggest that the conformational flexibility of α1-helix and the dynamics of $V796_N$ are mechanically coupled within the αNcat-ABD. This mechanism would be consistent with catch bond formation, if the conformation change of α1-helix exposes $V796_N$ and enhances the bond strength between αcat-ABD and F-actin.

To assess whether the α1-helix affects the α-catenin-F-actin interaction, we performed in vitro actin cosedimentation assays with three ABD variants of αE- and αN-catenin (αE-catenin residue numbers are shown): a wild type form of ABD (ABD-WT; residues 652-906), an ABD with a structure-guided *h*elix-1 mutation (H1) designed to unfold α1-helix (ABD-H1; RAIM670-673GSGS) (Fig. 1c), and an ABD with a partially deleted α1-helix (ABD-Δα1; residues 671-906)[33]. The structural integrity of αcat-ABD was not affected by these mutations (Supplementary Fig. 4c–e). We observed a nearly two-fold increase in the cosedimented amount of either ABD-H1 or ABD-Δα1 compared to ABD-WT (Fig. 1f). These results indicate that the α1-helix attenuates the

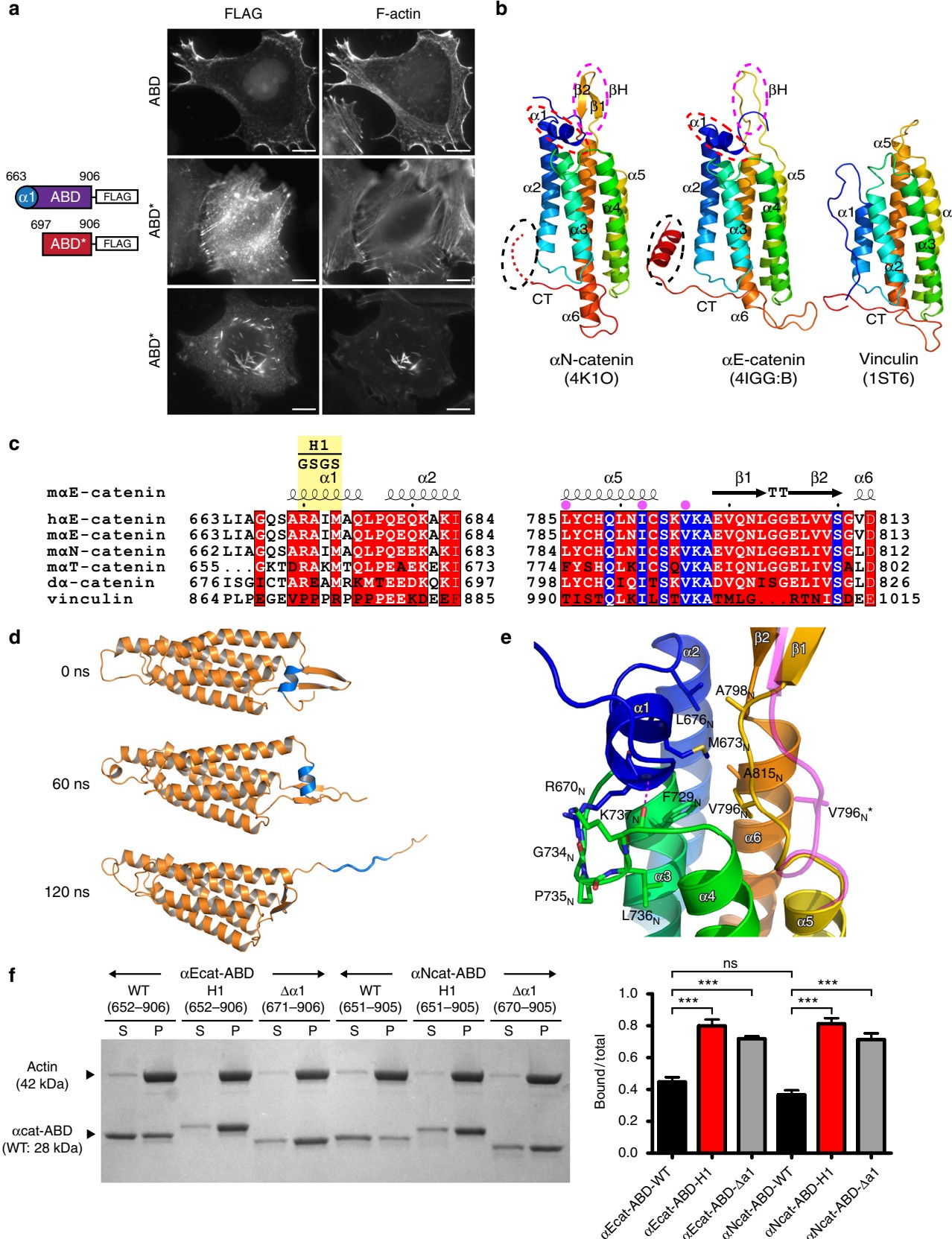

**Fig. 1** Force-induced unfolding of α1-helix enhances the F-actin-binding activity of the αcat-ABD. **a** R2/7 cells transiently expressing ABD (residues 663-906) or ABD* (residues 697-906). αcat-ABD/ABD*-FLAG and actin were labeled with the anti-DDDDK antibody and phalloidin, respectively. Scale bar, 10 µm. **b** Comparison of the ABD crystal structures of αN-catenin, αE-catenin and vinculin. The αcat-ABD contains three distinct structural motifs: α1-helix (α1; red circle), β-hairpin (βH; magenta circle), and C-terminal tail (CT; black circle). PDB ID codes are indicated in parentheses. **c** Multiple sequence alignment of α-catenin and vinculin primary sequences. The α1-helix and βH sequences are highly conserved among three paralogs of α-catenin (E, N and T; h, human; m, mouse), as well as in *Drosophila* α-catenin (dα-catenin). The H1 mutation (RAIM670-673GSGS) is indicated. Conservation of three actin-binding site residues in α-catenin, as well as the vinculin actin-binding site residues, I997 and V1001, are marked by purple dots. **d** Snapshots of the structure of αNcat-ABD at select time points during a constant-force SMD simulations (100-pN pulling force for 120 ns) (Supplementary Movie 1). Cartoon representation shows α1-helix (blue) starts to unfold at ~60 ns. **e** A close-up view of α1-helix and $V796_N$ in the αNcat-ABD crystal structure. During constant-force SMD simulations, $V796_N$ in a cryptic state is exposed ($V796_N$*; magenta) at 45 ns, shortly before α1 unfolding occurred at 60 ns (**d**). Two conserved α1-helix residues, $R670_N$ and $M673_N$, engage in critical interactions with the five-helix bundle of ABD to attenuate the ABD-F-actin interaction. **f** Actin cosedimentation assays comparing WT, H1 and Δα1 variants of αEcat-ABD and αNcat-ABD. Whereas less than half of total ABD-WT (0.37-0.45) cosedimented with F-actin for both αE-catenin and αN-catenin, alterations in α1-helix, either by deletion or unfolding via the H1 mutation, significantly increased the amount of mutant αcat-ABD proteins cosedimented with F-actin (0.71–0.81). Supernatant (S) and pellet (P) fractions are indicated. Data are presented as mean ± standard error of the mean (SEM) (*N* = 3). Significance by ANOVA: ***$P < 0.001$

αcat-ABD-F-actin interaction, and alterations in α1-helix significantly enhance the F-actin-binding activity of both αEcat-ABD and αNcat-ABD.

**α1-helix unfolding induces weak αcat-ABD homodimerization.** To examine the structural details of αcat-ABD with enhanced F-actin binding, we determined crystal structures of αNcat-ABD-H1 (Fig. 2a, Supplementary Fig. 5, and Supplementary Table 2). The αNcat-ABD-H1 structure closely resembles the overall fold of αNcat-ABD-WT (PDB ID: 4K1O)[27] (Supplementary Fig. 6a), except for the α1-helix residues. However, unlike the monomeric αNcat-ABD-WT structure, αNcat-ABD-H1 crystallized as a homodimer connected by two βH motifs (Fig. 2a and Supplementary Fig. 6b). The dimer interface involves $L807_N$ of the βH, which mimics $M673_N$ of α1-helix interacting with the hydrophobic patch in the αNcat-ABD-WT structure (Fig. 2b and Supplementary Fig. 6c). Moreover, the observation of αNcat-ABD-H1 dimerization, which occludes $3100 \, \text{Å}^2$ of solvent-accessible surface (Fig. 2a), in two distinct crystal forms (Supplementary Fig. 6b) provides a basis for further examining the physiological relevance of this ABD-dimer interface. Our NMR analysis of αNcat-ABD-H1 in solution showed that concentration-dependent chemical shift perturbations (CSPs) mostly occurred in the βH motif (Fig. 2c and Supplementary Fig. 7a, b). In addition, we observed the increased propensity for αEcat-ABD-H1 to dimerize, albeit very weakly, in a concentration-dependent manner compared to αEcat-ABD-WT by size-exclusion chromatography-coupled multiangle light scattering (SEC-MALS) (Fig. 2d and Supplementary Fig. 7c). These results further support that the unfolded α1-helix propagates the weak dimerization of αcat-ABD through the βH-dependent interface.

One functional implication for α-catenin dimerization is actin bundling[5], which has been presumed to occur through N-domain dimerization of αcatFL[20,27,38]. However, the ability of α-catenin to homodimerize through the ABD suggests an alternative actin-bundling mechanism. Indeed, actin bundling assays showed that both isolated αEcat-ABD-WT and αEcat-ABD-H1 proteins are capable of actin bundling (Fig. 2e). We next examined the involvement of α1-helix and βH motifs in ABD-dependent actin bundling. A βH-deletion mutant (αEcat-ABD-ΔβH) and a construct carrying both the H1 and βH-deletion mutations (αEcat-ABD-H1ΔβH) were well folded (Supplementary Fig. 4d, e), but cosedimented markedly less with F-actin at a high centrifugal force (100,000×*g*), indicating that the ΔβH mutation inadvertently affected F-actin binding of αEcat-ABD (Fig. 2e). Nevertheless, αEcat-ABD-ΔβH displayed residual actin bundling, whereas αEcat-ABD-H1ΔβH was unable to bundle F-actin

(Fig. 2e). These results suggest that actin bundling can be facilitated by ABD dimerization through the βH-dependent interface, as well as through an unknown mechanism involving the α1-helix in our assays. In addition, our NMR transferred cross saturation (TCS) experiments with $^{15}\text{N}/^{2}\text{H}$-labeled αNcat-ABD-WT and unlabeled F-actin indicated that the ABD directly interacts with F-actin through α5- and α6-helices, likely involving $I792_N$ and $V796_N$, and, unexpectedly, through α3- and α4-helices on the opposite side of ABD (Fig. 2f and Supplementary Fig. 7d). This finding may point to a secondary contact site involved in actin bundling (Fig. 2f). Collectively, these results support the view that α-catenin facilitates actin bundling through ABD homodimerization.

**ABD mutations compromise AJ remodeling in cells and embryos.** Our finding that α-catenin can dimerize and mediate actin bundling independent of the N domain implicates the AJ-associated pool of α-catenin in reorganization of the actin cytoskeleton. To determine how alterations of α1-helix and βH would affect cadherin-mediated cell-cell adhesion, we tested the function of α-catenin mutants in R2/7 cells and *Drosophila* embryos. First, we examined R2/7 cells stably expressing αEcatFL fused with monomeric GFP (Supplementary Fig. 8a). Cells expressing αEcatFL or αEcat-H1 showed the typical cobblestone appearance of well-adhered epithelial cells with consistent colocalization of α-catenin and actin at AJs (Supplementary Fig. 8b). In contrast, cells expressing αEcat-ΔβH, αEcat-H1ΔβH, or a construct that lacks ABD entirely (αEcat-ΔABD) did not form cohesive cell monolayers and showed increased presence of α-catenin in protrusions (Supplementary Fig. 8b). Similarly, both αEcatFL or αEcat-H1 cells formed three-dimensional spheroids on ultra-low-attachment plates, whereas cells expressing other mutants remained in a semi-aggregated state (Supplementary Fig. 8c).

To find out how the H1 and ΔβH mutations affect the cell-cell adhesive strength, we performed an epithelial sheet disruption assay[40]. αEcatFL or αEcat-H1 cell monolayers lifted as a continuous sheet from the culture plate upon dispase treatment prior to mechanical disruption (Fig. 3a), but αEcat-ΔβH, αEcat-H1ΔβH, and αEcat-ΔABD cell sheets disintegrated into numerous pieces (Supplementary Fig. 8d). Subsequent mechanical disruption of cell monolayers caused αEcatFL monolayers to fragment, whereas αEcat-H1 monolayers remained mostly intact (Fig. 3a, b). These observations indicate that monolayers formed by αEcat-H1 cells have increased resistance towards mechanical stress compared to αEcatFL cells.

Next, we challenged R2/7 cells in scratch wound assays. In contrast to unchallenged cells, αEcatFL cells at the wound front

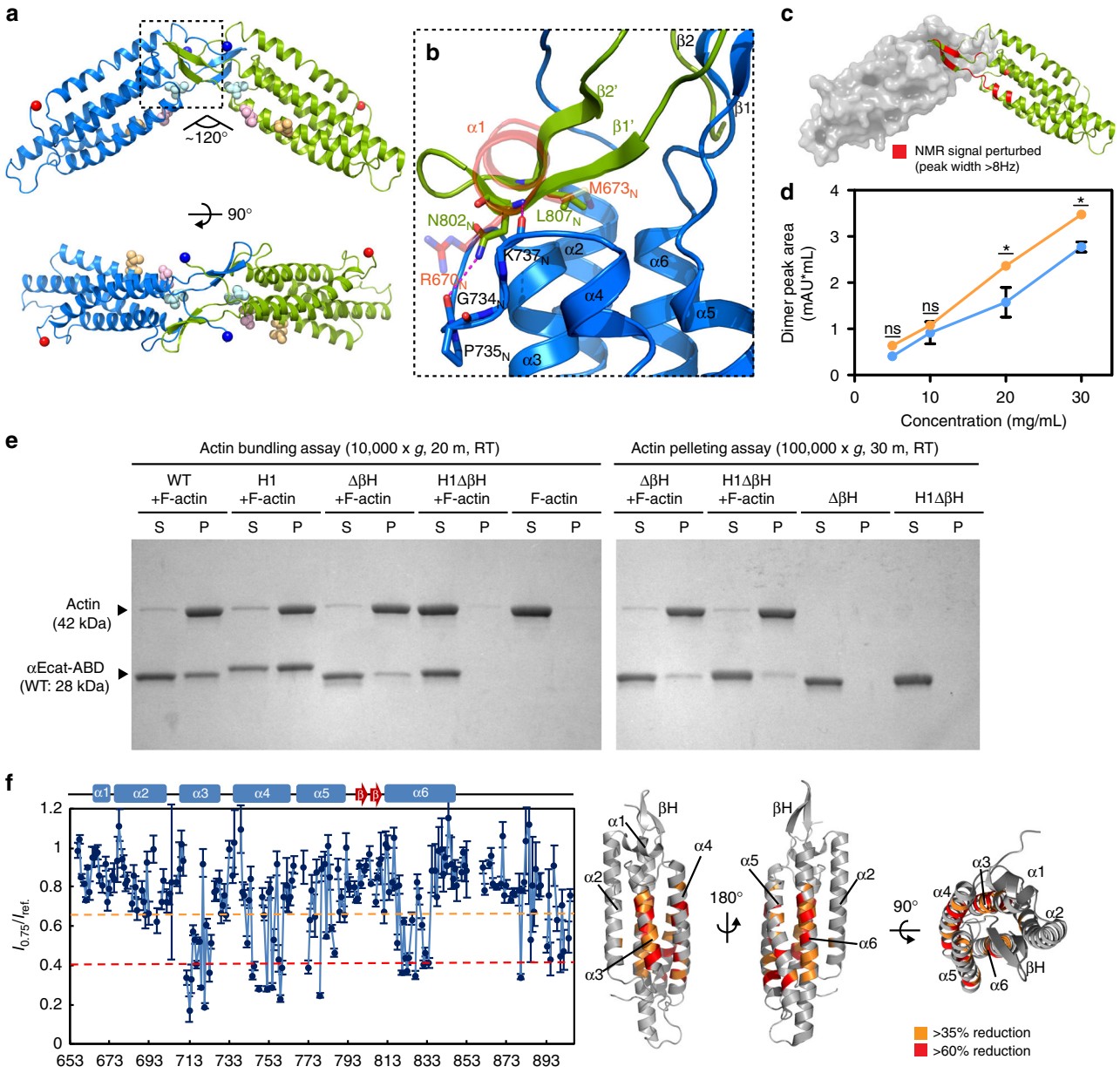

**Fig. 2** Crystal structure of αNcat-ABD-H1 reveals a novel ABD dimer interface. **a** Crystal structure of the αNcat-ABD-H1 dimer in form A (two protomers shown as blue and green). The N and C termini of ABD are indicated by blue and red spheres, respectively. Three actin-binding site residues, $L785_N$, $I792_N$ and $V796_N$, are shown as light blue, pink and orange spheres. **b** A close-up view of the ABD dimer interface. The dashed-line box in **a** is rotated by ~90° CCW. The βH motif from one protomer covers the hydrophobic patch exposed by α1-helix unfolding in the adjacent protomer (the α1-helix of αNcat-ABD-WT is shown in red). **c** Concentration-dependent CSPs of αNcat-ABD-H1 are localized to the βH residues. Residues with CSP greater than 8 Hz are indicated on the αNcat-ABD-H1 structure in red. **d** SEC-MALS analysis of αEcat-ABD. The integrated dimer peak area was plotted against the αEcat-ABD concentration for αEcat-ABD-WT (blue) and αEcat-ABD-H1 (orange). Data are presented as mean ± SEM ($N = 3$). Significance by ANOVA: *$P < 0.05$. **e** In vitro actin cosedimentation assays of αEcat-ABD variants, WT, H1, ΔβH, and H1ΔβH. Actin bundling was analyzed by sedimentation at low RCF (10,000×$g$). The F-actin-bound ABD was sedimented at high RCF (100,000×$g$). **f** TCS experiments with unlabeled F-actin and $^{15}$N/$^2$H-labeled αNcat-ABD-WT. Plots of the reduction ratios of the backbone amide signal intensities observed with and without presaturation. Residues with >60% and >35% signal reduction are indicated on the αNcat-ABD-WT structure (right). The affected residues are mostly located in the last four α-helices (α3-α6)

displayed punctate AJs connected to actin cables aligned along the wound edge, whereas αEcat-H1 cells formed less organized punctate AJs and actin assemblies (Fig. 3c and Supplementary Fig. 9a). High-resolution live-imaging revealed that αEcat-H1 AJs were less organized towards the wound front, resulting in unproductive cell-cell tugging events that appeared to interfere

with forward sheet migration (Supplementary Movie 3). In fact, αEcat-H1, αEcat-ΔβH, and αEcat-H1ΔβH cells were all inferior to αEcatFL cells in wound closure, and no better than αEcat-ΔABD cells (Fig. 3d, e and Supplementary Fig. 9b). By tracking cells individually, we found that αEcat mutant cells moved with similar speeds as αEcatFL cells, but less persistently, contributing to

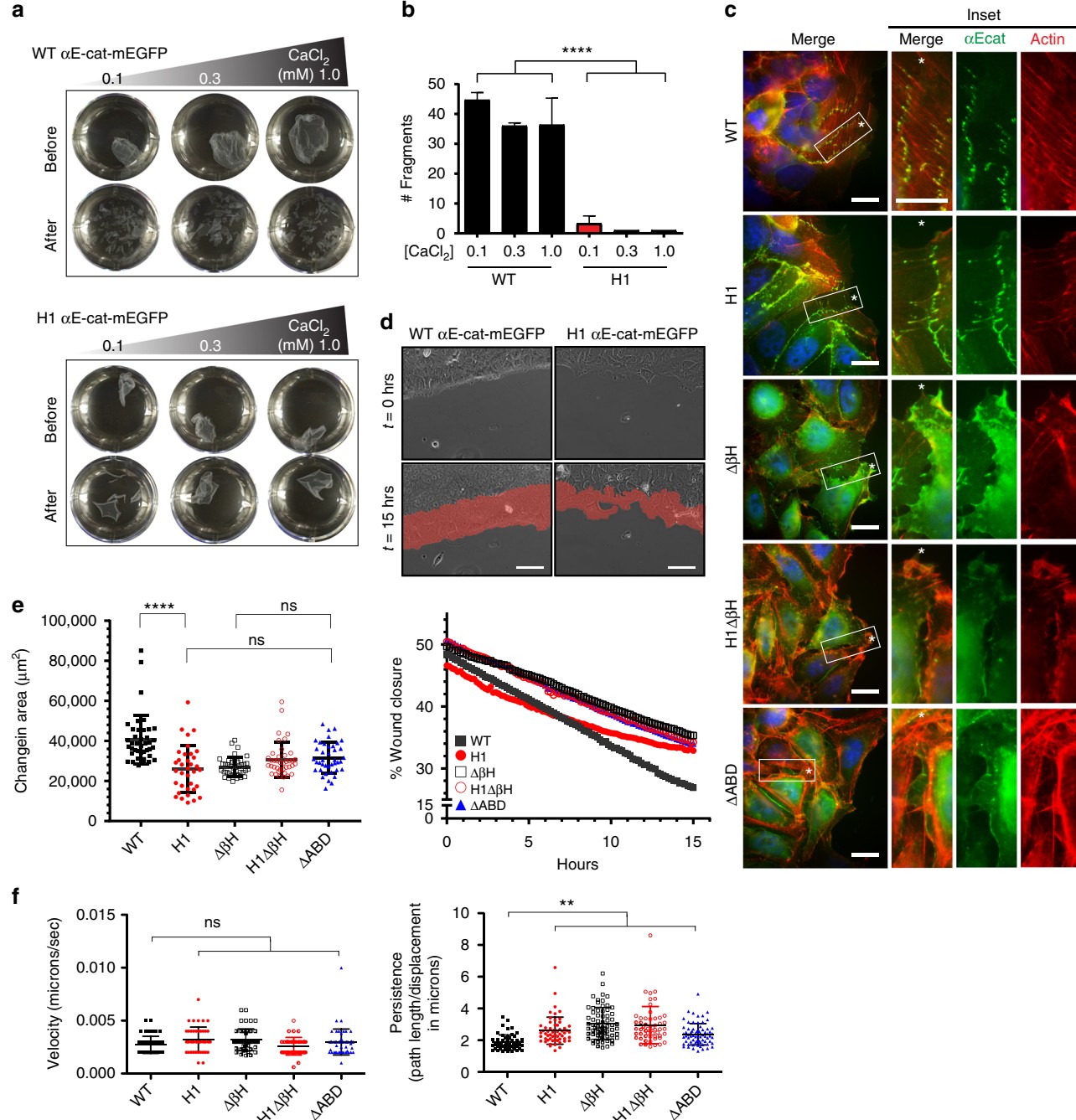

**Fig. 3** α1-helix and βH are critical for the formation of multicellular structures and wound healing. **a** Epithelial sheet disruption assay of R2/7 cells expressing α-catenin variants. Representative αEcat monolayers before and after mechanical stress treatment are shown. **b** Plots showing total cell monolayer fragments after mechanical stress treatment. Mechanical disruption caused αEcatFL cell monolayers to fragment, whereas αEcat-H1 monolayers remained intact with only few fragments forming at a low calcium concentration. Data are presented as mean ± standard deviation (SD) ($N = 3$). Significance by ANOVA; ****$P < 0.0001$. **c** Confocal images of R2/7 cells expressing αEcat variants at the wound fronts. Close-up views of inset boxes are shown. Scale bar, 20 μm. **d** Scratch wound healing assays with R2/7 cells expressing αEcat-WT or αEcat-H1. The areas of wound healing after 15 hrs are shown in red. Scale bar = 50 μm. **e** Plots showing changes in total wound closure area and the wound closure percentage over time. Data are presented as mean ± SD (>35 fields of view (FOV); > 5 biological replicates (BR)). Significance by ANOVA; ****$P < 0.0001$. **f** Plots showing changes in the persistence, but not the velocity, of αEcat mutant cells at the wound front compared to αEcatFL cells. Data are presented as mean ± SD (>35 FOV; > 5 BR). Significance by ANOVA; **$P < 0.01$

overall reduced epithelial sheet migration (Fig. 3f, Supplementary Fig. 9b and Supplementary Movie 4). In addition, differences between αEcatFL and αEcat-H1 cell trajectories were independently validated using a particle image velocimetry (PIV)-based tracking method (Supplementary 9c). These observations suggest that α-catenin with a defective α1-helix can support AJs in static epithelia but fails to support dynamic AJ rearrangements and cell movements.

To further assess α1-helix and βH functions in tissue organization we generated mutants in *Drosophila* α-Catenin (α-Cat) and tested their function in transgenic animals. Both α1-helix and βH regions are conserved in α-Cat (Fig. 1c), and previous work showed that the ABD of *Drosophila* α-Cat (αCat-ABD) is essential for cell adhesion[7]. Moreover, αCat-ABD-H1 showed enhanced actin binding and bundling activity compared to αCat-ABD (Supplementary Fig. 10a) similar as mammalian proteins (Fig. 2e). Zygotic null mutants for *α-Cat* (αCat$^{-/-}$) show embryonic lethality and severe defects in head morphogenesis[9] (Fig. 4a, b). Expression of full-length α-Cat (αCatFL) did rescue αCat$^{-/-}$ mutants to adulthood. In contrast, αCat-H1, αCat-Δα1, αCat-ΔβH, and αCat-H1ΔβH did not rescue the embryonic lethality of αCat$^{-/-}$ mutants similar to αCat-ΔABD (Fig. 4a). Expression of αCat-H1, αCat-Δα1, and αCat-H1ΔβH led to some improvements in head morphogenesis, and a small number of animals expressing αCat-H1 or αCat-H1ΔβH survived to larval stages (Fig. 4a). Immunoblot analysis (Supplementary Fig. 10b) and tissue staining in an αCat$^{-/-}$ mutant background (Fig. 4c) showed that our constructs are expressed at levels similar to endogenous α-Cat and are effectively recruited to AJs. Efficient recruitment of α-Cat proteins to AJs in a wildtype background indicated that mutant proteins are not outcompeted by endogenous α-Cat (Supplementary Fig. 10c). We noted that overexpression of αCat-H1 or αCat-Δα1 had a toxic effect on survival with most animals dying as larvae, whereas over-expression of other α-Cat constructs led to pupal lethality or adult survival (Fig. 4a). The failure of α-Cat proteins with a compromised α1-helix to substantively rescue αCat$^{-/-}$ mutants was surprising as those variants are likely capable of coupling cadherin to the actin cytoskeleton to promote intercellular adhesion. On the other hand, enhanced F-actin binding of αCat-ABD-H1 (Supplementary Fig. 10a) could explain the observed toxicity upon overexpression of these constructs. Our findings indicate that the function of the α1-helix in attenuating interactions between α-catenin and F-actin is instrumental for AJ function in developing epithelia.

We further examined the role of α1-helix in wound repair, which is driven by the polarized assembly of actin at the interface between wounded and adjacent cells in the *Drosophila* embryonic epidermis[41]. Polarization of actin (and the non-muscle myosin II) in the cells adjacent to the wound results in the assembly of a supracellular contractile cable around the perimeter of the wound that drives tissue repair[42]. The quantified wound closure dynamics revealed that αCat$^{-/-}$ embryos expressing αCatFL repaired damage to their epidermis faster than αCat$^{-/-}$ embryos, whereas αCat-H1 expression did not significantly accelerate wound closure in an αCat$^{-/-}$ epidermis. (Fig. 4d, e). These results are consistent with our whole animal rescue experiments, as well as our scratch wound healing assays (Fig. 3d–f), and collectively suggest that a compromised α1-helix severely interferes with α-catenin function in tissue morphogenesis.

**Actin-binding site residues are essential for α-cat function**. Considerable in vitro evidence suggests that α-catenin can directly interact with F-actin[5,7,8,20,27,29,33,35,43,44]. A previously determined low resolution (18 Å) cryo-EM map of an αcat-ABD-F-actin complex precluded any detailed analysis of the complex interface[29]. Nonetheless, it suggested that the αcat-ABD interacts with two actin monomers adjacently aligned on the long axis of F-actin. A similar arrangement was observed in a recently determined 8.5-Å cryo-EM structure of a vin-ABD-F-actin complex, which revealed that the last two α-helices of vin-ABD interact with F-actin[45]. Considering the relatively high sequence identity shared between αcat- and vin-ABDs (~30%)[27], we

generated an atomic model of the αNcat-ABD-H1-F-actin complex based on the vin-ABD-F-actin structure. In this model, α5- and α6-helices of the αNcat-ABD-H1 interact with two axially arranged actin monomers of F-actin (Fig. 5a). In particular, the α5-helix contains the highly conserved residues, I792$_N$ and V796$_N$ (Fig. 1c). I792$_N$ of αN-catenin assumes an exposed position closely resembling the vinculin actin-binding site residue I997[30]. In contrast, the conformation of V796 remains ambiguous, partly due to poorly defined electron density of this region in the 3.7-Å crystal structure of human αE-catenin[38], and a cryptic position of V796$_N$ in the αNcat-ABD structure[27] (Supplementary Fig. 11a) compared to the fully exposed V1001 of vinculin[30].

To better characterize the αE-catenin actin-binding site, we elucidated a crystal structure of αEcat-ABD-WT at 2.2-Å resolution (Fig. 5b, Supplementary Fig. 5, and Supplementary Table 2). The electron density map of α5-helix clearly shows that V796 adopts a conformation that exposes its side chain on the ABD surface, along with two additional hydrophobic residues L785 and I792 (Supplementary Fig. 5g and 11b). Our site-directed mutagenesis and actin cosedimentation assays with the αEcat-ABD variants support critical roles of these hydrophobic residues in F-actin-binding: Ala substitutions of L785, I792 and V796, individually or together as 3A, led to a range of reduction (75, 36, 47, and 78%, respectively) in the amount of ABD cosedimenting with F-actin compared to αEcat-ABD-WT (Fig. 5c). The effects of I792A and V796A were greater in the H1 background (reduction of 70% and 73%, respectively), confirming that alterations of these residues significantly reduce F-actin binding by αEcat-ABD-H1 (Fig. 5c). In contrast, Ala substitution of V714, which is located on the α3-helix surface, resulted in no reduction (Fig. 5c). Also, none of the above mutations appear to interfere with the ability of αEcat-ABD to bundle F-actin (Supplementary Fig. 12). The equally significant reduction observed with either the L785A mutation alone or 3A suggests that L785 plays a central role in establishing the critical hydrophobic interface between F-actin and αcat-ABD. The measurable reduction in F-actin binding with I792A or V796A suggests that I792 and V796 are likely involved in further stabilizing this interface, and any changes to these residues could modulate the F-actin-binding activity of αcat-ABD. These results confirm that the hydrophobic residues on the α5-helix surface constitute an important binding surface for F-actin interaction.

Next we tested the in vivo importance of this interaction by expressing an αCat-3A (L798A + I805A + V809A) mutant in *Drosophila*. All three key hydrophobic residues identified in mammalian αE-catenin or αN-catenin are conserved in *Drosophila* α-Cat (Fig. 1c). αCat-3A was recruited normally to the cadherin-catenin complex (Fig. 4c and Supplementary Fig. 10c) but failed to show a rescue of the αCat$^{-/-}$ mutant phenotype; a fraction of embryos showed more severe defects than αCat$^{-/-}$ mutants, consistent with a mild dominant-negative effect of αCat-3A expression (Fig. 4a, b). We conclude that direct interaction between α-catenin and F-actin is essential for AJ assembly and function during development.

**Allosteric coupling between α1-helix and V796 dynamics**. Our observations of the cryptic (attenuated) and exposed (activated) conformations of V796 (Supplementary Fig. 11a, b), despite the nearly identical primary sequences of αEcat- and αNcat-ABDs (87% identity; Supplementary Fig. 3), indicate that this residue resides within a conformationally dynamic region. Consistent with this idea, the αEcat-ABD-WT structure contains an internal cavity that could accommodate V796 in the cryptic state similar to V796$_N$ in the αNcat-ABD-WT structure (Fig. 5b, Supplementary Fig. 11c). This internal cavity is partly formed by the side

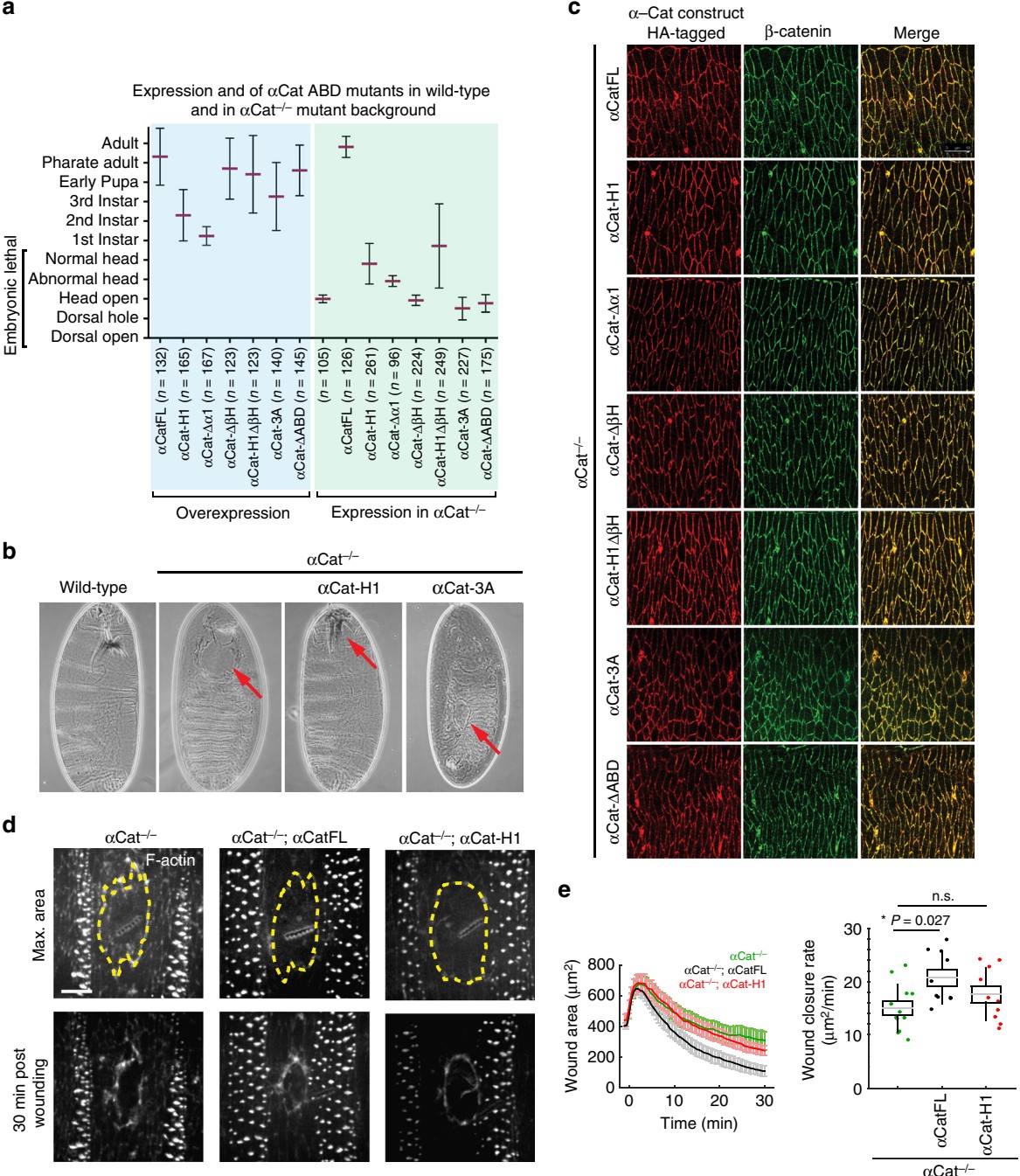

**Fig. 4** αCat ABD mutants fail to rescue αCat function in *Drosophila*. **a** Phenotypic consequences of the overexpression of αCat ABD mutants and rescue activity of αCat ABD mutants expressed in *αCat⁻/⁻* zygotic null mutants. Overexpression: all mutant constructs showed significantly reduced survival (*P* < 0.0001) compared to αCatFL overexpression. Rescue experiments: Mutant constructs showed a significant rescue (αCatFL, αCat-H1, αCat-Δα1, αCat-H1ΔβH [*P* < 0.0001]) or enhancement (αCat-3A [*P* < 0.0001], αCat-ΔABD [*P* = 0.0071]) of the *αCat⁻/⁻* zygotic mutant phenotype. Expression of αCat-ΔβH did not significantly modify the *αCat⁻/⁻* mutant phenotype. Data are presented as mean ± SD. **b** Cuticles of wild-type embryo, of *αCat⁻/⁻* mutant showing failure in head morphogenesis ('head open'; arrow), of *αCat⁻/⁻* mutant expressing αCat-H1 showing a defective head skeleton ('abnormal head'; arrow), and of of *αCat⁻/⁻* mutant expressing αCat-3A showing dorsal hole (arrow) in addition to an open head. **c** HA-tagged αCat variants were expressed with *Act5c-Gal4 da-Gal4* in the epidermis of *Drosophila* embryos mutant for *α-Cat (αCat⁻/⁻)* at stage 15. AJs marked by β-catenin. **d** Epidermal wounds in *αCat⁻/⁻*, *αCat⁻/⁻* mutant expressing αCatFL, and *αCat⁻/⁻* mutant expressing αCat-H1. F-actin was labeled with GFP::UtrophinABD. Top panels show time of maximum wound area (yellow lines outline the wounds) and bottom panels show epidermis 30 min after wounding. Anterior left, dorsal up. Scale bar, 10 μm. **e** Wound area over time (left) and wound closure rate (right) for *αCat⁻/⁻* (red, *n* = 12 wounds), *αCat⁻/⁻* mutants expressing αCatFL (cyan, *n* = 10 wounds), and *αCat⁻/⁻* mutants expressing αCat-H1 (green, *n* = 10 wounds). αCatFL, *αCat⁻/⁻* embryos repaired damage to their epidermis significantly faster than *αCat⁻/⁻* embryos (*P* = 0.027), whereas αCat-H1 *αCat⁻/⁻* embryos did not show a significant difference to *αCat⁻/⁻* embryos. The box plot shows the mean (gray line), SEM (box), and SD (black lines)

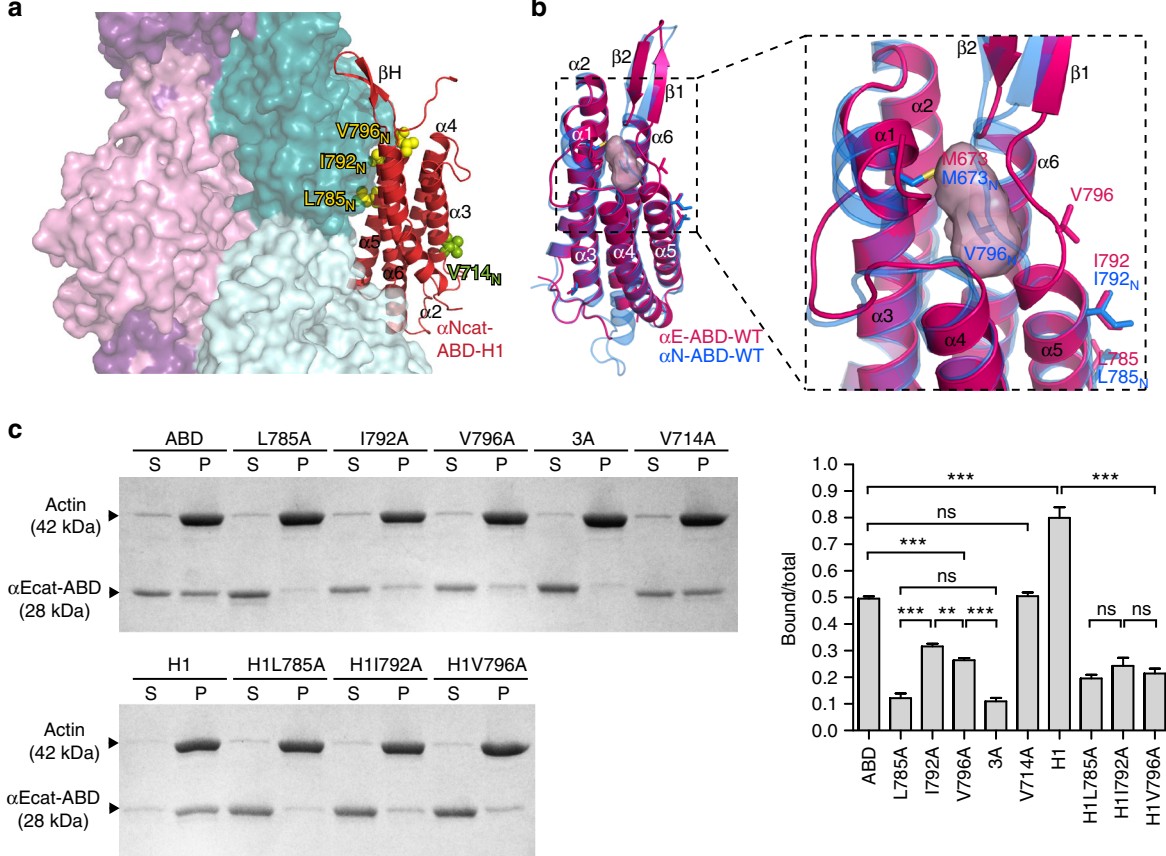

**Fig. 5** Identification of the critical actin-binding site residues in α-catenin. **a** Model of the αcat-ABD (red) bound to two axially adjacent actin monomers (dark and light teal) within F-actin, based on the vin-ABD/F-actin cryo-EM structure (PDB ID: 3JBI). **b** Comparison of high-resolution crystal structures of αEcat-ABD-WT (red) and αNcat-ABD-WT (blue). The overall structure of αEcat-ABD-WT closely resembles αNcat-ABD-WT, as two ABD structures can be superposed with RMSD of 0.53 Å over 156 residues. A close-up view (right) shows that αEcat-ABD-WT contains a cavity (pink molecular envelope), which could accommodate V796 in a cryptic state similar to V796$_N$ in the αNcat-ABD-WT structure. **c** Actin cosedimentation assays of αEcat-ABD variants: WT, L785A, I792A, V796A, 3A, V714A, H1, H1L785A, H1I792A, and H1V796A. Data are presented as mean ± SEM (N = 3). Significance by ANOVA: **P < 0.01, ***P < 0.001

chain of M673 from α1-helix (Fig. 1e), hence raising the possibility that α1-helix unfolding allosterically affects the F-actin-binding site by changing the conformational dynamics of V796.

To determine the influence of α1-helix on the actin-binding site of α-catenin, we developed a new bio-layer interferometry (BLI) approach to measure the kinetics of the αcat-ABD-F-actin interaction. We immobilized F-actin onto the streptavidin-coated optical sensor with biotinylated LifeAct actin-binding peptides (LAbio)[46], and measured subsequent association and dissociation of αcat-ABD (Fig. 6a and Supplementary Fig. 13a). We determined that concentration-dependent F-actin binding curves of αEcat-ABD-WT fit well with a 2:1 hetero-ligand:receptor model with two $K_D$ values, $K_{D1} = 2.0\ \mu M$ and $K_{D2} = 0.3\ \mu M$ (Fig. 6b, Table 1, and Supplementary Fig. 13b). This model supports αEcat-ABD-WT in an equilibrium between the attenuated and activated actin-binding states, respectively. The lower $K_{D2}$ value is consistent with the positive cooperativity of F-actin binding by αEcat-ABD as previously reported[8,29]. In contrast, the BLI data of αEcat-ABD-H1 fit well with a 1:1 ligand:receptor model with the single $K_D$ value of 0.58 μM (Fig. 6c, Table 1 and Supplementary Fig. 13b), reflecting a predominantly activated state of αEcat-ABD-H1. The effects of mutations in the α-catenin actin-binding site, as well as ΔβH mutation, resulted in decreased affinity (Table 1 and Supplementary Fig. 14a, b) that are consistent with our actin cosedimentation assay results (Figs. 2e and 5c). Our data support the

conclusion that the unfolded α1-helix contributed to an apparent equilibrium shift towards an activated state of αcat-ABD.

Comparison of the $^{15}N/^1H$ TROSY NMR spectra of αNcat-ABD-WT[47] and αNcat-ABD-H1 showed that a region (αN-catenin residues 794–814) containing V796$_N$ and βH was one of three regions affected by the H1 mutation (Supplementary Fig. 15a), likely indicating an altered conformation in this region (Supplementary Fig. 15b). We further confirmed by NMR relaxation and MD simulations studies that the unfolded α1-helix increased molecular motions in the V796$_N$/βH region (Supplementary Fig. 15c, d). In addition, we performed chemical shift (CS)-based Rosetta comparative modeling (CM)[48] to show that V796$_N$ of αNcat-ABD-WT remained in the cryptic state, whereas αNcat-ABD-H1 displayed a large conformational change that exposed V796$_N$ on the surface (Fig. 6d), resembling V796 in the crystal structure of αEcat-ABD-WT (Fig. 5b). Our extended equilibrium MD calculations of αcat-ABDs support that the unfolded α1-helix accelerates the conformational change to favor the exposed state of V796 (Fig. 6e, f). As the exposure of V796$_N$ precedes complete unwinding of α1-helix during the constant-force simulation (Supplementary Movie 2), we expect that α1-helix unfolding 'locks' V796$_N$ in the activated state. Taken together, our observations indicate that allosteric coupling between α1-helix and the actin-binding residue V796 is central to the force-induced association of α-catenin with F-actin.

## Discussion

We show that the unique molecular features of αcat-ABD, α1-helix, V796, and βH, confer mechanosensitivity to α-catenin and its ability to dynamically regulate and reorganize actin filaments directly associated with cadherin-catenin complexes at intercellular junctions. The importance of α-catenin to directly associate with F-actin in a mechanosensitive manner is underscored by experiments showing that αcat-H1 with enhanced F-actin binding was equally inferior to αcatFL function as mutants with diminished F-actin binding (e.g., 3A) during mammalian and *Drosophila* wound healing, and *Drosophila* development (Figs. 3 and 4). Although a high-resolution structure of the αcat-ABD-F-actin complex remains to be solved, we have shown that the critical actin-binding site residues, L785, I792, and V796, are located away from the αcat-ABD mechanosensory motif, α1-helix, thus raising the possibility that the N-terminal region of ABD acts allosterically to regulate F-actin binding. Based on these observations, we propose that the coupled conformational states of α1-helix and V796 provide the structural basis of force-dependent allosteric regulation of the α-catenin-F-actin interaction.

In the proposed mechanism, the ABD of α-catenin in the attenuated state can weakly associate with F-actin, whereas its interaction with F-actin under force would trigger α1-helix unfolding and the exposure of V796 to form a catch bond interaction between the cadherin-catenin complex and F-actin at nascent contacts[8] (Fig. 7). As nascent contacts grow, multiple α-

catenin molecules will bind to F-actin in a cooperative manner[8,29] to promote the formation of cadherin-catenin complex clusters (Fig. 7). Although αcat-H1 or other constructs without the α1-helix can support AJ formation in R2/7 cells (Supplementary Fig. 1b and 8b), these do not restore normal α-catenin function (Supplementary Fig. 2a, b) and may reflect a lack of extensive junctional remodeling in these cells. A similar discrepancy between confluent R2/7 cells, wound-healing assays, and in vivo performance was noted for αEcat-NM_I (residues 1–402): this ABD-deficient construct forms AJs in R2/7 cells through the recruitment of vinculin[6], but does not support normal α-catenin function during wound closure[6], and a corresponding *Drosophila* construct (αCat-NM1) showed no rescue of αCat$^{-/-}$ embryos (R. S. and U.T., unpublished).

Cadherin clustering and AJ maturation likely require *trans*-interactions and *cis*-interactions of cadherin ectodomains, as well as an active process involving intracellular coupling of the cadherin-catenin complex to actin networks[49]. Our αNcat-ABD-H1 crystal structures revealed an unexpected ABD homo-dimerization (Fig. 2a), which can facilitate F-actin bundling in vitro (Fig. 2e). It involves the βH motif forming an extensive dimer interface with the hydrophobic patch uncovered by α1-helix unfolding (Fig. 2b and Supplementary Fig. 6c). Considering the very weak αcat-ABD dimerization (Fig. 2c and Supplementary Fig. 7c), which is marginally increased by the H1 mutation in solution (Fig. 2d), it is possible that tension-induced unfolding of α1-helix allosterically changes the conformational dynamics of

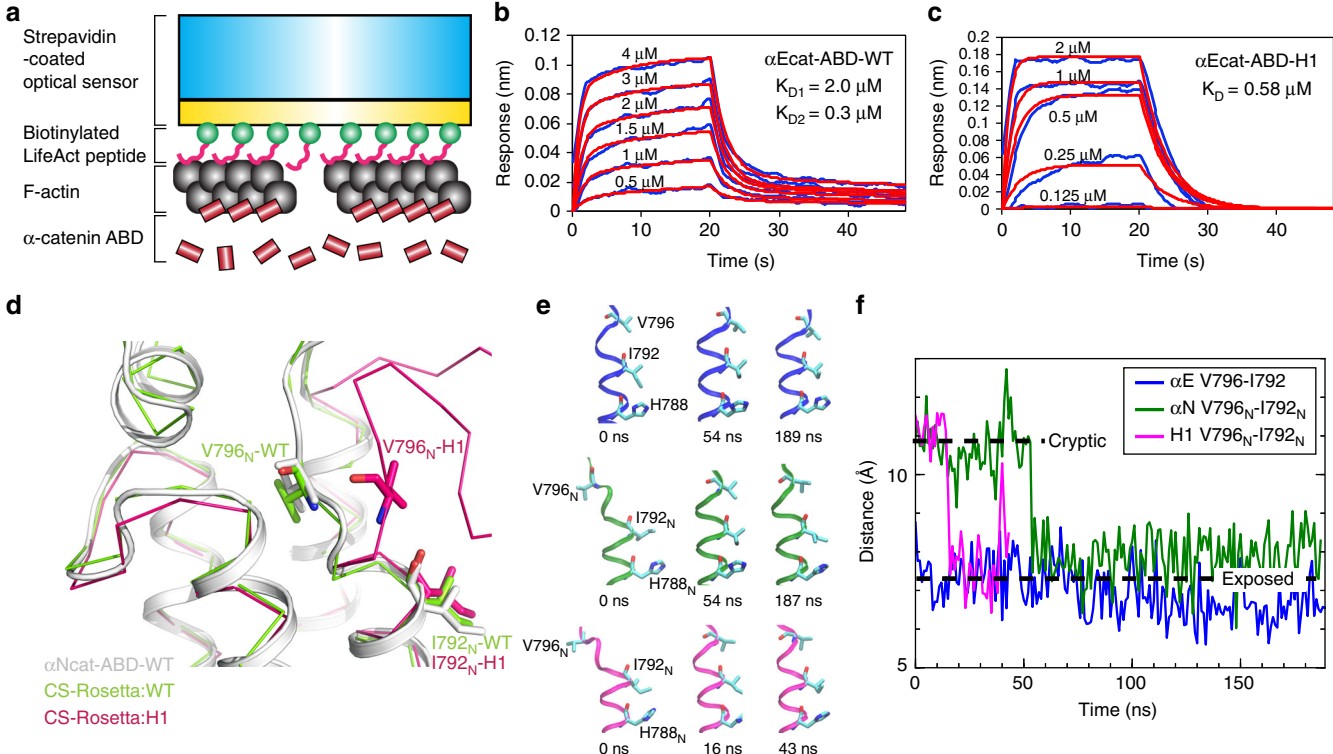

**Fig. 6** Unfolding of α1-helix affects the conformational dynamics of V796. **a** A scheme of BLI experiment for a kinetic analysis of direct interaction between αEcat-ABD and F-actin. The streptavidin-coated optical sensor with LAbio peptides immobilizes F-actin through high avidity, thereby restricting the movement of attached F-actin to minimize the occurrence of αcat-ABD-induced actin bundling. **b** BLI responses curves of the αEcat-ABD-WT. The $K_D$ values were obtained by fitting concentration-dependent F-actin binding curves (blue) to a 2:1 heterogeneous binding model (red curves). **c** BLI response curves of the αEcat-ABD-H1. The $K_D$ value was obtained by fitting concentration-dependent F-actin binding curves (blue) to a 1:1 binding model (red curves). **d** CS-Rosetta-CM models of αNcat-ABD-WT and αNcat-ABD-H1 based on NMR CS data and the αNcat-ABD-WT crystal structure as the template. **e** Conformational states of V796 during the equilibrium MD simulations of αEcat-ABD-WT (blue), αNcat-ABD-WT (green) and αNcat-ABD-H1 (magenta). Snapshots of the region of α5-helix containing V796 at specified time points are shown. **f** Evolution of distance between the β-carbon atoms of V796 and I792 during the equilibrium MD simulations. Dotted lines mark the approximate inter-residue distances when V796 is in the cryptic and exposed positions

**Table 1 BLI data for αEcat-ABD variants binding to F-actin**

| αEcat-ABD | Fitting model | $K_{D1}$ (µM) | $K_{D2}$ (µM) | $k_{on1}$ (1/Ms) | $k_{on2}$ (1/Ms) | $k_{off1}$ (1/s) | $K_{off2}$ (1/s) | $K_{D1}/K_{D2}$ (%)[b] |
|---|---|---|---|---|---|---|---|---|
| WT | 2:1 HL[a] | 2.0 | 0.3 | $2.34 \times 10^5$ | $3.19 \times 10^4$ | $4.65 \times 10^{-1}$ | $9.64 \times 10^{-3}$ | 82/18 |
| H1 | 1:1 | 0.58 | – | $4.55 \times 10^5$ | – | $2.64 \times 10^{-1}$ | – | – |
| L785A | 2:1 HL[a] | 27.8 | 4.8 | $2.49 \times 10^4$ | $2.21 \times 10^3$ | $6.90 \times 10^{-1}$ | $1.07 \times 10^{-2}$ | 71/29 |
| I792A | 2:1 HL[a] | 4.0 | 0.6 | $2.34 \times 10^5$ | $1.45 \times 10^4$ | $9.45 \times 10^{-1}$ | $8.81 \times 10^{-3}$ | 76/24 |
| V796A | 2:1 HL[a] | 3.7 | 1.8 | $2.11 \times 10^5$ | $6.91 \times 10^3$ | $7.73 \times 10^{-1}$ | $1.25 \times 10^{-2}$ | 63/37 |
| 3A | 2:1 HL[a] | 38.4 | 6.1 | $2.08 \times 10^4$ | $1.40 \times 10^3$ | $7.99 \times 10^{-1}$ | $8.49 \times 10^{-3}$ | 65/35 |
| V714A | 2:1 HL[a] | 2.6 | 0.7 | $2.21 \times 10^5$ | $2.01 \times 10^4$ | $5.68 \times 10^{-1}$ | $1.35 \times 10^{-2}$ | 77/23 |

[a] 2:1 heterogeneous ligand (HL) model provides two sets of kinetics parameters ($k_{on1}$, $k_{off1}$, $K_{D1}$) and ($k_{on2}$, $k_{off2}$, $K_{D2}$)
[b] The percentage of two kinetic interactions in the total binding was determined based on $R_{max}$ values

the actin-binding site without affecting dimerization. Nonetheless, AJ-localized α-catenin cooperatively binding to F-actin would likely increase the propensity of αcat-ABD to dimerize and promote F-actin bundling. Uncovering this ABD dimerization interface motivated us to propose a new monomer-dimer model for α-catenin at the cadherin-actin interface (Fig. 7). Although both in vitro[8] and in vivo[7] studies consistently concluded that monomeric α-catenin forms the essential link between the cadherin-β-catenin complex and F-actin, the current model fails to account for the capacity of α-catenin to bundle F-actin[5] at AJs. The ABD-dependent dimerization as demonstrated here allows actin filaments to be tightly bundled in an antiparallel fashion (Fig. 8) and places the α3-α4 surface of ABD in close proximity with F-actin, which is consistent with our NMR saturation transfer data (Fig. 2f). In addition, the ABD dimerization allows F-actin bundling to occur while the N domain of α-catenin remains associated with cadherin-bound β-catenin (Fig. 7). Hence, our proposed model differs from the previous monomer-dimer model of α-catenin by arguing that (i) the E-cadherin/β-catenin/α-catenin/F-actin complex regulates the cadherin-actin linkage without disrupting the β-catenin-α-catenin interaction; (ii) that α-catenin as a component of the complex can bundle F-actin, and (iii) that α-catenin controls actin binding through force-dependent allosteric regulation of the actin-binding site within the ABD. The versatility of α-catenin to modulate the attachment of the cadherin-catenin complex to F-actin from transient interaction to stable actin bundling, and to the dynamic cortical actin network[50], will likely involve additional dynamic connections provided by the recruitment of other F-actin-binding proteins, such as vinculin, afadin, ZO-1 and EPLIN, to intercellular junctions[6,27,51–53].

We have employed an integrative structure/function approach to show that the structural motifs of αcat-ABD involved in the regulation of tension-sensitive actin binding are essential for normal tissue morphogenesis and wound healing. Although the occurrence of actin bundling involving ABD-linked α-catenin dimers at intercellular junctions remains to be tested, our model reconciles previous observations of α-catenin as a critical mechanosensor engaged in reorganization of AJs by facilitating dynamic F-actin association[8,21], and actin bundling through homodimerization[5,20]. Moreover, the significance of this mechanism lies in the ability of α-catenin to modulate cadherin-mediated cell adhesion through force-dependent F-actin binding and actin remodeling without dissociation from the cadherin-β-catenin complex.

## Methods

**Protein expression and purification.** The cDNA corresponding to the actin-binding domain (ABD) of mouse αE-catenin (652-906), mouse αN-catenin (651-905) and all related mutants (e.g., the αE-catenin H1 mutation RAIM670-673GSGS) were amplified by PCR and individually subcloned into the pGEX4T1

vector (GE Healthcare). The fly αcat-ABD (659-917) cDNA was amplified by PCR and subcloned into a modified pET-SUMO vector. Site-directed mutagenesis was performed using the Quickchange protocol (Stratagene) to produce all single-/multiple-residue and deletion mutants. Recombinant proteins were expressed as N-terminal glutathione S-transferase (GST) fusion proteins in *Escherichia coli* BL21-CodonPlus cells. Cells were grown to an O.D.600 of 0.8 at 37 °C and the recombinant protein expression was induced with 0.5 mM isopropyl β-D-1-thiogalactopyranoside for 16 h at 16 °C. Cells harvested by centrifugation were resuspended in the lysis buffer (50 mM Tris-HCl, pH 8.0, 300 mM NaCl, 10 mM β-mercaptoethanol, 1 mM Tris(2-carboxyethyl)phosphine (TCEP)), sonicated on ice, and subjected to centrifugation to isolate soluble proteins. GST-fusion proteins were isolated using the glutathione-sepharose resin (GE Healthcare). His-SUMO fusion proteins were isolated using the Ni²⁺-NTA resin (ThermoFisher Scientific). GST-fusion and His-SUMO proteins were cleaved by thrombin or SUMO protease (Ulp-1), respectively. The cleaved proteins were further purified by size-exclusion chromatography using Superdex 75 (GE Healthcare) in the running buffer (50 mM Tris-HCl, pH 8.0, 300 mM NaCl, 1 mM TCEP). The purified proteins were exchanged into protein storage buffer (50 mM Tris-HCl, pH 8.0, 100 mM NaCl, 1 mM TCEP).

**Size-exclusion chromatography-multiangle light scattering.** Purified protein (5 mg/mL, 100 µL injection volume) was subjected to size-exclusion chromatography (SEC) using a Superdex-200 Increase 10/300 GL column (GE Healthcare) equilibrated in SEC-MALS buffer (20 mM Tris-HCl pH 7.0, 100 mM NaCl) at a flow rate of 0.5 mL/min. Multi-angle light scattering (MALS) measurements were performed in-line with SEC by using a three-angle (45°, 90°, and 135°) miniDawn light-scattering instrument and an Optilab rEX differential refractometer (Wyatt Technologies). Molecular weight was calculated by using the ASTRA software (Wyatt Technologies). Dimer peak area integration was performed by using ImageJ[54]. Statistical analysis was performed by Two-way ANOVA followed by Bonferroni's comparison test.

**Actin cosedimentation assay.** Monomeric rabbit skeletal muscle actin was purified from rabbit muscle acetone powder[55] (Pel-Freez Biologicals). Purified globular actin (G-actin) was diluted to 20 µM in a fresh Buffer-G (2 mM Tris-HCl, pH 8.0, 0.2 mM ATP, 0.5 mM DTT, 0.1 mM CaCl₂), and subsequently polymerized in Buffer-F (5 mM Tris-HCl, pH 8.0, 50 mM KCl, 2 mM MgCl₂, 1 mM ATP, 0.2 mM CaCl₂, 0.5 mM DTT) for 1 h at RT. The αcat-ABD samples were subjected to buffer exchange into Buffer-F. Samples of F-actin and ABD were mixed (the protein mixture contains 5 µM ABD and 5 µM actin in 50 µL) in Ultra-Clear Centrifuge Tubes (Beckman Coulter) and incubated for 1 h at RT. F-actin with bound protein samples were cosedimented by centrifugation using a Beckman Coulter Airfuge with a chilled A-100/30 rotor at 28 psi (≥100,000×g) for 20 min at RT. To assess actin bundling, F-actin with bound protein samples were cosedimented by centrifugation using a benchtop microcentrifuge at low relative centrifugal force (RCF; 10,000×g) for 30 m at 4 °C. Supernatant and pellet fractions were analyzed by SDS-PAGE with coomassie blue stain. Gel band intensity was measured by using ImageJ[54]. Statistical analysis of three or more groups was performed by One-way ANOVA followed by Tukey's multiple comparison test. Statistical analysis of two groups was performed by Two-way ANOVA followed by Bonferroni's comparison test.

**Crystallization and data collection.** Crystals of the αN-catenin ABD-H1 were grown at 277 K by vapor diffusion. For crystallization the αNcat-ABD-H1 sample was exchanged into Buffer-P (20 mM K/Na phosphate, pH 6.0, 150 mM NaCl, 1 mM TCEP), and the protein solution (30 mg/mL) was mixed with an equal volume of the reservoir solution, which consists of either solution A (2.0 M (NH₄)₂SO₄, 10 mM CoCl₂) for form A crystals, or solution B (100 mM Na acetate/acetic acid, pH 4.5, 0.8 M NaH₂PO₄, 1.2 M K₂HPO₄) for form B crystals. Similarly, crystals of the αEcat-ABD-WT in Buffer-P (30 mg/mL) were grown at 277 K by vapor diffusion with the reservoir solution consisting of 0.2 M KBr, 2.2 M (NH₄)₂SO₄ and 3% (w/v)

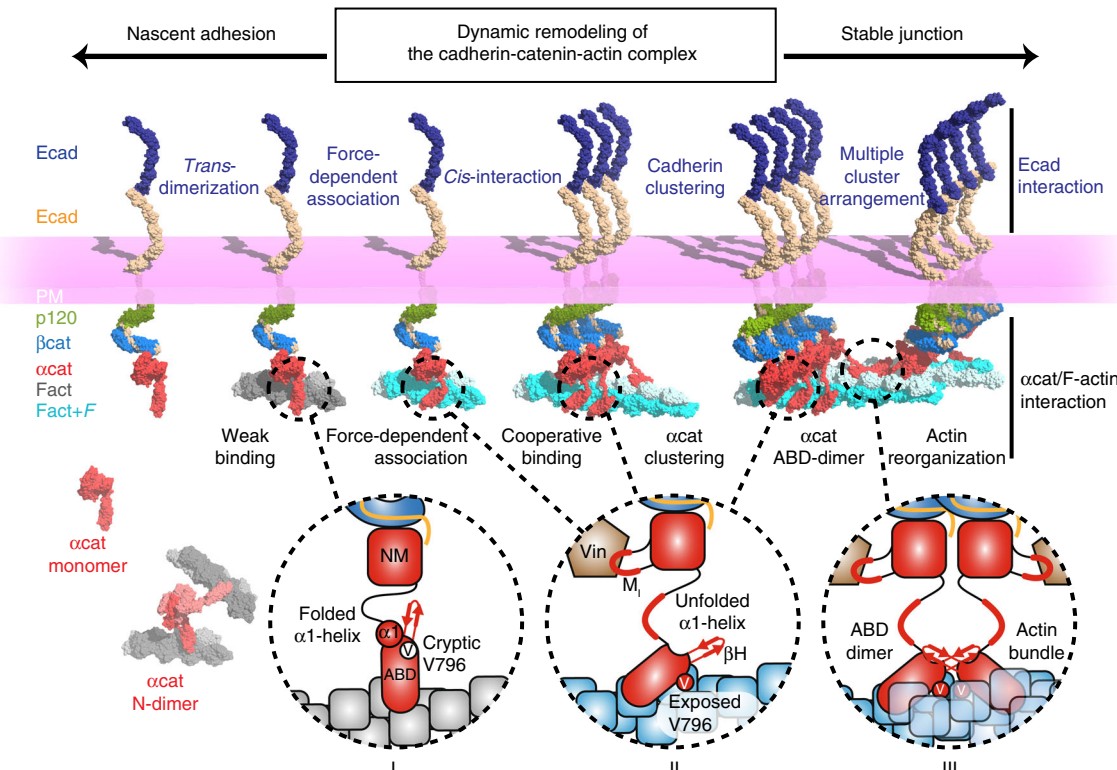

**Fig. 7** Dynamic remodeling of the cadherin-catenin-actin complex. A model of α-catenin-dependent cadherin-actin linkage, cadherin clustering and F-actin bundling involved in the regulation of cadherin-mediated cell-cell adhesion facilitating nascent and stable junctions. The ABD of α-catenin bound to the cadherin-β-catenin complex is in an attenuated state with the folded α1-helix and cryptic V796 to form weak interactions with F-actin (I). α-catenin dissociated from β-catenin can exist as a monomer and an N-terminally linked homodimer (N-Dimer). When the cadherin-catenin complex encounters F-actin under force (Fact + F), αcat-ABD exposes V796 on the surface while the α1-helix unfolds to form a catch bond with F-actin (II). The force propagates through α-catenin to unfold the M$_I$ region, which facilitates the recruitment of vinculin (Vin) to AJs. Strong F-actin binding promotes cooperative binding of ABD. As α-catenin clusters together on F-actin, ABD dimerization between two ABD-coated actin filaments promotes actin bundling and lateral clustering of cadherin-catenin complexes at AJs (III)

D-Galactose. Crystallization of αEcat-ABD-H1 was unfruitful. Crystals were briefly soaked in crystallization solution containing 25% glycerol for data collection at 100 K. Diffraction data were collected at the Canadian Light Source-Canadian Macromolecular Crystallography Facility (CMCF) beamline 08ID-1 (Saskatoon, Canada) and processed with HKL2000[56]. Br-SAD data were collected with αEcat-ABD crystals. Statistics pertaining to the diffraction data are presented in Supplementary Table 2.

**Crystal structure determination and refinement.** Crystal structures of the αNcat-ABD-H1 in forms A and B were determined at 2.2 and 2.8 Å resolution, respectively. The structure solution was solved by molecular replacement using PHASER[57] with the αNcat-ABD-WT crystal structure (PDB ID: 4K1O) as a search model. Successive rounds of manual model building and refinement were performed by using Coot[58] and PHENIX[59] to refine the models of αNcat-ABD-H1. The crystal structure of the αEcat-ABD-WT was initially determined at 2.3 Å resolution by the single-wavelength anomalous dispersion method, and further refined at 2.2 Å by using PHENIX. Refinement statistics are presented in Supplementary Table 2. Molecular graphics representations were prepared using PyMOL (http://www.pymol.org/).

**NMR spectroscopy.** The NMR experiments of $^{15}$N/$^{13}$C labeled αNcat-ABD-WT and αNcat-ABD-H1 were performed on Bruker AVANCE II 800 MHz (Bruker Biospin) spectrometer equipped with a cryogenic triple-resonance z-gradient probe. Labeled proteins were expressed in *E. coli* BL21-CodonPlus with M9 minimal media containing $^{15}$N-ammonium chloride and $^{13}$C-glucose for 15 h at 288 K. The purification of labeled ABD proteins was performed in a similar manner as described above. The backbone assignment of αNcat-ABD-H1 was processed using standard $^{1}$H-$^{15}$N experiments. $^{15}$N relaxation data were acquired at 288 K in the presence and absence of a 3 s $^{1}$H saturation period prior to $^{15}$N excitation using the $^{15}$N-$^{1}$H heteronuclear NOE pulse sequence[60]. NMR spectra were processed using NMRPipe[61] and resonance assignment was carried out using

NMRView[62]. Errors in peak intensity values were estimated from the signal-to-noise ratio of each spectrum.

The transferred cross saturation (TCS) experiments were performed at 293 K to detect the resonances of αNcat-ABD-WT in the free state after binding to F-actin in solution. The $^{15}$N/$^{2}$H-labeled αNcat-ABD-WT was mixed with unlabeled F-actin at the molar ratio of 1:0.1 (ABD:G-actin) in the modified actin polymerization buffer (20 mM Tris-HCl, pH 7.0, 150 mM NaCl, 0.2 mM ATP, 0.1 mM CaCl$_2$, 1 mM TCEP, 50 mM KCl, 2 mM MgCl$_2$) containing 12% H$_2$O to avoid the dipole coupling between the amides[63]. Control TCS experiments were carried out without F-actin to assess the effects of the residual aliphatic protons in the ABD.

**CS-Rosetta-CM.** NMR chemical shift (CS)-guided structure modeling of the αNcat-ABD-WT and αNcat-ABD-H1 (28 kDa) was performed by employing the CS-Rosetta-CM approach[48] with NMR chemical shift data (αNcat-ABD-WT and αNcat-ABD-H1) and the αNcat-ABD-WT crystal structure (PDB ID: 4K1O) as the template. This approach enables CS-Rosetta modeling to be effective for proteins larger than 15 kDa. The POMONA server (https://spin.niddk.nih.gov/bax/nmrserver/pomona/) was used to prepare the Rosetta input files.

**Biolayer interferometry.** To determine a dissociation constant for the αcat-ABD-F-actin interaction, we devised a biolayer interferometry approach which uses label/modification-free F-actin and minimizes any occurrence of actin bundling. All BLI experiments were performed at 26 °C using Octet384 (Fortebio). All proteins used in BLI experiments were buffer exchanged into the assay buffer (2 mM Tris-HCl, pH8.0, 0.2 mM ATP, 0.5 mM DTT, 0.1 mM CaCl$_2$, 50 mM KCl, 2 mM MgCl$_2$, 0.1% BSA, 0.02% Tween-20). F-actin was polymerized for 1 h at RT, and subsequently diluted to 1 μM for the assay. We first load the optical surface of the Streptavidin (SA) biosensors with the widely-used F-actin-binding peptide LifeAct[46] containing a C-terminal biotinylation (LAbio) at the concentration of 2 μg/mL. The SA sensors coated with the N-terminally biotinylated LA (bioLA) did not produce any response signals in the presence of F-actin (Supplementary

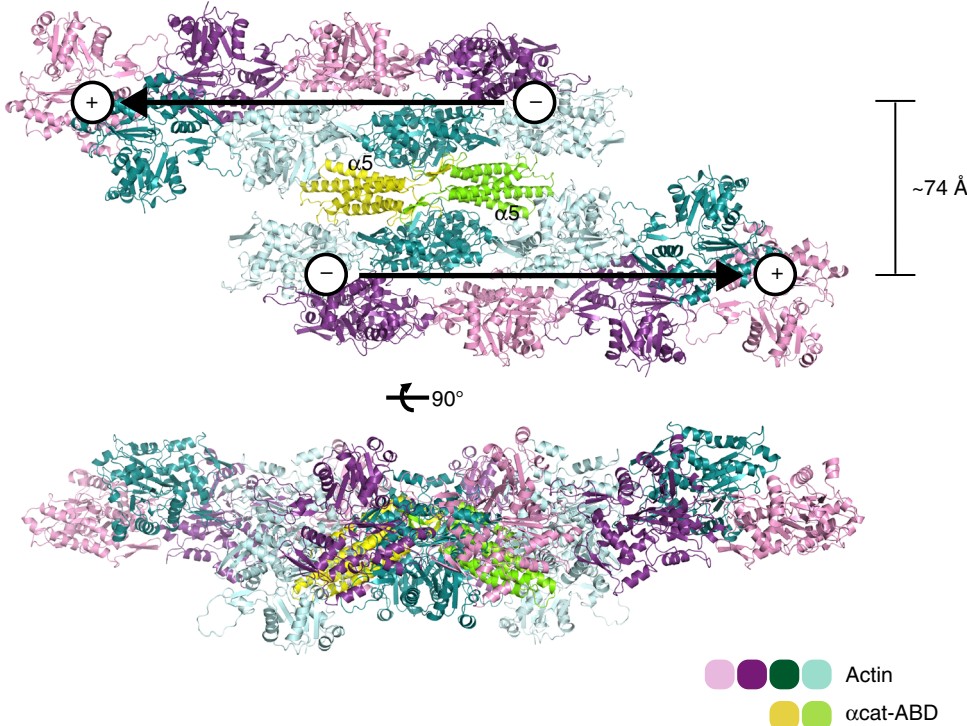

**Fig. 8** A molecular model of the αcat-ABD homodimer facilitating actin bundling. The αcat-ABD/F-actin complex model based on the vin-ABD/F-actin cryo-EM structure suggests that two αcat-ABD molecules discretely bound to different actin filaments (e.g., αcat-ABD binds to F-actin through the α5-helix) can homodimerize through the crystallographically-identified ABD-dimer interface without causing any steric clash with actin filaments. We propose that the ABD-dependent dimerization of α-catenin facilitates actin bundling in an anti-parallel manner ('+' and '-' indicate the barbed and pointed ends, respectively). The tight spacing of actin filaments (7-8 nm) in our atomic model is consistent with the inter-filament distances observed within the actin rods formed by ABD*-expressing cells (Supplementary Fig. 2c)

Fig. 13a). The SA-LAbio sensor was used to immobilize F-actin through the LifeAct-F-actin interaction with high avidity. To measure the sub-micromolar affinity binding of αEcat-ABD-H1, the Super-Streptavidin (SSA) biosensors with increased Streptavidin density were used. The SA-LAbio-F-actin sensors were incubated with αcat-ABD at various concentrations for the association step, and transferred to a buffer to monitor its dissociation. Octet analysis software (Fortebio) was used to perform the kinetic analysis of ABD-F-actin interaction.

**Circular dichroism**. Circular dichroism spectroscopy data for 40 μM αcat-ABD samples were collected on a Jasco J-815 CD spectrometer (Jasco, Tokyo, Japan) at 293 K using a 0.1 cm path length cuvette with a scanning speed of 20 nm/min (1 nm increments). Thermal denaturation data were acquired at 220 nm with a scan rate of 1 °C/min.

**Sequence alignment**. Multiple sequence alignment of human αE-catenin (accession P35221), mouse αE-catenin (accession NP_033948.1), mouse αN-catenin (accession NP_663785.2), mouse αT-catenin (accession Q65CL1.2), *Drosophila* α-Catenin (P35220.2), and mouse vinculin (accession Q64727) was performed by using T-Coffee (http://tcoffee.crg.cat)[64] and ESPript (http://espript.ibcp.fr)[65].

**Mammalian cell culture**. The αE-catenin cDNA was amplified by PCR and subcloned into a modified pCAH vector[66] to express αE-catenin-mGFP (monomeric EGFP with the dimerization-disrupting A206K mutation[67]). The full-length or deletion variant αE-catenin cDNA was subcloned into pCA vector to express αE-catenin-FLAG. Site-directed mutagenesis was performed using the Quikchange protocol (Stratagene) to produce αE-catenin deletion mutants, $NM_I ABD$, $NM_I ABD*$ and $NM_I$. with the constitutively active $M_I$/vinculin-binding site (VBS; residues 305-352). R2/7 cells were cultured in Dulbecco's modified Eagle's medium (Thermo Fisher Scientific) supplemented with 10% v/v fetal bovine serum (Invitrogen), 100 U/ml penicillin, and 100 mg/ml streptomycin (Sigma-Aldrich). Cells were transfected with expression vectors by using Lipofectamine 3000 (Thermo-Fisher Scientific), and cells with stable protein expression was selected based on antibiotic resistance (300 μg/mL hygromycin for pCAH vectors) and subsequently isolated by flow cytometry. For live imaging analysis of αE-catenin-mGFP-expressing R2/7 monolayers post-scratch wounding, cells were also infected with a lentiviral LifeAct-Ruby purchased from Addgene (pLenti.PGK.LifeAct-Ruby, Addgene Plasmid #51009).

**Antibodies**. The following primary antibodies were used: hybridoma mouse anti-α-catenin (5B11; undiluted), polyclonal rabbit anti-phospho-serine 641 αCat (21330; 1:300, Signalway Antibody), monoclonal mouse anti-GAPDH (9484; 1:5000, Abcam). Secondary antibodies for Western blotting included fluorescently labeled donkey anti-mouse and anti-rabbit antibodies (680RD or 800RD; 1:5000, LiCor Biosciences). Secondary antibodies for immunofluorescence included IgG Alexa Fluor 488 or 568-conjugated goat anti-mouse or anti-rabbit antibodies (1:200, Invitrogen). For R2/7 cells expressing αE-catenin-FLAG, the following antibodies were used: anti-DDDDK tag rabbit polyclonal antibody, which recognizes the FLAG tag (1:200, MBL), anti-ZO-1 mouse monoclonal antibody (T8-754; undiluted, a gift from Sa. Tsukita, Osaka University, Japan), anti-E-cadherin mouse monoclonal antibody (clone36/E-cadherin; 1:100, BD) and anti-E-cadherin rat monoclonal antibody (ECCD2; 1:200, a gift from M. Takeichi, Riken, Japan). Alexa Fluor 488-conjugated or 555-conjugated secondary antibodies (1:200) were purchased from Invitrogen. Alexa Fluor 647-conjugated phalloidin (1:50, Invitrogen) was used for staining actin filaments.

**Fluorescence microscopy and confocal imaging**. Cells were grown on coverslips, fixed in 2% paraformaldehyde (Electron Microscopy Services, Hatfield, PA) for 15 m, quenched with glycine, permeabilized with 0.3% Triton X-100 (Sigma), and blocked with normal goat serum (Sigma). Primary and secondary antibody incubations were performed at RT for 1 h, interspaced by multiple washes in PBS, and followed by mounting coverslips in ProLong Gold fixative (Life Technologies). Confocal images were acquired with an Olympus IX81 inverted confocal microscope (Olympus) through a Plan-Apochromat x60/1.42 NA oil-immersion objective lens (Olympus). Images were processed by Olympus FV10-ASW.

For R2/7 cells expressing αE-catenin-FLAG, images were taken using Olympus BX51 microscope with a UPlanFLN 40 × /0.75 Ph2 lens or a PlanApo 60 × /1.40 Oil Ph3 lens.

The images of R2/7 cells expressing αE-catenin variants at the scratch wound fronts were collected on a Nikon Ti Eclipse inverted microscope equipped with a Yokogawa CSU-X1 spinning disk head, Perfect Focus system (Nikon), a 100 × 1.49 NA APO TIRF objective, and an Andor xION EMCCD camera (Andor

Technology), controlled by MetaMorph 7.7.7.0 software (Molecular Devices). Cells were maintained at 37 °C plus 5% $CO_2$ during imaging using a Tokai-Hit stage-top incubator (Tokai-Hit) or an Okolab gas mixer (Okolab).

For imaging z-stacks of cell piling, slides were imaged at RT with 60× oil Apo TIRF NA 1.49 objective on the Nikon A1R laser scanning confocal inverted microscope equipped with two standard PMTs (408, and 640) and two high sensitivity GaAsP detectors (488, 561).

**Scratch wound assay**. 250,000 R2/7 cells were plated for 24 h on LabTek #1 4-well chamber slide (43300-776, Thermo Fisher Scientific), wounded with a P200 micropipette tip and cells were allowed to recover for 2 h. Prior to imaging, DMEM media was replaced with FluoroBright DMEM (Life Technologies) and 10 g/mL Mitomycin C (Sigma) to limit cell proliferation. Cells in Fig. 3d were imaged with the 20× objective every 10 min (both phase contrast and fluorescent channels) on the Nikon Biostation IM-Q with the slide holder module (located in Northwestern University Nikon Imaging Facility) at 37 °C, 5% $CO_2$, for 15 h. 10–12 fields of view (FOV) were captured along the wound edge. Instrument controlled by Biostation IM software, version 2.21 build 144. To quantify change in wound area, the resulting .ids file was imported to ImageJ[54], and the wound edge of the phase-contrast image was traced with the polygon tool at time = 0 and time = 15 h. The area of the resulting polygon was measured in pixels² and the resulting data (35 FOV; 5BR) were compared by performing One-way ANOVA statistical analysis followed by Tukey's multiple comparison test. Images presented in the paper were adjusted for brightness/contrast but were otherwise unprocessed.

To determine the velocity (μm/sec) and persistence (total path length (μm)/net displacement (μm)) of cells at the wound front, tracked individual cells at each interval of the time lapse sequence using the MTrackJ plug-in (FIJI) (see colored tracks in Supplementary Fig. 9b). The center of the nucleus (centroid) was hand-marked at each interval. Data points for each of the 5 constructs were compiled from scratch wounds carried out on three different days to ensure biological reproducibility and significance by ANOVA.

To validate cell trajectories for the WT versus H1 αE-catenin mutant comparison using a fully unbiased method, a particle image velocimetry (PIV) analysis (i.e., template matching by cross-correlation) was carried out using the PIV_jar plug-in (FIJI). Briefly, each acquired image frame was divided into an interrogation window size that approximated single cells or a small cell group depending on degree of cell packing (100 μm²). Vectors were averaged from all 91 frames/movie for 3 αEcat-WT and 4 αEcat-H1 wounds (two biological replicates from separate days).

**Epithelial sheet disruption assay**. R2/7 cells expressing α-catenin variants were plated in a 12-well culture plates (Corning) and allowed to reach confluency. After 36 h, the monolayer was washed 2x in DPBS supplemented with 0.5 mM $Mg^{2+}$ and 1 mM $Ca^{2+}$ (HyClone) and then incubated for 30 m at 37 °C in 1 mg/ml Dispase (Roche) diluted in PBS and supplemented with the indicated amount of $Ca^{2+}$. Subsequently monolayers were disrupted by subjecting the plate to a shaking force of 1400 rpm for 15 s. An image of each well was captured before and after shaking, and monolayer fragments were counted. Seventy-five epithelial fragments were established as the upper limit for counting. The assay results were collected from 3 technical replicates. Statistical analysis was performed by One-way ANOVA followed by Tukey's multiple comparison test.

**Measurement of transepithelial electric resistance**. Confluent monolayers of R2/7 cell lines expressing various α-catenin mutants grown in Transwell[TM] chamber were used. Transepithelial electric resistance (TER) was measured using a Millicell-ERS epithelial volt-ohmmeter (Millipore) and normalized by the area of the monolayer. The background TER of blank Transwell[TM] filters was subtracted from the TER of cell monolayers. Statistical analysis was performed by One-way ANOVA followed by Tukey's multiple comparison test.

**Electron microscopy**. Cells were fixed with 2% paraformaldehyde, 2.5% glutaraldehyde in 0.1 M sodium cacodylate buffer, pH 7.5 for 2 h at RT. Samples were conventionally dehydrated and embedded in resin (Polybed 812; Polyscience). Ultrathin sections were cut and stained with uranyl acetate and lead citrate before observation with an electron microscope (JEM-1010; JEOL).

**Molecular dynamics simulations**. The structure of the αNcat-ABD-WT was obtained from the Protein Data Bank (PDB ID 4K1O[27]). The structure of αNcat-ABD-H1 was obtained from the form A crystal structure of αN-catenin ABD-H1. The structure of αEcat-ABD-WT was obtained from the crystal structure of αEcat-ABD-WT chain A. All-atom MD simulations were performed using NAMD2[68], the CHARMM27 force field for proteins and ions[69,70], and the TIP3P model for explicit water[71]. The Solvate and Autoionize plugins of VMD were used to solvate the system in a water box with at least 15 Å between the protein and the boundary of the box, and to add 150 mM NaCl and neutralize the net charge of the system, respectively[72]. All simulations were performed using periodic boundary conditions and a time step of 2 fs. A constant temperature of 310 K was maintained using Langevin dynamics with a damping coefficient of 0.5 ps⁻¹(refs. [73,74]). Short-range, non-bonded interactions were calculated using a cutoff distance of 12 Å, and long-

range electrostatic forces were described with the Particle Mesh Ewald method[75]. Throughout the simulations, bond distances involving hydrogen atoms were fixed using the SHAKE algorithm[76]. After an initial 10,000 steps of energy minimization with all Cα atoms fixed, the system was equilibrated in an NVT (constant Number, Volume, and Temperature) ensemble for 500 ps, during which all protein Cα atoms were fixed to allow relaxation of the side chains and water. Subsequent equilibrium simulations were performed in an NVT ensemble. In constant-force SMD simulations of the αcat-ABD, a 100-pN pulling force was applied to the Cα atom of N-terminal residue 668/669 (αN-/αE-catenin). It should be noted that the force-dependent binding between α-catenin and F-actin was previously detected in vitro at ~5 pN[8]. We chose a 100-pN pulling force for the SMD simulations by considering the trade-off between the improved accuracy and longer time required to observe conformational changes under low force. Assuming the SMD simulation successfully captures the physiological unfolding pathway, we expect unfolding of α1-helix to occur at lower force (<100 pN) over a longer timescale (>60 ns). The center of mass (the sulfur atom of M722/723) was fixed to prevent the overall translation of the system in response to the applied external force. The force was directed along the x-axis between the two anchoring points, and the vector pointing from the center of mass to the N-terminal residue was defined as the positive direction. Simulation outputs were analyzed using VMD[72] and plotted using the matplotlib plotting library[77]. The portion of each equilibrium simulation during which the RMSD monotonically increased (the first 7–13 ns) was excluded from analysis to avoid biasing the results toward the initial coordinates.

**3D cell culture**. We observed for the ability of R2/7 cells expressing αE-catenin variants to form spheroid structures by using Non-adherent Corning 96-well round bottom ultra-low attachment microplate (Corning). In each well, 25–200 cells were suspended in 100–200 μL of DMEM containing 10% FBS. The microplate was placed in a $CO_2$ incubator, and occasionally taken out of the incubator to acquire phase contrast images with a stereomicroscope equipped with a digital camera at various time points.

***Drosophila* stocks, overexpression, and rescue experiments**. *Paired-GAL4*, *Act5c-GAL4*, and *da-GAL4* were obtained from the Bloomington Drosophila stock center (https://bdsc.indiana.edu). Transgenic constructs inserted at attP2 were recombined with α-Cat[1] and balanced over *TM3, Ser, twi-Gal4, UAS-GFP*. These flies were crossed to *act-Gal4 da-Gal4 α-Cat[1]/TM3, Ser twi-Gal4 UAS-GFP* and eggs were collected on apple juice agar plates at 25 °C.

For quantification of whole-animal rescue experiments, a total of 100–300 fertilized non-GFP embryos were collected, allowed to develop at 25 °C, and monitored daily. Dead embryos were mounted in Hoyer's medium and lactic acid (1:1 ratio) for examination of the embryonic cuticle. Lethality counts were performed on larvae that hatched, at each of the stages of *Drosophila* development indicated below. A specific score from −2 to 8 was given to each rescued and control animal with a score of 0 denoting the phenotype most frequently observed in α-Cat[1] zygotic null mutant embryos[9]. The following scoring criteria were applied to measuring the extent of enhancement or rescue of the α-Cat mutant phenotype: (−2) embryonic lethal with both a severe head defect ('head open') and a dorsal open phenotype indicating a failure of dorsal closure; (−1) embryonic lethal with both the head open defect and a hole in the dorsal epidermis indicating incomplete closure; (0) embryonic lethal with a head open defect; (1) embryonic lethal with weak head defects ('abnormal head'); (2) embryonic lethal with normal head; (3) lethal at first larval instar; (4) lethal at second larval instar; (5) lethal at third larval instar; (6) early pupa lethal; (7) late pupa lethal; (8) adult. Statistical significance was assessed with a non-parametric Kolmogorov-Smirnov test. Data are presented as mean plus SD.

**Immunohistochemistry and histology of *Drosophila* embryos**. To prepare embryonic cuticle, embryos were de-chorionated in 50% bleach for 5 min, washed with 0.1% TritonX-100, mounted in 1:1 solution of Hoyer's and lactic acid and incubated overnight at 60 °C. Embryos were examined under phase contrast using a Zeiss Axiophot microscope.

The heat fixation method[78] was used for antibody staining of α-Cat[1] embryos expressing UAS-driven transgenes with *da-Gal4*. Embryos over-expressing transgenes using *Paired-Gal4* were fixed in a 1:1 mixture of 5% formaldehyde in 1× PBS and Heptane for 20 m. In both cases, embryos were devitellinized using a 1:3 mixture of Heptane and Methanol. Primary antibodies used were: rat mAb anti-HA (3F10, 1:500, Sigma), guinea pig pAb anti-α-Catenin (p121; 1:1000, Sarpal et al., 2012), mouse mAb anti-Arm (N2-7A1, 1:50; Developmental Studies Hybridoma Bank [DSHB]). Fluorescent secondary antibodies were used at a dilution of 1:400 (Jackson ImmunoResearch Laboratories and Invitrogen). Samples were analyzed on a Leica SP8 scanning laser confocal microscope using a 40x oil immersion lens (NA 1.3). Images were prepared and assembled in ImageJ, Adobe Photoshop, and Adobe Illustrator.

**Generation of *Drosophila* transgenes**. To generate the *Drosophila* UASp-αCat constructs, full-length αCat (2751 nucleotides) was cloned into Gateway pENTRTM/D-TOPO entry vector (Invitrogen) digested with Not1 and Asc1, using 3-part Gibson assembly reaction (NEB). αCat cDNAs carrying various mutations

were cloned using 2-part or 3-part Gibson assembly reactions using UASp-αCat in pENTRTM/D-TOPO as the backbone digested with following restriction enzymes: Not1 and Blp1 to generate αCat-H1 and αCat-Δα1; Blp1 and Asc1 to generate αCat-ΔβH, Not1 and Asc1 to generate αCat-H1ΔβH; Blp1 and Asc1 to generate αCat-3A. The Gateway® LR® Clonase Enzyme mix was used to clone all entry vector constructs into the pPWH (pUASP-Gateway Cassette with C-terminal 3x HA) vector containing an attB recombination site that was added using the NSi1 restriction site. Transgenic animals were produced by Best Gene Inc., by using flies carrying the attP2 recombination site on the left arm of the third chromosome. Amino acids of α-Cat proteins are: αCatFL (1–917), αCat-H1 (683REAM > 683GSGS), αCat-Δα1 (1–658/691–917), αCat-ΔβH (1- 811/824–917), αCat-H1ΔβH (1–811/824–917 + 683REAM > 683GSGS), αCat-3A (L798A + I805A + V809A), and αCat-ΔABD (1–708)[7].

**Immunoblotting with fly samples**. Transgenic flies were crossed to da-Gal4. An overnight collection of embryos was dechorionated and homogenized in cold lysis buffer (RIPA-25 mM Tris-HCl, pH 7.6, 150 mM NaCl, 1% TritonX-100, 1% sodium deoxycholate, 0.1% SDS) containing protease inhibitors (Roche). The protein concentrations were measure using the Bradford assay. 25 μg of total protein lysates were resolved by SDS-PAGE. Immunoblotting was performed according to recommended protocols provided by Life Technologies iBlot 2® Dry Blotting system. Primary antibodies used were anti-HA (rat monoclonal 3F10, 1:500; Abcam), anti-β-tubulin (mouse monoclonal, E7-1:1000; DSHB) and anti-α-Cat (guinea pig polyclonal, p121; 1:1000; Sarpal et al., 2012). Secondary antibodies used were: anti-rat 680LT (Goat-1:5000; LI-COR), anti-mouse 800CW (Donkey-1:5000, LI-COR) and anti-guinea pig 800LT (1:5000, LI-COR). Protein bands were visualized using the LI-COR Odyssey® Fc Dual-Mode Imaging system.

**Wound healing assay**. For time-lapse imaging stage 15 Drosophila embryos were dechorionated in 50% bleach for 2 min, rinsed with water, and mounted ventrolateral-side-down onto a coverslip using heptane glue. Embryos were covered with a 1:1 mix of halocarbon oil 27:700 (Sigma-Aldrich), and imaged using a Revolution XD spinning disk confocal (Andor) with a 60x oil-immersion lens (NA 1.35; Olympus). Images were captured with an iXon Ultra 897 camera (Andor) and Metamorph (Molecular Devices) as the image acquisition software. 16-bit Z-stacks were acquired in 0.2 μm steps (21 slices per stack) every 30 s. Maximum intensity projections were used for analysis. Wounds were made using a pulsed Micropoint N₂ laser (Andor) tuned to 365 nm. The laser produced 120 μJ pulses at the source with a duration of 2–6 ns. To wound embryos, ten laser pulses were delivered at each of seven spots along a 14 μm line.

To quantify wound closure dynamics, wounds were delineated using the Livewire algorithm, a semi-automated optimal path search method for image segmentation, in which the user traces the wound margin with the mouse, and the algorithm automatically identifies the brightest pixels that follow the trajectory of the mouse. We used the Livewire implementation in SIESTA, an image analysis platform that we develop[79,80]. The wound closure rate was calculated by subtracting the wound area 30 min after wounding from the maximum wound area for each wound, divided by the elapsed time.

We compared sample variances using the F-test for statistical analysis. To compare mean sample values, we used Student's t test for populations with equal or unequal variances (depending on the outcome of the F-test), applying Holm's correction to account for the comparison between three groups. For time series, error bars indicate SEM. For box plots, error bars show SD, the box indicates the SEM, and gray lines denote the mean.

## Data availability
Crystal structure coordinates and structure factors of αNcat-ABD-H1 form A, αNcat-ABD-H1 form B and αEcat-ABD-WT are deposited in the Protein Data Bank under accession codes: 6DUW, 6DUY, and 6DV1, respectively. Backbone chemical shift assignments of αNcat-ABD-H1 are deposited in the Biological Magnetic Resonance Data Bank under accession code 27526. All reagents and experimental data are available from the authors upon request. A Reporting Summary for this Article is available as a Supplementary Information file. Source Data are provided for Figs. 1f, 2d, 2f, 3b, 3e, 3f, 4a, 4e, 5c, 6b, 6c, 6f, and Supplementary Figs. 2b, 2c, 4b–d, 7a, 7c, 8a, 10a–b, 14a–b, 15a–c as a Source Data file.

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

## Acknowledgements

Mammalian expression vector pCA-αEcat-GFP was a gift from M. Takeichi. R2/7 cells were kindly provided by F. van Roy. We thank the CMCF beamline at the Canadian Light Source for assisting data collection. We thank G. Seabrook for help with NMR. We are grateful to K. Kakiguchi and H. Endoh for their help with electron microscopy. We thank the CSB Imaging Facility (U. of Toronto) and the Center for Advanced Microscopy (Northwestern U.) for support. We thank D. Kirchenbuechler for help with PIV analysis. We thank T. Hakoshima and T.Q.P. Uyeda for technical advice. Funding: Foundation Theme grant from the Canadian Institutes for Health Research (CIHR) (to M.I.), the Princess Margaret Cancer Foundation (to M.I.), the Canadian Foundation for Innovation (to M.I., U.T. and R.F.-G.), a project grant from the CIHR (to U.T.), a grant from the Canada First Research Excellence Fund (to U.T. and R.F.-G.), National Institutes of Health (NIH) grants GM076561 (to C.J.G.), P01HL071643 (to C.J.G.), HL134800 (to C.J.G.) and GM129312 (to C.J.G., N.I., M.I., U.T. and D.E.L.), National Science Foundation (NSF) grant CHE 12-3755 (to D.E.L.), a grant supported by CREST from JST, Japan (to S.Y.), and NSF Graduate Research Program DGE-1324585 (to M.N.W.). M.I is the Canadian Research Chair (CRC) in Cancer Structural Biology, U.T. is the CRC in Epithelial Polarity and Development, and R.F.-G. is the CRC in Quantitative Cell Biology and Morphogenesis.

## Author contributions

N.I. and M.I. conceived the study. N.I performed X-ray crystallography, actin cosedimentation, SEC-MALS, CD, confocal imaging, 3D cell culture, BLI and CS-Rosetta-CM experiments. R.S. and U.T. performed *Drosophila* overexpression and rescue experiments. M.N.W., A.S.F., A.Y., and C.J.G. performed scratch wound assays and epithelial sheet disruption studies. S.K.B and D.E.L. performed MD simulations. T.N., N.I., and M. I. performed NMR experiments. H.H. and S.Y. performed cell junction remodeling, electron microscopy and TER experiments. A.B.K. and R.F.-G. performed *Drosophila* wound healing assays. N.I., U.T., and M.I. wrote the paper with input from all authors.

## Additional information

**Competing interests:** The authors declare no competing interests.

