## [Peer Review File · Nature Communications]

Reviewers' comments:

Reviewer #1 (Remarks to the Author):

Force-dependent allostery of the α -catenin actin-binding domain controls adherens junction dynamics and functions

Ishiyama et al

Ishiyama and coworkers report here the mechanistic basis for a novel catchbond in the alpha-catenin actin binding domain (ABD). The existence of the catchbond was previously described by the authors, and this paper seeks to determine the mechanistic basis of the catch bond. They propose that the alpha1-helix (H1) folds on top of the structural core of the ABD, and inserts M672 into a cavity in the ABD. The authors argue that the cavity is allosterically coupled to V795, which can flip out of the cavity to make additional contacts with actin, stabilizing that interaction. Further, when the alpha-helix 1 is destabilized, a beta-turn from a second copy of the ABD can insert L806 into the cavity. Thus, when alpha catenin is under tension between membrane anchored components and actin filaments, the domain undergoes a conformational change which both enhances actin binding, and promotes dimer formation to lock in this conformational change.

Overall, the authors bring an interdisciplinary battery of methods to the question, including high resolution X-ray, bioinformatics, molecular dynamics, hetero-nuclear NMR with complete assignments, binding assays, mutagenesis, cellular and even organismal experiments, all of which support their overall thesis. The breadth of methods applied to this problem is impressive. Unfortunately, in many cases, experiments are not described in detail and the data failed to convince me on two key points. Owing to the breadth of the work and the interesting model proposed for such a well studied and important system, I think the paper could meet the standards for publication following a major revision. I worry that such a revised manuscript will be constrained by the length available in the Nat Comm format.

The revision needs to address two major points, and several smaller points.

Major issues:

Major concern 1)

Page 7) "These results further support that the unfolded α 1-helix propagates α cat-ABD to weakly dimerize through the β H-dependent interface."

Page 8) "Collectively, these results support the view that α -catenin facilitates actin bundling through ABD homodimerization."

The data establishing that the H1 helix disruption promotes ABD dimer formation was unconvincing. This is presented as one of the central points to their mechanism, and thus is important to establish clearly. They do have some limited data in support of this idea:

--The model stems from an observed crystallographic dimer, which is a type of observation that demands in solution corroboration.

-- Crosslinking data will not convince me on its own, it is prone to false positives and false negatives, which I have reason to suspect is the case here. A lack of dominant higher order species would help (but not prove) their case. They call attention to this idea in the legend for Fig S2e, "Note the predominant formation of ABD dimers without having larger oligomers resulting from non-specific cross-linking reactions." I disagree, I DO see some higher order species on the gel. Moreover, the abundance of crosslinked dimers hardly changes across their mutants. Their model predicts substantial changes increases in dimerization for the H1 mutant, and substantial decreases for the Δ BH mutant. The latter removes much of the

dimer interface, which the authors described as 'the BH dependent interface' after all. Instead they see changes of ~+10% for these two and ~-30% for the combined mutants, which I would argue is inconsistent with their model.

-- They have some limited gel filtration coupled with light scattering analysis (Supplementary Figure S1j). This is EXACTLY the sort of definitive experiment that is needed here, but the results indicate minimal change in particle size between the wild type ABD and the H1 ABD mutant. This directly speaks against their assertion that the H1 mutant supports dimer formation. H1 does elute slightly earlier, as might be expected for the less compact structure that results from disrupting helix α 1, but the masses do not change, implying no difference in oligomerization. Moreover, the molecular weight estimate across the size exclusion peaks begins between the monomer and dimer mass, and then decays to around 20 kDa, which is below the monomer mass. I will note that this is one of those points where almost no effort to explain the data was made, so I had to estimate the monomer mass (as ~27 kDa) myself. Such light scattering analyses can be surprisingly accurate, and dropping below the monomer mass calls into question the whole experiment.

-- They also see differences in line broadening/chemical shift between 50 μ M and >1 mM NMR samples. This indicates that if this dimer is real, the affinity is extremely weak. If the Kd was ~5 μ M (fairly modest in most contexts) by 50 μ M the effect should be nearly saturated. A weak affinity is actually plausible given their model in the context of AJs, where there may be substantial positive cooperativity in assembling the dimers on bundles of actin, but this isn't directly addressed in the text.

Overall, I think the authors need to definitively demonstrate a dimer using a well-established hydrodynamic method of their choice, and that H1 favors formation of the dimer. If they conclude that this is difficult or impossible due to very low ABD-ABD self-affinity (>100 μ M), then they need to address this in how they present the data and their model. As presented it feels like they are arguing that the H1 dimer is fairly stable, which begs demonstration.

Major concern 2)

Page 12) "Our site-directed mutagenesis and actin cosedimentation assays with the α Ecat-ABD variants support critical roles of L785, I792 and V796 in F-actin-binding: Ala substitutions of these hydrophobic residues, individually or together as 3A, led to 40–90% reduction in the amount of ABD cosedimenting with F-actin compared to α Ecat-ABD-WT (Fig. 5c), whereas mutating K795, preceding V796, resulted in less reduction (~30%) (Fig. 5c). These results confirm that V796/795 and other hydrophobic residues on the α 5-helix surface constitute an important binding surface for F-actin interaction."

This data is presented in a fashion that makes me uncomfortable, it feels like a case of false equivalence. V796 is central to most of the conversation up to this point, but I792A and V796A have ~40% and ~50% reduction, respectively. Significant, but not huge. The K795A mutant, which is presented as a sort of negative control was around 30% reduction in binding. This is not much of a difference, and the qualitative distinction between these feels arbitrary. The 90% reduction was from L785A, which was also in the 3A mutant and probably was the origin of the difference there, not the other two mutants. Basically, the V796A mutation only marginally disrupted actin binding. These sorts of spin down assays are reasonable where the effects are large and a qualitative statement is sufficient, but when effects are modest, one really wants a more quantitative approach. Moreover, presenting this as confirming the importance of V796 feels disingenuous. The reasonable conclusion is that L785 is important, and V796 may or may not be.

Further, this is only half of the experiment. The model is that V796 is largely important for

actin binding when BH is engaged, which it largely should not be in the wild-type background. Wouldn't a better experiment be to compare these mutants again in the H1 context for bundling and binding assays? A true negative control would also help; pick a surface exposed L or V that is remote from your proposed binding site. Does that alter binding?

Substantive but less major points:

A) I cannot assess the proposed M672 vs L806 mimicry without any information on how well the model fits the electron density. I actually like the concept, but I need some information on data/model quality here. A supplemental figure with model fit into density at various locations throughout the structure would suffice. Or sharing the structure factors, but I understand why the authors may be hesitant on that point.

B) NMR intensity data, please include normalized NH overlays comparing the high and low concentration spectra for at least a few residues that shifted for the analysis in Fig 2c and Fig S2d.

C) Page 6) "The structural integrity of acat-ABD was not affected by these mutations (Supplementary Fig. 1j,k)."

The data provided do not support this claim sufficiently, they only show that the protein variants have similar secondary structure. At least for the H1, $\Delta\alpha 1$, ΔBH and H1BH mutants, I would like to see that they still cooperatively unfold, with the folded state being stable at temperatures used for experiments in the paper. Substantial changes in stability are expected and not a concern, just establishing that the mutants are indeed stably folded under their experimental conditions is important for the claim made in the text. The fact that they were able to obtain a crystal structure of aNcat-ABD-H1 clearly indicates that this isn't likely a fatal flaw, but this is an important point to establish, and there is not sufficient data at this point.

D) Actin sedimentation assay methods: Please supply rotor, tube, sample volume, speed and temperature, not just "100,000 g" or "10,000 g".

E) The NMR assignments reflect a huge quantity of work, and are not documented. Typically at least a few strip plots are provided to provide a hint to the overall spectral quality. In addition, a statement as to the agreement of chemical shifts to those predicted from the crystal structure would be helpful, how well does this compare to community standards? Are other validations used (for example ARECA)? Will these be posted to BMRB?

F) Some of the supplementary figures were huge multipart figures that spanned multiple pages. Is there any reason not to break them into additional figures (I don't see a figure limit for the SI in the GTA)? I see the supplementary figures are connected to main text figures in the Cell Press style, not sure that is needed here, and it is not helping clarity.

G) SEC-MALS methods accompanying Fig S1j were not described.

H) The discussion needs a statement answering the question: Is the model in Supplementary Figure 7 consistent or inconsistent with the saturation transfer data in Figure 2f?

I) Is it possible to come up with a way of using the same numbering for a-Ecat and a-Ncat? The constant jumping back and forth by one amino acid is distracting.

J) From the preliminary structure validation reports the structures seem reasonable, but I have two concerns. Chain B, residues 704-712 may be worth rebuilding for the structure

model validated in report D_9100016477. V795 is flagged as a Ramachandran outlier in report D_9100016475. This structure is generally of lower quality, consistent with the lower resolution of the data. The key concern that I have is how this data is being used. V795 appears at least partly unreliable. Was this the structure used to infer the 'V795 flipped out' state?

Reviewer #2 (Remarks to the Author):

This work by Ishiyama et al describes a novel regulatory element (helix $\alpha 1$) in the F-actin binding domain (ABD) of α -catenin, which controls the affinity for F-actin and the α -catenin-mediated bundling of F-actin filaments. Molecular dynamics analysis supports the notion that $\alpha 1$ is sensitive to mechanical stretching. A mutant that destabilizes $\alpha 1$ increases the affinity for F-actin by releasing a neighbouring Val residue into a solvent-exposed F-actin-interacting conformation. Disruption of $\alpha 1$ also exposes a dimerization site in the ABD as revealed by two crystal structures of α Ncat-ABD. Using in vivo models of epithelial cell cultures and *Drosophila* embryos, it is shown that $\alpha 1$ -mediated regulation is required for proper dynamic remodelling of adherens junctions. This is an elegant, multidisciplinary, and sound work that presents a novel and compelling mechanism for the force-mediated regulation of α -catenin and its role in the maturation of adherence junctions.

I would like to address the following issues:

1) Page 7. Could you please give a bit more information about the α Ncat-ABD-H1 structures? How many different monomers are in the asymmetric unit of each crystal? How similar are the structures in both crystal forms? The descriptions in the legend of supplementary figure 2 are a bit too vague.

2) I miss a description of the $\Delta\beta$ H mutation; both in the α Ecat-ABD- $\Delta\beta$ H construct (e.g. page 8) and the α Ecat- $\Delta\beta$ H construct (e.g. page 9). Same for the α Ecat- Δ ABD (page 9) (is this the 1-651 region?).

The amino acid limits of the constructs are defined scattered throughout the text. It would be very useful to the potential readers if the details of the constructs would be defined in a single place, for example in a table in supplementary information.

3) Role of L785, I792, and V796 in binding to F-actin. (page 12) According to the text, the single mutations L785A, I792A, and V796A significantly reduced the interaction with F-actin, while the mutation K795A reduced the binding to F-actin to a lesser extent. Yet, Fig 5c shows that only L785A had a strong reduction of the binding, in contrast the interaction was similar for the I792A, K795A, and V796A. In addition, the triple mutant does not seem to reduce the binding further than the single L785A (similar amount of ABD in the pellet). BLI data in supplementary fig 6b also indicates that the α Ecat-ABD 3A mutant does not have lower binding to F-actin than the single mutant L785A. Could you please comment about this?

4) Regarding the BLI data shown in supplementary fig 6b, only a single α Ecat-ABD concentration for each mutant is shown. Please, indicate the concentrations used in the experiments in supplementary fig 6b and 6c (are they all the same?). Do the authors have complete titration experiments for each mutant that could help to understand the relative contribution of the mutations?

In supplementary fig 6c, please indicate what are the dBH2 and the H1dBH2 samples.

5) Also regarding the analysis of the BLI data (page 14 and fig 6 b-c). The fact that different models fit the data of the WT and H1 mutant is a major finding, and it is interpreted as differences in the activation states. To support this, it would very informative to show that a

single-site model fits worse the data of the WT, and similarly that a two-site is not justified for the H1 mutant. This could be shown as supplementary information.

6) Residues L785, I792, and V796 make direct contacts with one of the helical bundles of the M region in the structure of the full length α -catenin. Have the authors analyzed the effect of the triple-Ala mutation on the conformation of resting α -catenin?

7) Related to the previous issue: the effect of the triple-Ala mutant in the rescue of the α Cat^{-/-} Drosophila embryos is assumed to be linked to the disruption of the interaction with F-actin. Could a potential effect of these mutations on the conformational state of α -catenin (see above) also contribute to the observed behaviour in vivo?

8) Page 17. The α -catenin-mediated F-actin bundle model shown in supplementary figure 7 seems quite important to sustain the idea that the dimeric structure of the ABD, as seen in the crystals of the α Ncat-ABD-H1, is relevant and compatible with the bundling. Thus, it should be better shown in one of the main figures of the manuscript.

9) Related with the bundling model: are helices α 3 and α 4 in the model in contact or near the neighbouring actin filament to suggest a role for the secondary site detected by NMR?

10) The structure of α Ecat-ABD revealed that V796 is in a solvent-exposed F-actin-binding active conformation. Could the conformation of V796 in the α Ecat structure be related to the crystal packing? Showing a figure of the crystal packing, indicating the position of V796, would be very informative. This could be part of supplementary fig 5.

11) Regarding supplementary fig 5. Please, also show a map with reduced model bias, such as a (composite-) simulated annealing omit map, and/or a map of this region calculated with using the Br-SAD phases.

12) The structural data suggest that α Ncat-ABD is constitutively in an attenuated conformation, while α Ecat-ABD is in an activated state. The SDS-PAGE co-sed data in fig 1f suggest that indeed α Ecat-ABD sediments more efficiently than the α Ncat-ABD. Yet, in the bar-plot on the right of fig 1f the quantification suggests that they bind to a very similar extent. Could you please comment?

13) Crystal structures. General comments:

13.a) Please, submit all structures to the PDB, provide the accession numbers and the actual PDB-validation reports. The reports included are preliminary and not suitable for submission to the journals (as indicated in the first page of each report).

There are discrepancies between some values reported in the table 1 and in the validation reports (see below), which suggest that the reports and the table refer to different stages of refinement.

13.b) The structures are apparently correct (as judge by the preliminary validation reports). Nonetheless, there seem to be room to reduce the clashscore; in particular please look into the close contacts between pairs of non-hydrogen atoms.

13.c) It seems that the I/sigI in the outer resolution shell are still relatively high. I would like to encourage the authors to check whether data beyond the actual resolution limits still contains information, and in that case extend the resolution limit.

14) Specific comments about the crystal structures:

14.a) Structure of α Ncat-ABD-H1, form A (P6522)

There is a discrepancy between the number of water molecules reported in the validation report (90) and in the table 1 (89). Please, correct.

V795 is the only residue in a disallowed region of the Ramachandran plot. Given that V795 is biologically-relevant, please comment whether the backbone conformation is important to maintain the buried orientation of V795. Also, please, show an electron density map (e.g. SA-omit map) around the region of V795 to support this apparently unusual conformation (this could be in a supplementary figure).

14.b) Structure of α Ncat-ABD-H1, form B (P212121)

V795 of both molecules in the asymmetric unit occupies a disallowed region of the Ramachandran plot. As for the structure in the crystal form A, please show an electron density map (e.g. SA-omit map) around the region of V795 to support this apparently unusual conformation.

There is a discrepancy between the reported completeness in the validation report (97.6%) and in the table (99.9%). Please, check and correct if necessary.

The R factors in Table 1 (19.8/25.4 %) are different from those in the validation report (19.1/25.1 %). Again, it seems that the report does not correspond to the structure described in the manuscript.

14.c) Structure of α Ecat-ABD-WT.

There is a discrepancy between the reported completeness in the validation report (97.6%) and in the table (100%). Please, check and correct if necessary.

There is a discrepancy between the number of water molecules reported in the validation report (103) and in the table 1 (97). Please, correct.

15) Please, report CC1/2 values in Supplementary Table 1: .

16) Fig 2a. To which crystal form does the structure in the figure corresponds.

17) Fig 2a. Please, label residues L784, I791, and V795. Also, the use of a red sphere to mark the C termini is a bit confusing; the colour is too similar to that of the aforementioned residues.

18) Fig 2d. What do the lines in this plot represent? Are they any type of fitting? Also, what was the incubation time with dimethyl suberrimidate?

19) Fig 2f. Please, indicate the secondary structure elements in the plot on the left side (similarly as in Supplementary Fig. 6). That should help the non-expert reader to relate the plot with the molecular representations. Same for supplementary fig 2d.

20) Page 28. The circular dichroism methods describe thermal melting experiments, but, as far as I can tell, no thermal denaturation experiments are shown (e.g. in supplementary fig 1)

21) Supplementary Fig 1j. Scale of the absorbance signal is missing; please, correct.

Reviewer #3 (Remarks to the Author):

In this manuscript, the authors from at least five major labs renowned in the field of cadherin

mediated adhesion, present an interesting multiscale and multimethod approach to try to solve the complex problem of mechanosensing and mechanotransduction at adherens junction. The question asked here is of major importance, since despite numerous previous papers on the subject, it remains elusive to have a comprehensive picture of what happens at the molecular and cellular level during cadherin-mediated contact formation and rearrangements. The authors focus on one of the more important and complex proteins of the system, α -catenin. α -Catenin is indeed identified as a key mechanosensor that forms force-dependent interactions with F-actin, thereby coupling the cadherin-catenin complex to the actin cytoskeleton.

Here the authors combined structural and simulation analysis as well as functional cellular and whole organism analysis (in human epithelial cells and *Drosophila* embryos respectively) centered on the F-actin binding domain of the protein and in particular to its N-terminal region that was suspected from previous observations by Troyanovsky lab as well by others to have a modulatory role on the dynamics of adhesions.

The authors provide evidence that the α 1-helix of the α -catenin actin-binding domain (α cat-ABD) can unfold under force which may regulate tension-dependent F-actin binding. FABD containing an α 1-helix-unfolding mutation (H1) shows enhanced binding to F-actin *in vitro*. They also identify close to this region a β H motif that acts as a novel α -catenin homodimerization domain that may only be active upon force unfolding of the α 1-helix. They provide evidence that, in contrast to the previously identified α -catenin homodimerization domain, this domain may allow α -catenin dimerization while it interacts with β -catenin. They show that this domain is involved in F-actin bundling. Furthermore, structural and simulation analyses suggest that α 1-helix allosterically controls the actin-binding residue V796 exposure. They generated epithelial monolayers and embryos expressing α -catenin bearing these mutations and show data that suggest that the generated AJ can resist mechanical disruption, but fail to support normal AJ regulation *in vivo*. At the end, although the molecular mechanisms by which α -catenin engages F-actin under tension as well as the complete structural organization of α -catenin under its different conformations remained elusive, the authors integrate their findings in a proposed model that goes far beyond the previous understanding. They propose that force-dependent allosteric regulation of α -catenin ABD promotes dynamic interactions with F-actin involved in actin bundling, cadherin clustering, and finally AJ remodeling.

Altogether, this is a consistent piece of work, of wide interest. However, I see also a large number of presentation as well as data flaws that preclude publication of this manuscript under its present form. Many of these points are strangely misleading for the comprehension of all the subtleties of this complex work, others are related to a lack of clear representation of the various constructs used and others to a lack of quantification and statistics.

More details on these comments:

- 1) First I would say that because of my background I am not in a position to evaluate the structural data, but I feel having enough expertise for the cell and molecular biology aspects.
- 2) There is in this study a mix of use of mammalian α E-catenin and α N-catenin, as well as of *Drosophila* α -catenin. It is difficult to follow where we are and what is used each time. If the use of mammalian and *Drosophila* constructs is comprehensive. It is not clear why the authors shifted from α E to α N and vice-versa along the paper. Authors should find a way to clearly label the constructs they use so the species and isoform is self-explanatory. In addition, they use a lot of constructs for bacteria, mammalian cells and *Drosophila* with various very close deletion junctions. It is not well explained for all the constructs (in particular I cannot find in the text or method the description

of the expression constructs for mammalian cells and for Drosophila. The best would be to produce a single figure with the 3 catenins as well as vinculin, and the various constructs with clear indications on boundaries on the model of Fig S1A (adding the aa numbers). **The authors should also indicate in the text each time they change of isoform or species and why (in particular for αE and αN).**

Personally, I did not find the rationale. For example why αN is used for simulation in suppl Fig 1H-K, while αE is studied in panels A-F (without naming it on the figure)?

- 3) Along this line, it is difficult for the reader to see at the first read that in Suppl figure 1, authors compare cells expressing FL α -catenin and cells expression $\alpha 1$ mutated molecule in a Δ MII-III background. What gives a $\alpha 1$ mutated molecule with intact Δ MII-III domains. It is puzzling that this is not even discussed.**
- 4) I do not understand the message of the first sentence of result section (page 4): by definition a catch bond increases its stability with force....
- 5) For EM in the same figure, there are single images and no quantifications. How are we sure that these panels are representative of each condition. We need to see more example and/or quantifications. In addition there are holes in the prep (in D), where are taken the zoom, could you give in supplemental larger views with a few cells to testify of the preservation....**
- 6) Along this criticisms, there is no statistical treatment in Fig 1F, 2D, 4E and 5C and in Supl figure 1E, 2F, 4A. This is not acceptable today.**
- 7) Less important, but nice for the reader, I understand that the data have been collected in the different labs, but this does not preclude to presents histograms within a unified mode and color code, or at least to try to.
- 8) There a general lack of analysis of the mammalian cell expressing the constructs. What is the relation between clusters in Supl Fig 1C and actin roods in suppl Fig 1F, If there is E-cad in these clusters, they cannot be roods. Are there dues to missfolding of αE molecules? Strangely the structural analysis is then done mostly with αN !
- 9) Page 9, Accumulation “of α -catenin in protrusions” is not clear for me or not well illustrated (Suppl Fig 3B), I do not understand the piling described for Suppl Fig 3E and its relation to cell rearrangement and resilience to mechanical stress. These are some epiphenomenon’s for my that do not bring more to the message in absence of a better analysis.
- 10) It is not clear for what purpose the authors analyze scratch assays. They could have been tested the junctions stability by FRAP or Photoconversion as did other previously. Well, they decided to perform scratch assays, but in this case they should have pushed the analysis to have meaningful informations: they say that cell velovity is change but this is not quantified, this is a shame since the authors did single cell tracking (Supl Figure 3G) without extracting velocities. This must be done. They speak also about changes in cell coordination, this is not quantified either. It is easily feasible by different ways: from trajectories calculating MSD, by PIV as are doing many others, or by calculating neighbor exchange. This needs to be done one way or the other.**
- 11) In the first paragraph of the result, a simulation of force dependent-unfolding is nicely presented, without however mentioning the applied force either in the text or in the figure**

legend. The authors come back on this at the end of the results and discussion to say “our observations indicate that.... Is central to force-induced association of α -catenin to F-actin.” ...and so one and they produce a nice model. However, going to the method, you realize that the simulation is done under 100 pN while the force-dependent regulations described by Yao and Buckley are all centered around 5 pN (force developed by a few myosin motor). 100 pN is closer to the force needed to dissociate cadherin trans-dimers (Baumgartner, AFM experiments). This is never discussed and does not fit with the model.

12) Figure 3C is not explanatory without more information about the position of the leading edge. Quantification is needed here too.

13) There are two Yonemura citations page 16 not well integrated.

14) Molecular weight should be reported on each blot.

15) Why expressing constructs in a wild type drosophila background, Figure 4

16) For me, most figure legend are lacking important informations.

Some important improvement need to be added: see in particular points 3,4,6,10, 11 and 12 for my major concerns

Reviewer #4 (Remarks to the Author):

The manuscript by Noboru Ishiyama and co-workers describes how conformational changes in the actin-binding domain (ABD) of alpha-catenin relate to the sensing of mechanochemical stress upon force-dependent interactions with filament actin. The authors propose that a coupled conformational change of valine 796 in the ABD is connected via an allosteric interaction with the unfolding mechanism of a distant helix (alpha-helix 1) of a-catenin.

The authors present a technically fine executed study using a broad range of biochemical and biophysical techniques, including X-ray structure determination and advance NMR spectroscopy analysis. Using such broad range of techniques they study the binding mode of the a-catenin ABD to F-actin and find the coupled allosteric mechanism.

I have only some minor comments before suggesting publication of this study.

Title: I am uncertain if the catchy term allostery in the title:

“Force dependent allostery of the a-catenin actin-binding domain controls adherens junction dynamics and functions”

explains sufficiently well the study. Maybe: “Force-dependent allosteric conformational changes in a-catenin control adherens junction dynamics and functions” or “Force-dependent coupled allosteric changes” But I am not very decided on this.

I am a bit puzzled by the term “cryptic binding site”. I have seen that the term is used occasionally, but would “hidden” or “concealed binding site” binding site also work? E.g., on page 13 “Our observations of the cryptic (attenuated) and exposed (activated) conformations of V796 ...”

Page 3, last paragraph: The authors could provide the molecular mass of human a-catenin, which is 100.1 kDa.

Page 4, first and second line: The authors could specify the domain architecture as “The central mechanosensitive modulatory (M) domain ...” or “The C-terminal actin-binding domain ...” as the corresponding bar diagram displaying the domain architecture is only shown in Supplementary Figure 1a.

Page 6, first paragraph: The authors use two times the expression “We observed” for the description of a conformational change in the molecular dynamics simulations. As these changes are not shown by experimental evidence (as e.g. NMR spectroscopy would provide) but by simulations, I would prefer a wording like: “The MD simulations showed” or “suggested” ...

Page 12, second paragraph: The contribution of the L785 residue to the actin-binding site is a bit underrepresented and not discussed. In Figure 5c, the L785A mutant showed indeed the biggest effect on the loss of actin-binding in the co-sedimentation assay.

Page 17, first paragraph: It is not clear that the previous (or current) model is meant in the first sentence of this page. Why not say: Although ..., the current model fails to account for the capacity of a-catenin to bundle F-actin at AJs. ... Hence, our proposed model differs from the previous monomer-dimer model ...

Figure 5b, right panel: The display of the cavity the V795 can move in the aE-ABD-WT structure is not obvious from the figure layout. The cavity rather looks as a surface representation of the V795 side chain from the aN-ABD-WT structure. Maybe the authors could display a surface representation of the aE-ABD-WT structure, that would indicate the cavity.

BTW: In the crystal packing of the protein, is this site occupied by any moiety of a symmetry-related molecule or are there any water molecules observed in this cavity? Please comment on this.

Also, please provide the PDB accession numbers in the figure legend. As the L785 residue shows such severe effect in the F-actin binding assay in panel C, it would also be good to include this side chain in panel b, right inlet, and not only in the supplementary figure.

Typos:

Page 16, middle: ..., and a corresponding Drosophila construct ...

Authors' responses (AR) to the reviewers' comments (RC)

Reviewer #1:

RC1-1) Ishiyama and coworkers report here the mechanistic basis for a novel catchbond in the alpha-catenin actin binding domain (ABD). The existence of the catchbond was previously described by the authors, and this paper seeks to determine the mechanistic basis of the catch bond. They propose that the alpha1-helix (H1) folds on top of the structural core of the ABD, and inserts M672 into a cavity in the ABD. The authors argue that the cavity is allosterically coupled to V795, which can flip out of the cavity to make additional contacts with actin, stabilizing that interaction. Further, when the alpha-helix 1 is destabilized, a beta-turn from a second copy of the ABD can insert L806 into the cavity. Thus, when alpha catenin is under tension between membrane anchored components and actin filaments, the domain undergoes a conformational change which both enhances actin binding, and promotes dimer formation to lock in this conformational change.

AR1-1) We thank the reviewer for highlighting the key structural findings presented in our manuscript regarding the α -catenin actin binding domain (ABD).

RC1-2) Overall, the authors bring an interdisciplinary battery of methods to the question, including high resolution X-ray, bioinformatics, molecular dynamics, hetero-nuclear NMR with complete assignments, binding assays, mutagenesis, cellular and even organismal experiments, all of which support their overall thesis. The breadth of methods applied to this problem is impressive. Unfortunately, in many cases, experiments are not described in detail and the data failed to convince me on two key points. Owing to the breadth of the work and the interesting model proposed for such a well studied and important system, I think the paper could meet the standards for publication following a major revision. I worry that such a revised manuscript will be constrained by the length available in the Nat Comm format.

AR1-2) Thank you for emphasizing the importance of our work in the cell-cell adhesion field, and recognizing our efforts to reveal the molecular basis of α -catenin mechanosensing function by employing an interdisciplinary approach. We have extensively revised the manuscript based on helpful comments from all the reviewers.

RC1-3) The revision needs to address two major points, and several smaller points.

AR1-3) We have made substantial changes in the revised manuscript to address all the major and minor points raised by the reviewer.

RC1-4) Major issues:

Major concern 1)

Page 7) "These results further support that the unfolded α 1-helix propagates α cat-ABD to weakly dimerize through the β H-dependent interface."

Page 8) "Collectively, these results support the view that α -catenin facilitates actin bundling through ABD homodimerization."

The data establishing that the H1 helix disruption promotes ABD dimer formation was unconvincing. This is presented as one of the central points to their mechanism, and thus is important to establish clearly. They do have some limited data in support of this idea:

AR1-4) α cat-ABD dimerization is one of the major focal points of our manuscript, yet it is difficult to study due to its relatively weak interaction affinity in solution. Hence, we have employed numerous biophysical and biochemical methodologies to provide experimental evidence to support the weak dimerization of ABD, including newly added size-exclusion chromatography-multiangle light scattering (SEC-MALS) data in the revised manuscript.

RC1-5) --The model stems from an observed crystallographic dimer, which is a type of observation that demands in solution corroboration.

AR1-5) We have provided several “in solution” experimental results to support our crystallographic observation of the α Ncat-ABD dimer.

RC1-6) -- Crosslinking data will not convince me on its own, it is prone to false positives and false negatives, which I have reason to suspect is the case here. A lack of dominant higher order species would help (but not prove) their case. They call attention to this idea in the legend for Fig S2e, “Note the predominant formation of ABD dimers without having larger oligomers resulting from non-specific cross-linking reactions.” I disagree, I DO see some higher order species on the gel. Moreover, the abundance of crosslinked dimers hardly changes across their mutants. Their model predicts substantial changes increases in dimerization for the H1 mutant, and substantial decreases for the Δ BH mutant. The latter removes much of the dimer interface, which the authors described as ‘the BH dependent interface’ after all. Instead they see changes of \sim +10% for these two and \sim -30% for the combined mutants, which I would argue is inconsistent with their model.

AR1-6) Based on our preliminary size-exclusion chromatography results, we realized that it would be a challenge to provide supporting evidence for this weak ABD dimerization occurring in solution. Chemical cross linking approach was one of several methods we employed to provide supporting evidence to show that the α cat-ABD is capable of weakly dimerizing in solution. While we successfully observed crosslinked ABD dimers, we agree that ABD mutations, such as H1 and Δ β H, did not show substantial changes in the amount of crosslinked dimers, likely owing to the occurring of non-specific crosslinking reactions. Therefore, we have removed the crosslinking data and collected the new in solution evidence of α cat-ABD dimerization by using SEC-MALS.

RC1-7) -- They have some limited gel filtration coupled with light scattering analysis (Supplementary Figure S1j). This is EXACTLY the sort of definitive experiment that is needed here, but the results indicate minimal change in particle size between the wild type ABD and the H1 ABD mutant. This directly speaks against their assertion that the H1 mutant supports dimer formation. H1 does elute slightly earlier, as might be expected for the less compact structure that results from disrupting helix α 1, but the masses do not change, implying no difference in oligomerization. Moreover, the molecular weight estimate across the size exclusion peaks begins between the monomer and dimer mass, and then decays to around 20 kDa, which is below the

monomer mass. I will note that this is one of those points where almost no effort to explain the data was made, so I had to estimate the monomer mass (as ~27 kDa) myself. Such light scattering analyses can be surprisingly accurate, and dropping below the monomer mass calls into question the whole experiment.

AR1-7) Based on the reviewer's helpful comments, we decided to further optimize our SEC-MALS setup to improve our analysis of α Ecat-ABD-WT and -H1 proteins. We realized that the analytical gel filtration column (~10-year-old Superdex-75 10/300 GL) used to provide the original data was not optimal for the SEC-MALS analysis. Hence we recollected our data using our new Superdex-200 Increase 10/300 GL with a better resolution power. As you can see from our new data (Sup.Fig.4c), we have obtained superior SEC-MALS data with the MALS-based estimation of the molecular mass that agrees well with the theoretical mass of our α Ecat-ABD construct (28,522 Da). Furthermore, we have performed additional SEC-MALS experiments using this new SEC column to provide experimental results that support: a) α Ecat-ABD proteins can weakly dimerize in solution at high protein concentration, and b) the H1 mutation contributed to increased propensity for ABD homodimerization (Fig. 2d). We hope that the addition of this new data provides a stronger case for our observation of α cat-ABD-H1 homodimerization both in crystal and in solution.

RC1-8) -- They also see differences in line broadening/chemical shift between 50 μ M and >1 mM NMR samples. This indicates that if this dimer is real, the affinity is extremely weak. If the K_d was ~5 μ M (fairly modest in most contexts) by 50 μ M the effect should be nearly saturated. A weak affinity is actually plausible given their model in the context of AJs, where there may be substantial positive cooperativity in assembling the dimers on bundles of actin, but this isn't directly addressed in the text.

AR1-8) Based on our NMR results and the new SEC-MALS data, we also believe that α cat-ABD has a very weak affinity for homodimerization in solution. We have directly addressed this by adding the following statement in the Discussion section (page 17) of the main text: "Although α cat-ABD appears to dimerize with a very weak affinity in solution (Fig. 2c,d), AJ-localized α -catenin cooperatively binding to F-actin would likely increase the propensity of α cat-ABD to dimerize and promote F-actin bundling."

RC1-9) Overall, I think the authors need to definitively demonstrate a dimer using a well-established hydrodynamic method of their choice, and that H1 favors formation of the dimer. If they conclude that this is difficult or impossible due to very low ABD-ABD self-affinity (>100 μ M), then they need to address this in how they present the data and their model. As presented it feels like they are arguing that the H1 dimer is fairly stable, which begs demonstration.

AR1-9) Please see our response AR1-7. We have also further emphasized the weak affinity of α cat-ABD dimerization in the Discussion as mentioned in the above response AR1-8.

RC1-10) Major concern 2)

Page 12) "Our site-directed mutagenesis and actin cosedimentation assays with the α Ecat-ABD variants support critical roles of L785, I792 and V796 in F-actin-binding: Ala substitutions of these hydrophobic residues, individually or together as 3A, led to 40~90% reduction in the

amount of ABD cosedimenting with F-actin compared to α Ecat-ABD-WT (Fig. 5c), whereas mutating K795, preceding V796, resulted in less reduction (~30%) (Fig. 5c). These results confirm that V796/795 and other hydrophobic residues on the α 5-helix surface constitute an important binding surface for F-actin interaction.”

This data is presented in a fashion that makes me uncomfortable, it feels like a case of false equivalence. V796 is central to most of the conversation up to this point, but I792A and V796A have ~40% and ~50% reduction, respectively. Significant, but not huge. The K795A mutant, which is presented as a sort of negative control was around 30% reduction in binding. This is not much of a difference, and the qualitative distinction between these feels arbitrary. The 90% reduction was from L785A, which was also in the 3A mutant and probably was the origin of the difference there, not the other two mutants. Basically, the V796A mutation only marginally disrupted actin binding. These sorts of spin down assays are reasonable where the effects are large and a qualitative statement is sufficient, but when effects are modest, one really wants a more quantitative approach. Moreover, presenting this as confirming the importance of V796 feels disingenuous. The reasonable conclusion is that L785 is important, and V796 may or may not be.

AR1-10) We agree with the reviewers (the same issue was raised by Reviewers #1, #2 and #4) that the important role of L785 in F-actin binding was underrepresented in the original manuscript. We also realized that the incorrect values were reported for the range of reduction in F-actin binding. We have made the following modifications in the revised manuscript to further emphasize the major contribution of L785, and supportive roles of 792 and V796 in F-actin binding on page 13: “...Our site-directed mutagenesis and actin cosedimentation assays with the α Ecat-ABD variants support critical roles of these hydrophobic residues in F-actin-binding: Ala substitutions of L785, I792 and V796, individually or together as 3A, led to a range of reduction (75%, 36%, 47% and 78%, respectively) in the amount of ABD cosedimenting with F-actin compared to α Ecat-ABD-WT (Fig. 5c). Whereas Ala substitution of V714, which is located on the α 3-helix surface, resulted in no reduction (Fig. 5c). Also, none of the above mutations appear to interfere with the F-actin bundling activity of α Ecat-ABD (Supplementary Fig. 12). The equally significant reduction observed with either the L785A mutation alone or 3A suggests that L785 plays a central role in establishing the critical hydrophobic interface between F-actin and α cat-ABD. The measurable reduction in F-actin-binding with I792A or V796A suggests that I792 and V796 are likely involved in further stabilizing this interface, and any changes to these residues could modulate the F-actin-binding activity of α cat-ABD.”

The F-actin cosedimentation method takes advantage of the large polymeric property of actin to easily perform “qualitative” protein-F-actin binding studies, but it remains “qualitative” as noted. Hence, we have developed a new quantitative F-actin-binding assay using the bio-layer interferometry method to perform the kinetic analysis of the α cat-ABD-F-actin interaction, and the results were included in the original manuscript. We have further expanded our BLI analyses and measured the binding constants of various ABD mutants (including a new negative control V714A), and these results are now provided in Table 1 and Supplementary Fig. 14.

RC1-11) Further, this is only half of the experiment. The model is that V796 is largely important for actin binding when BH is engaged, which it largely should not be in the wild-type background. Wouldn't a better experiment be to compare these mutants again in the H1 context

for bundling and binding assays? A true negative control would also help; pick a surface exposed L or V that is remote from your proposed binding site. Does that alter binding?

AR1-11) We have now tested Ala substitution of the key actin-binding site residues in the H1 background, confirming that these mutations significantly reduce F-actin binding by α Ecat-ABD-H1 (Fig. 5c). We chose V714, which is located on the α 3-helix surface, as an alternate negative control, and we now show that Ala substitution of this residue does not reduce F-actin binding of α Ecat-ABD (Fig. 5c).

Substantive but less major points:

RC1-12) A) I cannot assess the proposed M672 vs L806 mimicry without any information on how well the model fits the electron density. I actually like the concept, but I need some information on data/model quality here. A supplemental figure with model fit into density at various locations throughout the structure would suffice. Or sharing the structure factors, but I understand why the authors may be hesitant on that point.

AR1-12) As requested by the reviewer, we now have a new supplementary figure, Sup. Fig. 5, showing a good agreement between our atomic model and electron density for all three newly determined crystal structures, α Ncat-ABD-H1 (Form A), α Ncat-ABD-H1 (Form B) and α Ecat-ABD-WT.

RC1-13) B) NMR intensity data, please include normalized NH overlays comparing the high and low concentration spectra for at least a few residues that shifted for the analysis in Fig 2c and Fig S2d.

AR1-13) We have added a new supplementary figure, Sup. Fig. 7b, which shows an overlay of ^1H - ^{15}N HSQC spectra of α Ncat-ABD-H1 at four different concentrations, 0.05 mM, 0.20 mM, 0.8 mM and 1.6 mM, showing concentration dependent chemical shift perturbations of peaks corresponding to α N-catenin residues K796, A797 and V808.

RC1-14) C) Page 6) “The structural integrity of α cat-ABD was not affected by these mutations (Supplementary Fig. 1j,k).”

The data provided do not support this claim sufficiently, they only show that the protein variants have similar secondary structure. At least for the H1, $\Delta\alpha$ 1, Δ BH and H1BH mutants, I would like to see that they still cooperatively unfold, with the folded state being stable at temperatures used for experiments in the paper. Substantial changes in stability are expected and not a concern, just establishing that the mutants are indeed stably folded under their experimental conditions is important for the claim made in the text. The fact that they were able to obtain a crystal structure of α Ncat-ABD-H1 clearly indicates that this isn't likely a fatal flaw, but this is an important point to establish, and there is not sufficient data at this point.

AR1-14) To further substantiate our above mentioned claim, we have examined the thermal stability of the α cat-ABD variants suggested by the reviewer, as well as the single-residue mutant V796A (Sup. Fig. 4e). As the reviewer predicted, the H1 mutation did not alter the thermal stability and folding of the α Ecat-ABD, as both WT and H1 constructs have the same melting

temperature (T_m) of 71 °C. Two other mutations, $\Delta\alpha 1$ and V796A, lowered the T_m by 1 and 2 degrees, respectively. On the other hand, we were pleasantly surprised by the increased thermal stability we observed with two β -hairpin deletion ($\Delta\beta H$) mutations, $\Delta\beta H$ and H1 $\Delta\beta H$, with T_m values of 77 and 75 °C, respectively. These new results strongly support that α cat-ABD variants were stably-folded under the experimental conditions used in various *in vitro* assays in this study.

RC1-15) D) Actin sedimentation assay methods: Please supply rotor, tube, sample volume, speed and temperature, not just “100,000 g” or “10,000 g”.

AR1-15) We have added the requested information in the Actin Cosedimentation Assay section of the revised Methods as: “... Samples of F-actin and ABD were mixed (total volume = 50 μ L) in Ultra-Clear Centrifuge Tubes (Beckman Coulter) and incubated for 1 h at RT. F-actin with bound protein samples were cosedimented by centrifugation using a Beckman Coulter Airfuge with a chilled A-100/30 rotor at 28 psi (>100,000 x g) for 20 m at RT. To assess actin bundling, F-actin with bound protein samples were cosedimented by centrifugation using a benchtop microcentrifuge at 10,000 x g for 30 m at 4 °C.”

RC1-16) E) The NMR assignments reflect a huge quantity of work, and are not documented. Typically at least a few strip plots are provided to provide a hint to the overall spectral quality. In addition, a statement as to the agreement of chemical shifts to those predicted from the crystal structure would be helpful, how well does this compare to community standards? Are other validations used (for example ARECA)? Will these be posted to BMRB?

AR1-16) We have provided below an example of sequentially aligned CBCA(CO)NH and HNCACB strips that were used to perform the backbone resonance assignment of α Ncat-ABD-H1 (Fig. AR1-16). The complete assignment of α Ncat-ABD-H1 have been deposited to BMRB under accession number 27526, and now mentioned in the methods section under “Data availability”. The chemical shift assignments were validated by the AVS¹ software during the deposition.

Fig. AR1-16: An example of the backbone chemical shift assignments of $\alpha\text{Ncat-ABD-H1}$.

RC1-17) F) Some of the supplementary figures were huge multipart figures that spanned multiple pages. Is there any reason not to break them into additional figures (I don't see a figure limit for the SI in the GTA)? I see the supplementary figures are connected to main text figures in the Cell Press style, not sure that is needed here, and it is not helping clarity.

AR1-17) We have re-organized our supplementary figures as suggested by the reviewer to improve the manuscript clarity.

RC1-18) G) SEC-MALS methods accompanying Fig S1j were not described.

AR1-18) We would like to thank the reviewer for notifying us of this omission. The revised manuscript now contains: “**Size-Exclusion Chromatography-Multiangle Light Scattering (SEC-MALS):** Purified protein (5 mg/mL, 100 μL injection volume) was subjected to size-exclusion chromatography using a Superdex-200 Increase 10/300 GL column (GE Healthcare) equilibrated in MALS buffer (20 mM Tris-HCl pH 7.0, 100 mM NaCl) at a flow rate of 0.5 mL/min. Multi-angle light scattering (MALS) measurements were performed in-line with SEC by using a three-angle (45° , 90° , and 135°) miniDawn light-scattering instrument and an Optilab rEX differential refractometer (Wyatt Technologies). Molecular weight was calculated by using the ASTRA software (Wyatt Technologies).”

RC1-18) H) The discussion needs a statement answering the question: Is the model in Supplementary Figure 7 consistent or inconsistent with the saturation transfer data in Figure 2f?

AR1-18) Our α cat-ABD-dependent actin bundling model presented in Sup. Fig. 7 (now Fig. 8) is consistent with the saturation transfer data shown in Fig. 2f. We have added the following sentence in the Discussion (page 17) to state this: "...The ABD-dependent dimerization as demonstrated here allows actin filaments to be tightly bundled in an antiparallel fashion (Fig. 8) and places the α 3- α 4 surface of ABD in close proximity with F-actin, which is consistent with our NMR saturation transfer data (Fig. 2f)."

RC1-19) I) Is it possible to come up with a way of using the same numbering for α -Ecat and α -Ncat? The constant jumping back and forth by one amino acid is distracting.

AR1-19) Thank you for the suggestion. We now refer to α N-catenin residues by using the equivalent α E-catenin residue numbers accompanied by a subscripted "N" to further improve the readability of our manuscript.

RC1-20) J) From the preliminary structure validation reports the structures seem reasonable, but I have two concerns. Chain B, residues 704-712 may be worth rebuilding for the structure model validated in report D_9100016477. V795 is flagged as a Ramachandran outlier in report D_9100016475. This structure is generally of lower quality, consistent with the lower resolution of the data. The key concern that I have is how this data is being used. V795 appears at least partly unreliable. Was this the structure used to infer the 'V795 flipped out' state?

AR1-20) α Ecat-ABD-WT chain B residues 704-712 are located in a loop between α 2- and α 3-helices. These were flagged for poor fit into the electron density in the validation report. We have inspected this region and found that the poor fit resulted from relatively poorly defined electron density for this loop region, possibly due to inherent flexibility.

The unusual backbone conformation of V795 appears to be real and not from poorly fitting the model into electron density. We have provided Sup. Fig. 11 to demonstrate the fit of V795 in SA-omit map (as requested by Reviewer #2). This conformation represents the cryptic/hidden state. The 'flipped out' state is inferred from the exposed conformation of V796 in α Ecat-ABD-WT.

Reviewer #2 (Remarks to the Author):

This work by Ishiyama et al describes a novel regulatory element (helix α 1) in the F-actin binding domain (ABD) of α -catenin, which controls the affinity for F-actin and the α -catenin-mediated bundling of F-actin filaments. Molecular dynamics analysis supports the notion that α 1 is sensitive to mechanical stretching. A mutant that destabilizes α 1 increases the affinity for F-actin by releasing a neighbouring Val residue into a solvent-exposed F-actin-interacting conformation. Disruption of α 1 also exposes a dimerization site in the ABD as revealed by two crystal structures of α Ncat-ABD. Using in vivo models of epithelial cell cultures and Drosophila embryos, it is shown that α 1-mediated regulation is required for proper dynamic remodelling of adherens junctions. This is an elegant, multidisciplinary, and sound work that presents a novel and compelling mechanism for the force-mediated regulation of α -catenin and its role in the maturation of adherence junctions.

I would like to address the following issues:

RC2-1) Page 7. Could you please give a bit more information about the α Ncat-ABD-H1 structures? How many different monomers are in the asymmetric unit of each crystal? How similar are the structures in both crystal forms? The descriptions in the legend of supplementary figure 2 are a bit too vague.

AR2-1) We have supplied additional information regarding the α Ncat-ABD-H1 structures in the revised legend of Sup.Fig. 6: “a. Superposition of α Ncat-ABD-WT and α Ncat-ABD-H1 crystal structures. All crystallographically unique structures of α Ncat-ABD-H1 from two different crystal forms closely resemble the α Ncat-ABD-WT structure (PDB ID: 4K1O): the RMSD of C α atoms of 148 out of 186 residues is $<0.36 \text{ \AA}$, except for the α N-catenin residues 668-674 adopting an extended conformation instead of an α -helix due to the H1 mutation.
b. Overlay of α Ncat-ABD-H1 homodimer structures from crystal forms A and B. The asymmetrical unit of form A contains one molecule, thus the homodimer consists of two crystallographically related molecules, whereas in form B two α Ncat-ABD-H1 molecules found in the asymmetric unit constitute the homodimer.”

RC2-2) I miss a description of the $\Delta\beta$ H mutation; both in the α Ecat-ABD- $\Delta\beta$ H construct (e.g. page 8) and the α Ecat- $\Delta\beta$ H construct (e.g. page 9). Same for the α Ecat- Δ ABD (page 9) (is this the 1-651 region?).

The amino acid limits of the constructs are defined scattered throughout the text. It would be very useful to the potential readers if the details of the constructs would be defined in a single place, for example in a table in supplementary information.

AR2-2) We would like to thank the reviewer for this suggestion. We have now provided Sup. Table 1 summarizing the details of the constructs used in our study.

RC2-3) Role of L785, I792, and V796 in binding to F-actin. (page 12) According to the text, the single mutations L785A, I792A, and V796A significantly reduced the interaction with F-actin, while the mutation K795A reduced the binding to F-actin to a lesser extent. Yet, Fig 5c shows that only L785A had a strong reduction of the binding, in contrast the interaction was similar for the I792A, K795A, and V796A. In addition, the triple mutant does not seem to reduce the binding further than the single L785A (similar amount of ABD in the pellet). BLI data in supplementary fig 6b also indicates that the α Ecat-ABD 3A mutant does not have lower binding to F-actin than the single mutant L785A. Could you please comment about this?

AR2-3) Please see our response AR1-10.

RC2-4) Regarding the BLI data shown in supplementary fig 6b, only a single α Ecat-ABD concentration for each mutant is shown. Please, indicate the concentrations used in the experiments in supplementary fig 6b and 6c (are they all the same?). Do the authors have complete titration experiments for each mutant that could help to understand the relative contribution of the mutations?

In supplementary fig 6c, please indicate what are the dBH2 and the H1dBH2 samples.

AR2-4) BLI response curves shown in the original Sup. fig. 6b and 6c were all obtained by using the same α Ecat-ABD concentration of 4 μ M for all constructs. We have added this detail in the legend of Sup. Fig. 14 (originally Sup. Fig. 6). We have performed additional titration experiments (Sup. Fig. 14) for several mutants to better understand the effects of ABD mutations on F-actin binding. Kinetic rate constants (k_{on} and k_{off}) and the dissociation constant (K_D) determined from curve fitting analyses have been summarized in Table 1. In Sup. Fig. 14b, we have corrected the typos of dBH2 and H1dBH2 to $\Delta\beta$ H and H1 $\Delta\beta$ H.

RC2-5) Also regarding the analysis of the BLI data (page 14 and fig 6 b-c). The fact that different models fit the data of the WT and H1 mutant is a major finding, and it is interpreted as differences in the activation states. To support this, it would very informative to show that a single-site model fits worse the data of the WT, and similarly that a two-site is not justified for the H1 mutant. This could be shown as supplementary information.

AR2-5) We agree with the reviewer that two different binding models used to fit the BLI data of the WT and H1 mutant corresponds well with our interpretation of the α cat-ABD activation states. We have prepared a new supplementary figure (Sup. Fig. 13b) comparing the fit of 1:1 homogeneous and 2:1 heterogeneous ligand (HL) binding models with the WT and H1 BLI data. As you can see, both the fitting and residual (a plot of the difference between the response and fitted curves) views of the WT data suggest that a 2:1 HL binding model is a better binding model to fit the WT data than the 1:1 homogenous binding model (Residual view shows greater deviations between the response and fitted curves). In comparison, the shape of H1 response curves are noticeably different from the WT curves, and the H1 data fit well with the 1:1 binding model. The 2:1 HL binding model fits similarly well with the H1 data, but the calculated percentage of two kinetic interactions in the total binding revealed that the best fit was obtained when there was no contribution from the second set of kinetic interaction, hence confirming the good agreement between the H1 data and the 1:1 fitting model.

RC2-6) Residues L785, I792, and V796 make direct contacts with one of the helical bundles of the M region in the structure of the full length α -catenin. Have the authors analyzed the effect of the triple-Ala mutation on the conformation of resting α -catenin?

AR2-6) Direct contacts between the actin-binding site residues, L785, I792 and V796, and the M domain observed in the crystal structure of human α E-catenin (PDB ID: 4IGG) is a crystal artefact likely caused by a crystal dehydration procedure employed by Izard and colleagues to improve the crystal diffraction quality, contributing to the observation of the compact conformation and domain rearrangements². For this reason, the relative orientation of ABD domains of the two protomers in the human α E-catenin dimer differs by almost 180° and makes different intramolecular contacts with M domains. In our previous studies, we have presented both crystallographic and SAXS data to suggest that the ABD domain of α -catenin is freely mobile and does not adopt a fixed conformation (*i.e.*, by binding to N or M domain) in solution³. Since the effect of 3A mutation is likely localized to the ABD domain, we did not analyze the effect of these mutations on the resting conformation of α -catenin.

RC2-7) Related to the previous issue: the effect of the triple-Ala mutant in the rescue of the α Cat -/- *Drosophila* embryos is assumed to be linked to the disruption of the interaction with F-actin.

Could a potential effect of these mutations on the conformational state of α -catenin (see above) also contribute to the observed behaviour *in vivo*?

AR2-7) As we described in our response to the above point R2-6, we do not believe that the conformational state of the α Cat-3A mutant is affected in any way to contribute to the observed *in vivo* behaviour. Expression of α Cat variants in the epidermis of *Drosophila* embryos mutant for α -Cat (α Cat^{-/-}) (Fig. 4c) showed that all the α Cat constructs, including 3A, colocalized at AJs with β -catenin, confirming that these ABD mutations did not affect the N domain functionality of α -Catenin.

RC2-8) Page 17. The α -catenin-mediated F-actin bundle model shown in supplementary figure 7 seems quite important to sustain the idea that the dimeric structure of the ABD, as seen in the crystals of the α Ncat-ABD-H1, is relevant and compatible with the bundling. Thus, it should be better shown in one of the main figures of the manuscript.

AR2-8) Thank you for your suggestion. We now present this figure as Fig. 8.

RC2-9) Related with the bundling model: are helices α 3 and α 4 in the model in contact or near the neighbouring actin filament to suggest a role for the secondary site detected by NMR?

AR2-9) Our α cat-ABD-dependent actin bundling model presented in Sup. Fig. 7 (now Fig. 8) does place α 3- and α 4-helices in close proximity to the neighbouring actin filament, and this is consistent with the saturation transfer data (Fig. 2f) which suggested many residues from α 3- and α 4-helices came in close contact with F-actin. However, we do not believe that the α 3- α 4 surface acts as a *bona fide* actin-binding site considering that the homodimerization-defective α Ecat-ABD-H1 Δ β H variant was unable to bundle F-actin, suggesting that the α cat-ABD only has one primary F-actin-binding site on the α 5-helix-containing surface. We have added the following sentence in the Discussion (page 17) to state this: "...The ABD-dependent dimerization as demonstrated here allows actin filaments to be tightly bundled in an antiparallel fashion (Fig. 8) and places the α 3- α 4 surface of ABD in close proximity with F-actin, which is consistent with our NMR saturation transfer data (Fig. 2f)."

RC2-10) The structure of α Ecat-ABD revealed that V796 is in a solvent-exposed F-actin-binding active conformation. Could the conformation of V796 in the α Ecat structure be related to the crystal packing? Showing a figure of the crystal packing, indicating the position of V796, would be very informative. This could be part of supplementary fig 5.

AR2-10) We do not believe that the conformation of V796 in the α Ecat-ABD-WT crystal structure is related to the crystal packing because V796 is found in an exposed state in both crystallographically independent chains, chains A and B, of α Ecat-ABD-WT. As you can see in the figure below (Fig. AR2-10), the surrounding chemical environments for V796 in two chains surrounded by other symmetry-related molecules are different. Whereas V796 of chain B is clearly exposed to a large solvent channel of the crystal, V796 of chain A (V796_A) is near a crystal contact site formed by the α 5-helices of chains A and B. However, V796_A is not directly involved in crystal packing as it is surrounded by water molecules and does not appear to engage in any hydrophobic interaction. Since the electron density of V796_A and V796_B

clearly indicate that these residues are in the exposed state, we feel strongly that the conformational state of V796 in α Ecat-ABD-WT is not affected by the crystal packing. We have noted these observations in the following sentence added to the Sup. Fig. 10 legend: “**d,e.** The initial electron density map obtained from bromine-SAD phasing around V796 of α Ecat-ABD-WT (contoured at 1σ ; mesh). It shows the exposed conformational state of V796, which is not involved in crystal packing, in both chains of the asymmetric unit.”

Fig. AR2-10) Contact sites among symmetry related molecules of α Ecat-ABD-WT do not involve V796 (white sticks).

RC2-11) Regarding supplementary fig 5. Please, also show a map with reduced model bias, such as a (composite-) simulated annealing omit map, and/or a map of this region calculated with using the Br-SAD phases.

AR2-11) Please see our response AR2-10.

R2-12) The structural data suggest that α Ncat-ABD is constitutively in an attenuated conformation, while α Ecat-ABD is in an activated state. The SDS-PAGE co-sed data in fig 1f suggest that indeed α Ecat-ABD sediments more efficiently than the α Ncat-ABD. Yet, in the bar-plot on the right of fig 1f the quantification suggests that they bind to a very similar extent. Could you please comment?

AR2-12) While the crystal structures of α Ncat-ABD and α Ecat-ABD revealed two different conformational states of the actin-binding site residue V796, these structures are very similar to each other, as these ABD structures can be superposed with RMSD of 0.53 \AA over 156 residues (Fig. 5b). These ABDs also share nearly identical amino acid sequences with 87% identity. Thus, our interpretation of these crystallographic structures is that the V796 residue is in a dynamic equilibrium between an attenuated state and an activated state when α Ncat-ABD and α Ecat-ABD molecules are in solution. We have also performed the statistical analysis of our actin-pelleting data in Fig. 1f, and confirmed that the difference in the amount of α Ncat-ABD-WT and α Ecat-ABD-WT cosedimented with F-actin is negligible, despite the appearance of the SDS-PAGE gel image in the figure. Thus, we believe that our interpretation of the structural data is consistent

with the similar levels of α Ncat-ABD-WT and α Ecat-ABD-WT proteins cosedimented with F-actin in the experimental conditions we tested.

RC2-13) Crystal structures. General comments:

13.a) Please, submit all structures to the PDB, provide the accession numbers and the actual PDB-validation reports. The reports included are preliminary and not suitable for submission to the journals (as indicated in the first page of each report).

There are discrepancies between some values reported in the table 1 and in the validation reports (see below), which suggest that the reports and the table refer to different stages of refinement.

AR2-13) We have deposited all three crystal structure to the PDB, as noted in the methods section as: “**Data availability:** Crystal structure coordinates and structure factors of α Ncat-ABD-H1 form A, α Ncat-ABD-H1 form B and α Ecat-ABD-WT are deposited in the Protein Data Bank under accession codes: 6DUW, 6DUY and 6DV1, respectively. Backbone chemical shift assignments of α Ncat-ABD-H1 are deposited in the Biological Magnetic Resonance Data Bank under accession code 27526. All reagents and experimental data are available from the authors upon request.”. Official PDB validation reports have been submitted with the revised manuscript. We have corrected all the discrepancies between values presented in Sup. Table 2 and in the validation reports.

RC2-14) 13.b) The structures are apparently correct (as judge by the preliminary validation reports). Nonetheless, there seem to be room to reduce the clashscore; in particular please look into the close contacts between pairs of non-hydrogen atoms.

AR2-14) We have examined these close contacts sites and confirmed that the atoms involved in high clashscores are optimally modeled according to electron density.

RC2-15) 13.c) It seems that the $I/\sigma I$ in the outer resolution shell are still relatively high. I would like to encourage the authors to check whether data beyond the actual resolution limits still contains information, and in that case extend the resolution limit.

AR2-15) We have re-evaluated the data and confirmed that the current resolution limit is the highest resolution that these crystal structures can be determined with high confidence.

14) Specific comments about the crystal structures:

RC2-16) 14.a) Structure of α Ncat-ABD-H1, form A (P6522)

There is a discrepancy between the number of water molecules reported in the validation report (90) and in the table 1 (89). Please, correct.

AR2-16) We have confirmed that the value presented in the Sup. Table 2 (originally 1) is correct. The new validation report has the same number of water molecules.

RC2-17) V795 is the only residue in a disallowed region of the Ramachandran plot. Given that V795 is biologically-relevant, please comment whether the backbone conformation is important to maintain the buried orientation of V795. Also, please, show an electron density map (e.g. SA-

omit map) around the region of V795 to support this apparently unusual conformation (this could be in a supplementary figure).

AR2-17) As the reviewer noted, V795 is in a disallowed region of the Ramachandran plots for both crystal forms of α Ncat-ABD-H1. Hence, we believe that this unusual backbone conformation of V795 is not due to poor fitting of the model (Please see electron density of SA-omit map around V795 in Sup. Fig. 11a) and biologically relevant. This is also consistent with the fact that V795 is located within a conformationally dynamic region of the α cat-ABD.

RC2-18) 14.b) Structure of α Ncat-ABD-H1, form B (P212121)
V795 of both molecules in the asymmetric unit occupies a disallowed region of the Ramachandran plot. As for the structure in the crystal form A, please show an electron density map (e.g. SA-omit map) around the region of V795 to support this apparently unusual conformation.

AR2-18) Please see our response AR1-17. Sup. Fig. 11a now shows electron density around V795 from SA-omit map to support this unusual conformation.

RC2-19) There is a discrepancy between the reported completeness in the validation report (97.6%) and in the table (99.9%). Please, check and correct if necessary.
The R factors in Table 1 (19.8/25.4 %) are different from those in the validation report (19.1/25.1 %). Again, it seems that the report does not correspond to the structure described in the manuscript.

AR2-19) We have confirmed that the completeness values from the Scalepack log files are correctly presented in Sup. Table 1 for data collection. The completeness value presented in the validation report was automatically extracted from the deposited mmcif file from PHENIX. We have updated the R factors in Sup. Table 2 (originally 1).

RC2-20) 14.c) Structure of α Ecat-ABD-WT.
There is a discrepancy between the reported completeness in the validation report (97.6%) and in the table (100%). Please, check and correct if necessary.
There is a discrepancy between the number of water molecules reported in the validation report (103) and in the table 1 (97). Please, correct.

AR2-20) We have confirmed that the correct completeness values from the Scalepack log file are reported in Sup. Table 2 from data processing. We have the correct number of water molecules and this is reflected in the updated validation report.

RC2-21) 15) Please, report CC1/2 values in Supplementary Table 1:

AR2-21) We have added the CC1/2 values in Sup. Table 2.

RC2-22) 16) Fig 2a. To which crystal form does the structure in the figure corresponds.

AR2-22) The dimer structure of α Ncat-ABD corresponds to form A. This is now noted in the figure legend.

RC2-23) 17) Fig 2a. Please, label residues L784, I791, and V795. Also, the use of a red sphere to mark the C termini is a bit confusing; the colour is too similar to that of the aforementioned residues.

AR2-23) Three actin-binding site residues, L784, I791 and V795, are now shown in three different colours in Fig. 2a. This allowed us to keep red spheres representing the C termini.

RC2-24) 18) Fig 2d. What do the lines in this plot represent? Are they any type of fitting? Also, what was the incubation time with dimethyl suberrimidate?

AR2-24) The original Fig. 2d has been removed from the revised manuscript due to the proneness of this technique to produce false positives and false negatives, as suggested by Reviewer #1. The plot lines simply connected the data points. We performed chemical crosslinking with 90 min incubation time.

RC2-25) 19) Fig 2f. Please, indicate the secondary structure elements in the plot on the left side (similarly as in Supplementary Fig. 6). That should help the non-expert reader to relate the plot with the molecular representations. Same for supplementary fig 2d.

AR2-25) We have added the secondary structure information in Fig. 2f and Sup. Fig. 7a (originally Sup. Fig. 2d).

RC2-26) 20) Page 28. The circular dichroism methods describe thermal melting experiments, but, as far as I can tell, no thermal denaturation experiments are shown (e.g. in supplementary fig 1)

AR2-26) We have now included the results of thermal denaturation experiments of various α E-catenin constructs as supplementary figure (Sup. Fig. 4e) to show that the α cat-ABD proteins are well folded over a range of temperature (4-37 °C) used to perform various experiments for the paper.

RC2-27) 21) Supplementary Fig 1j. Scale of the absorbance signal is missing; please, correct.

AR2-27) We have corrected this problem in the revised figure, Sup. Fig. 4a.

Reviewer #3 (Remarks to the Author):

Some important improvement need to be added: see in particular points 3,4,6,10, 11 and 12 for my major concerns

AC-EC) Please see below for our responses to Reviewer #3's comments, including the above

highlighted points. We have made substantial improvements to our manuscript based on many helpful suggestions from the reviewer.

In this manuscript, the authors from at least five major labs renowned in the field of cadherin mediated adhesion, present an interesting multiscale and multimethod approach to try to solve the complex problem of mechanosensing and mechanotransduction at adherens junction. The question asked here is of major importance, since despite numerous previous papers on the subject, it remains elusive to have a comprehensive picture of what happens at the molecular and cellular level during cadherin-mediated contact formation and rearrangements. The authors focus on one of the more important and complex proteins of the system, α -catenin. α -Catenin is indeed identified as a key mechanosensor that forms force-dependent interactions with F-actin, thereby coupling the cadherin-catenin complex to the actin cytoskeleton.

Here the authors combined structural and simulation analysis as well as functional cellular and whole organism analysis (in human epithelial cells and *Drosophila* embryos respectively) centered on the F-actin binding domain of the protein and in particular to its N-terminal region that was suspected from previous observations by Troyanovsky lab as well by others to have a modulatory role on the dynamics of adhesions.

The authors provide evidence that the α 1-helix of the α -catenin actin-binding domain (α cat-ABD) can unfold under force which may regulate tension-dependent F-actin binding. FABD containing an α 1-helix-unfolding mutation (H1) shows enhanced binding to F-actin in vitro. They also identify close to this region a β H motif that acts as a novel α -catenin homodimerization domain that may only be active upon force unfolding of the α 1-helix. They provide evidence that, in contrast to the previously identified α -catenin homodimerization domain, this domain may allow α -catenin dimerization while it interacts with β -catenin. They show that this domain is involved in F-actin bundling. Furthermore, structural and simulation analyses suggest that α 1-helix allosterically controls the actin-binding residue V796 exposure. They generated epithelial monolayers and embryos expressing α -catenin bearing these mutations and show data that suggest that the generated AJ can resist mechanical disruption, but fail to support normal AJ regulation in vivo. At the end, although the molecular mechanisms by which α -catenin engages F-actin under tension as well as the complete structural organization of α -catenin under its different conformations remained elusive, the authors integrate their findings in a proposed model that goes far beyond the previous understanding. They propose that force-dependent allosteric regulation of α -catenin ABD promotes dynamic interactions with F-actin involved in actin bundling, cadherin clustering, and finally AJ remodeling.

Altogether, this is a consistent piece of work, of wide interest. However, I see also a large number of presentation as well as data flaws that preclude publication of this manuscript under its present form. Many of these points are strangely misleading for the comprehension of all the subtleties of this complex work, others are related to a lack of clear representation of the various constructs used and others to a lack of quantification and statistics.

More details on these comments:

RC3-1) First I would say that because of my background I am not in a position to evaluate the structural data, but I feel having enough expertise for the cell and molecular biology aspects.

AR3-1) Thank you for thoroughly reviewing the cell and molecular biology aspects of our manuscript.

RC3-2) There is in this study a mix of use of mammalian α E-catenin and α N-catenin, as well as of *Drosophila* α -catenin. It is difficult to follow where we are and what is used each time. If the use of mammalian and *Drosophila* constructs is comprehensive. It is not clear why the authors shifted from α E to α N and vice-versa along the paper. Authors should find a way to clearly label the constructs they use so the species and isoform is self-explanatory. In addition, they use a lot of constructs for bacteria, mammalian cells and *Drosophila* with various very close deletion junctions. It is not well explained for all the constructs (in particular I cannot find in the text or method the description of the expression constructs for mammalian cells and for *Drosophila*). The best would be to produce a single figure with the 3 catenins as well as vinculin, and the various constructs with clear indications on boundaries on the model of Fig S1A (adding the aa numbers). The authors should also indicate in the text each time they change of isoform or species and why (in particular for α E and α N). Personally, I did not find the rationale. For example why α N is used for simulation in suppl Fig 1H-K, while α E is studied in panels A-F (without naming it on the figure)?

AR3-2) In the present study, we have utilized two subtypes of mammalian α -catenin, α E- and α N-catenin, as well as *Drosophila* α -catenin to perform comprehensive studies on the structure-function relationship of α -catenin. In the field of protein crystallography, it is a common practice to screen several different homologous proteins in attempt to find one form that can yield a high-resolution crystal structure of target protein. For example, we were able to crystallize α Ecat-ABD-WT, but not the H1 mutant, whereas α Ncat-ABD-H1 readily crystallized in two different crystallization conditions. Thus, our selection of which protein structure to study is often limited by the stability, solubility and crystallizability of recombinant proteins. Structure determination of both α Ecat-ABD-WT and α Ncat-ABD-H1 was actually crucial for revealing the cryptic and exposed conformations of V796, which is one of the major findings of this study. The use of α Ncat-ABD-WT for SMD simulations in the revised Sup. Fig. 4a,b (originally Sup. Fig. 1h,i; 1j was SEC-MALS data, 1k was CD data) was based on the fact that this was the highest resolution structure of α cat-ABD available at that time, which is crucial for performing precise SMD simulations.

Despite all the benefits of studying different α -catenin proteins, we agree with the reviewer that the use of three different α -catenin forms in the study has made the manuscript difficult to follow, especially when two equivalent residues in α E- and α N-catenin are referred by two different residue numbers. To alleviate any confusion, all α N-catenin amino acid residues are now referred by the equivalent residue numbers of α E-catenin, the most well characterized mammalian α -catenin subtype, accompanied by a subscripted 'N' (e.g., V795 as V796_N) for clarity. *Drosophila* α -catenin is presented in the manuscript with a capital 'C' (α Cat).

- In effort to better describe all the α -catenin constructs used in our study, we have prepared a new supplementary table (Sup. Table 1) which lists all the constructs used for mammalian cells and for *Drosophila*. Since we did not produce any vinculin constructs in this study, we did not include vinculin in the table.

We have updated the methods section to include additional details regarding the generation of the expression constructs for mammalian cells (please see below AR3-3) and *Drosophila* α Cat: "Generation of transgenes: To generate the *Drosophila* UASp- α Cat constructs,

full-length α Cat (2751 nucleotides) was cloned into Gateway pENTRTM/D-TOPO entry vector (Invitrogen) digested with Not1 and Asc1, using 3-part Gibson assembly reaction(NEB). α -Cat cDNAs carrying various mutations were cloned using 2- or 3-part Gibson assembly reactions using UASp- α Cat in pENTRTM/D-TOPO as the backbone digested with following restriction enzymes: Not1 and Bsp1 to generate α Cat-H1 and α Cat- $\Delta\alpha$ 1; Bsp1 and Asc1 to generate α Cat- $\Delta\beta$, Not1 and Asc1 to generate α Cat-H1- $\Delta\beta$; Bsp1 and Asc1 to generate α Cat-3A. The Gateway® LR® Clonase Enzyme mix was used to clone all entry vector constructs into the pPWH (pUASP-Gateway Cassette with C-terminal 3x HA) vector containing an attB recombination site that was added using the NSI1 restriction site. Transgenic animals were produced by Best Gene Inc., by using flies carrying the attP2 recombination site on the left arm of the third chromosome. Amino acids of α -Cat proteins are: α CatFL (1-917), α Cat-H1 (683REAM>683GSGS), α Cat- $\Delta\alpha$ 1 (1-658/691-917), α Cat- $\Delta\beta$ (1- 811/824-917), α Cat-H1- $\Delta\beta$ (1-811/824-917+683REAM>683GSGS), α Cat-3A (L798A+I805A+V809A), α Cat- Δ ABD (1-708; Desai et al., 2013).”

RC3-3) Along this line, it is difficult for the reader to see at the first read that in Suppl figure 1, authors compare cells expressing FL α -catenin and cells expression α 1 mutated molecule in a Δ MII-III background. What gives a α 1 mutated molecule with intact Δ MII-III domains. It is puzzling that this is not even discussed.

AR3-3) The rationale for testing α 1-mutation in a Δ M_{II-III} background is to examine the force-dependent direct interaction between α E-catenin and F-actin without the influence of force-dependent indirect α -catenin-F-actin interaction through other F-actin-binding proteins, such as vinculin⁴. We decided to constitutively activate the cryptic M_I/vinculin-binding site (VBS) by deleting the autoinhibitory M_{II-III} regions as previously shown⁴. This approach was briefly described in the original figure legend of Sup. Fig. 1b as, “M_I: M_I region (residues 290-395) contains a cryptic vinculin-binding site (residues 305-352) that becomes constitutively active when M_{II} and M_{III} are absent.”

Our results in Sup. Fig. 1b show that the Δ M_{II-III} deletion indeed activated the M_I/VBS in α -catenin, and did not interfere with AJ formation. Cells expressing NM_IABD (devoid of M_{II-III}) or NM_I (devoid of M_{II-III} and ABD) formed AJs at a similar pace as cells expressing α catFL (2 days), whereas cells expressing NM_IABD* (devoid of M_{II-III} and α 1-helix) displayed markedly delayed AJ formation. Furthermore, the AJ formation by cells expressing NM_IABD* was preceded by the unusual formation of internalized cadherin-catenin-F-actin clusters after trypsinization. We believe that the constitutive activation of VBS, instead of deleting the entire M domain, helped to enhance the effects of altered ABD-F-actin association in cells expressing NM_IABD*. Hence, by examining the α 1-mutation in the Δ M_{II-III} background, we have shown that the presence or absence of residues 663-696 can alter the direct interaction between α -catenin and F-actin during cell-cell junction remodelling.

To adequately explain our experimental details and observations to readers, we have now included the above information in the Methods (Mammalian cell culture section) and Sup. Fig. 1 figure legend of the revised manuscript as shown below (Changes are shown in bold):

“**Methods: Mammalian cell culture:** The full-length or deletion variant α E-catenin cDNA was subcloned into pCA vector to express α E-catenin-FLAG. **Site-directed mutagenesis was performed using the Quikchange protocol (Stratagene) to produce α E-catenin deletion**

mutants, NM_IABD, NM_IABD* and NM_I, with the constitutively active M_I/vinculin-binding site (VBS; residues 305-352). R2/7 cells were...”

“Sup.Fig.1 figure legend: “a. Schematic diagrams of α E-catenin variants. α catFL: full-length α E-catenin (residues 1-906). N: N-terminal dimerization domain (residues 1-262). M_I: M_I region (residues 290-395) contains a cryptic vinculin-binding site (VBS; residues 305-352) that becomes constitutively active when **the autoinhibitory M_{II-III} regions are absent. α 1: α 1-helix in the ABD (residues 663-906). ABD*: a N-terminally truncated ABD (residues 697-906), which is missing the α 1-helix. All α -catenin constructs contain a C-terminal FLAG tag.**

b. R2/7 cells stably expressing α -catenin variants were trypsinized, replated at 30% confluency and imaged after 2d (and after 5d for NM_IABD*). Since full-length α -catenin can both directly and indirectly interact with F-actin in a force-dependent manner, we selectively minimized the force-dependency of indirect F-actin binding by deleting the autoinhibitory M_{II-III} regions of α -catenin to constitutively activate the M_I/VBS. Cells expressing various α -catenin mutants (NM_IABD, NM_IABD* and NM_I) were compared to examine the effects of ABD mutations on the direct α -catenin-F-actin interaction during cell-cell junction remodelling. Whereas cells expressing NM_IABD or NM_I formed AJs at a similar pace as cells expressing α catFL (2 d), cells expressing NM_IABD* showed cadherin-catenin complexes trapped intracellularly with F-actin (2d) and required additional time to establish tight junction network stained with ZO-1 antibody (5d).”

RC3-4) I do not understand the message of the first sentence of result section (page 4): by definition a catch bond increases its stability with force....

AR3-4) The intended message of the first sentence of results section was to clarify that the catch bond formed between α -catenin and F-actin requires the discreet interaction between the α -catenin-ABD and F-actin that is stabilized under tension, independent of other potential F-actin-binding proteins, such as vinculin, to bind α -catenin and indirectly connect the cadherin-catenin complex to F-actin. To clearly present this message, we have modified the sentence to “The interaction between α -catenin and F-actin was demonstrated to be a catch bond, an interaction that is stabilized by increased force.”

RC3-5) For EM in the same figure, there are single images and no quantifications. How are we sure that these panels are representative of each condition. We need to see more example and/or quantifications. In addition there are holes in the prep (in D), where are taken the zoom, could you give in supplemental larger views with a few cells to testify of the preservation....

AR3-5) We have added two additional images (Sup. Fig. 2a) of both samples, α catFL and NM_IABD*, as well as expanded the view of the original lower magnification images to demonstrate that the “square-wave”-shaped AJs are distinct features frequently observed in cells expressing NM_IABD*, and are unlikely to be an artifact resulting from deformation of samples. To the best of our knowledge, “square-wave”-shaped AJ has not been observed during the course of normal formation of AJs by epithelial cells⁵. Although there are white patches in both samples that could be perceived as “holes”, we respectfully disagree with the reviewer that these are physical holes in our samples as these lack distinct round edges or bunched up creases often observed due to the tear/shrinkage of resin which the sample is embedded in. Hence, we believe

that these white areas in the micrographs likely represent thin sections that lack electron-dense material and do not interfere with the preservation of samples to contribute to any differences in the shape of AJs we observed between cells expressing α catFL and NM_IABD*.

RC3-6) Along this criticisms, there is no statistical treatment in Fig 1F, 2D, 4E and 5C and in Supl figure 1E, 2F, 4A. This is not acceptable today.

AR3-6) We have performed statistical analyses for Fig. 1f, 2d and 5c, described the statistical analysis methods in the Methods section, and now included the analysis results in these figures. In Fig. 4e, the statistical analyses were previously performed and the results were described within the figure legend. These results are now shown within the figure. The original Sup. Fig. 2f and other crosslinking results were removed from the revised manuscript as these results were deemed unclear to support ABD dimerization in solution and have been replaced with SEC-MALS data (Fig. 2d and Sup. Fig. 7c) in the revised manuscript.

RC3-7) Less important, but nice for the reader, I understand that the data have been collected in the different labs, but this does not preclude to presents histograms within a unified mode and color code, or at least to try to.

AR3-7) We have changed the color code of Fig. 4e and Sup. Fig. 10a to be consistent with the color coding previously introduced for WT and H1 in Fig. 3.

RC3-8) There a general lack of analysis of the mammalian cell expressing the constructs. What is the relation between clusters in Supl Fig 1C and actin roods in supl Fig 1F, If there is E-cad in these clusters, they cannot be roods. Are there dues to missfolding of α E molecules? Strangely the structural analysis is then done mostly with α N!

AR3-8) We appreciate the reviewer for highlighting ambiguities in our original manuscript regarding our experiments with mammalian cells expressing various α -catenin constructs. The relationship between clusters and actin rods shown in Fig. 1a and Sup. Fig. 2c (originally Sup. Fig. 1c) is that we observed these actin-rich structures in R2/7 cells expressing α -catenin constructs containing the ABD* (ABD without residues 663-696), either as NMIABD* (NMI connected to the ABD*) or ABD* alone. While cells expressing α catFL, NMIABD or NMI readily formed adherens junctions (AJs) in two days after trypsinization and replating (Sup. Fig. 1b), the AJ formation by cells expressing NMIABD* was considerably delayed under the same condition (5 days). In addition, NMIABD*-expressing cells formed prominent intracellular clusters (Sup. Fig. 1c), which contain E-cadherin (Sup. Fig. 1c), α E-catenin (Sup. Fig. 1b & 1c) and F-actin (Sup. Fig. 1b), presumably existing together as cadherin-catenin-F-actin complexes and may have impeded the swift expression of cadherins on the cell surface.

In contrast, actin rods are formed by cells expressing ABD* (residues 697-906), but not by cells expressing ABD (residues 663-906), hence highlighting a regulatory role of residue 663-696 (the difference between ABD and ABD*) in the ABD-F-actin interaction. These ABDs lack the β -catenin- and vinculin-binding sites present in NMI regions (residues 1-402), hence actin rods are unlikely to be associated with the E-cadherin- β -catenin complex. The colocalization of both ABD and ABD* with F-actin suggest that these proteins are well folded (Sup. Fig. 4d,e) to interact with F-actin. The structural analysis of α cat-ABD-H1 was carried out with α Ncat-ABD-

H1 as we were unable to crystallize α Ecat-ABD-H1.

To help present our results more clearly, we have added the above-mentioned information to our supplementary Fig. 1 figure legend: “**Sup. Fig. 1 legend: a.** R2/7 cells transiently expressing ABD (**residues 663-906**) or ABD* (**residues 697-906**). α cat-ABD/ABD*-FLAG and actin were labeled with the anti-DDDDK antibody and phalloidin, respectively. Scale bar, 10 μ m. **c.** Cadherin-catenin clusters found in R2/7 cells expressing NM₁ABD* (**as shown in Sup. Fig. 1b**) distribute intracellularly **2 days after replating**. Before triton treatment, ECCD2 that recognizes the extracellular domain of E-cadherin was applied and it stained E-cadherin expressed only on cell surface (left). After triton treatment, clone36/E-cadherin that recognizes the cytoplasmic domain of E-cadherin stained clusters in the cytoplasm, which were not stained before triton treatment, showing the clusters are isolated in the cytoplasm (middle). NM₁ABD* colocalizes with the E-cadherin cluster (right).”

A brief description of our effort to crystallize α E-cat-ABD-H1 has been added in the revised Methods section: “**Methods, Crystallization and data collection:**... Similarly, crystals of the α Ecat-ABD-WT in Buffer-P (30 mg/mL) were grown at 277 K by vapor diffusion with the reservoir solution consisting of 0.2 M KBr, 2.2 M (NH₄)₂SO₄ and 3 % (w/v) D-Galactose. **Crystallization of α Ecat-ABD-H1 was unfruitful.** Crystals were briefly soaked in crystallization solution containing 25 % glycerol for data collection at 100 K. ...”

RC3-9) Page 9, Accumulation “of α -catenin in protrusions” is not clear for me or not well illustrated (Suppl Fig 3B), I do not understand the piling described for Suppl Fig 3E and its relation to cell rearrangement and resilience to mechanical stress. These are some epiphenomenon’s for my that do not bring more to the message in absence of a better analysis.

AR3-9) The population of α -catenin accumulated in actin-rich protrusions of cells expressing the $\Delta\beta$ H, H1 $\Delta\beta$ H or Δ ABD variant is now indicated by white arrows to clearly illustrate our observation. Regarding the cell piling data presented in the original Sup. Fig. 3e, we thought those interested in cell-cell adhesion phenotypes relevant to cancer might be interested in the observation that H1-mutant expressing cells fail to pile like aCat WT expressing cells. But we agree that we don’t yet understand this phenomenon, and therefore for manuscript clarity we have removed the data.

RC3-10) It is not clear for what purpose the authors analyze scratch assays. They could have been tested the junctions stability by FRAP or Photoconversion as did other previously. Well, they decided to perform scratch assays, but in this case they should have pushed the analysis to have meaningful informations: they say that cell velocity is change but this is not quantified, this is a shame since the authors did single cell tracking (Supl Figure 3G) without extracting velocities. This must be done. They speak also about changes in cell coordination, this is not quantified either. It is easily feasible by different ways: from trajectories calculating MSD, by PIV as are doing many others, or by calculating neighbor exchange. This needs to be done one way or the other.

AR3-10) We now provide more detailed analysis of individual cell behaviors during scratch wound repair using both manual (MTrackJ, FIJI) and PIV-based (PIV_jar, FIJI) methods. The manual method allowed us to track the migration of individual cells exclusively at the wound front (see tracks in Sup. Fig. 9b, bottom), where we observed that α Ecat mutants displayed

decreased persistence (i.e., longer path length over net displacement) of cell movements without altering individual cell speed, leading to an overall reduced epithelial sheet migration (Fig. 3f, Sup. Fig. 9b and Supplementary Movie 4, 15-hour). We also sought to independently validate WT and aEcat-H1 cell trajectories using PIV analysis (Sup. Fig. 9c), where we focused exclusively on aEcat-H1 because it is a novel mutant that enhances α -catenin/F-actin-binding (a first in the field). Using an interrogation window that was narrowed to $\sim 100\mu\text{m}^2$ area (1-4 cells), averaged from 3 WT and 4 aEcat-H1 mutant wounds (times 91 frames for each wound), we found that aEcat-H1 cells at the wound front showed vectors orthogonal to the direction of migration, whereas WT cells were more aligned. No clear differences in vector direction or size were observed in cells distal to the wound edge (Sup. Fig. 9c). Altogether, these methods demonstrate that either enhanced or reduced α Ecat coupling to F-actin is sufficient to slow epithelial sheet migration by thwarting/antagonizing the normally persistent flow of cells at the wound front.

RC3-11) In the first paragraph of the result, a simulation of force dependent-unfolding is nicely presented, without however mentioning the applied force either in the text or in the figure legend. The authors come back on this at the end of the results and discussion to say “our observations indicate that.... Is central to force-induced association of α -catenin to F-actin.” ...and so one and they produce a nice model. However, going to the method, you realize that the simulation is done under 100 pN while the force-dependent regulations described by Yao and Buckley are all centered around 5 pN (force developed by a few myosin motor). 100 pN is closer to the force needed to dissociate cadherin trans-dimers (Baumgartner, AFM experiments). This is never discussed and does not fit with the model.

AR3-11) Steered molecular dynamics simulations generally involve larger than experimental forces to induce molecular transformations on time scales accessible with simulation, as it is currently not feasible to conduct simulations on the micro to millisecond time scales of typical experiments. This is well known and has been addressed in several publications which document the ability of simulations to predict experimental outcomes, despite the time scale and force differences. There are currently hundreds of MD simulations that have similarly accounted for or predicted key amino acid interactions during forced structural transitions, despite the different time and force scales involved. These examples include simulations of the cadherin catch bond, of selectin catch bonds, talin unfolding, and the exposure of RGD sites on fibronectin domains under force. Also, our previous simulations of α -catenin unfolding⁶ predicted experimentally verified roles of salt bridges in the stability of α -catenin and accounted for the force-extension signatures reported for magnetic tweezers experiments of Yao et al.⁷, when applying a 100pN constant force as in this study. We respectfully disagree with the reviewer's assertion that this "does not fit the model". The model predicts that force alters the conformation of the protein to expose a site that enhances actin binding. This is what the simulations demonstrate, and the results support the interpretation of the experimental data.

We also respectfully point out that Yao et al. measured α -catenin unfurling, and not actin binding.

RC3-12) Figure 3C is not explanatory without more information about the position of the leading edge. Quantification is needed here too.

AR3-12) The position of the leading edge is now indicated by an asterisk within the regions where we provided additional close-up views as insets. For quantification, we have performed PIV analysis of the leading edge of cells observed in Supplementary Movie 4.

RC3-13) There are two Yonemura citations page 16 not well integrated.

AR3-13) We have properly integrated these citations in the Nat. Comm. format in the revised manuscript. Thank you.

RC3-14) Molecular weight should be reported on each blot.

AR3-14) Molecular weights are indicated on the western blots shown in Sup. Fig. 8a (originally 3a) and 10b (originally 4b). Our coomassie blue stained images of SDS-PAGE gels now show the MW information of actin (42 kDa) and α cat-ABD (WT: 28 kDa) in Figs. 1f, 2e and 5c, and Sup. Fig. 10a (originally 4a).

RC3-15) Why expressing constructs in a wild type drosophila background, Figure 4

AR3-15) It was our intension to examine the potential toxicity of these constructs, and to ensure that the localization of these constructs was not affected by the presence of endogenous protein, as we have previously observed with some α Cat constructs⁸.

RC3-16) For me, most figure legend are lacking important informations.

AR3-16) We have included additional information to further improve our figure legends and the clarity of the revised manuscript.

Reviewer #4 (Remarks to the Author):

The manuscript by Noboru Ishiyama and co-workers describes how conformational changes in the actin-binding domain (ABD) of alpha-catenin relate to the sensing of mechanochemical stress upon force-dependent interactions with filament actin. The authors propose that a coupled conformational change of valine 796 in the ABD is connected via an allosteric interaction with the unfolding mechanism of a distant helix (alpha-helix 1) of a-catenin.

The authors present a technically fine executed study using a broad range of biochemical and biophysical techniques, including X-ray structure determination and advance NMR spectroscopy analysis. Using such broad range of techniques they study the binding mode of the a-catenin ABD to F-actin and find the coupled allosteric mechanism.

I have only some minor comments before suggesting publication of this study.

RC4-1) Title: I am uncertain if the catchy term allostery in the title: "Force dependent allostery of the a-catenin actin-binding domain controls adherens junction dynamics and functions"

explains sufficiently well the study. Maybe: “Force-dependent allosteric conformational changes in α -catenin control adherens junction dynamics and functions” or “Force-dependent coupled allosteric changes ...” But I am not very decided on this.

AR4-1) We have modified the title to “Force dependent allostery and dimerization of the α -catenin actin-binding domain control adherens junction dynamics and functions” to reflect the crucial findings presented in this study by making a stronger emphasis on the structure and function of α cat-ABD homodimer which has been now further supported by our new SEC-MALS data of weak ABD dimerization (Fig. 2d and Sup.Fig. 7c).

RC4-2) I am a bit puzzled by the term “cryptic binding site”. I have seen that the term is used occasionally, but would “hidden” or “concealed binding site” binding site also work? E.g., on page 13 “Our observations of the cryptic (attenuated) and exposed (activated) conformations of V796 ...”

AR4-2) Yes, hidden, occluded or concealed binding site would work, however we chose the term cryptic binding site, which is commonly used in the field of mechanobiology, to reflect that its conformational state can change under certain conditions. For example, it was previously used by Pokutta et al.⁹ to describe a hidden afadin-binding site found in full-length α -catenin.

RC4-3) Page 3, last paragraph: The authors could provide the molecular mass of human α -catenin, which is 100.1 kDa.

AR4-3) Thank you for the suggestion. We have now indicated the molecular mass of full-length α -catenin as well as the sizes of the individual domains on pages 3-4: “The structure of α -catenin (100 kDa) consists of three distinct domains. The N-terminal (N) domain (30 kDa) facilitates β -catenin binding and homodimerization in a mutually exclusive manner^{22,23}. The central mechanosensitive modulatory (M) domain (40 kDa) contains a cryptic binding site for another F-actin-binding protein vinculin^{6,24-27}. The C-terminal actin-binding domain (ABD) (28 kDa), which is connected to the rest by a flexible P-linker region (2 kDa), directly binds to F-actin, and closely resembles the vinculin ABD (vin-ABD). ...”

RC4-4) Page 4, first and second line: The authors could specify the domain architecture as “The central mechanosensitive modulatory (M) domain ...” or “The C-terminal actin-binding domain ...” as the corresponding bar diagram displaying the domain architecture is only shown in Supplementary Figure 1a.

AR4-4) We have made these changes. Please see our response to the above inquiry R4-3.

RC4-5) Page 6, first paragraph: The authors use two times the expression “We observed” for the description of a conformational change in the molecular dynamics simulations. As these changes are not shown by experimental evidence (as e.g. NMR spectroscopy would provide) but by simulations, I would prefer a wording like: “The MD simulations showed” or “suggested” ...

AR4-5) We have incorporated your suggestions in the revised manuscript on Page 6: “The SMD simulations showed α 1-helix unfolding after a constant pulling force was applied on α Ncat-ABD

for 60 ns (Fig. 1d, Supplementary Fig. 4a,b (originally 1h,i) and Supplementary Movie 1). Interestingly, shortly before α 1-helix unfolded (at ~45 ns), the side chain of V795 turned over from a cryptic position to an exposed position (Fig. 1e and Supplementary Movie 2). α N-catenin residues V796_N and I792_N are equivalent to the vinculin actin-binding site residues, V1001 and I997¹⁰ (Fig. 1c). These results suggest that the conformational flexibility of α 1-helix and the dynamics of V796_N are mechanically coupled within the α Ncat-ABD.”

RC4-6) Page 12, second paragraph: The contribution of the L785 residue to the actin-binding site is a bit underrepresented and not discussed. In Figure 5c, the L785A mutant showed indeed the biggest effect on the loss of actin-binding in the co-sedimentation assay.

AR4-6) Please see our responses AR1-10 and AR1-11.

RC4-7) Page 17, first paragraph: It is not clear that the previous (or current) model is meant in the first sentence of this page. Why not say: Although ..., the current model fails to account for the capacity of a-catenin to bundle F-actin at AJs. ... Hence, our proposed model differs from the previous monomer-dimer model ...

AR4-7) We have incorporated these suggestions in the revised manuscript. Thank you.

RC4-8) Figure 5b, right panel: The display of the cavity the V795 can move in the α E-ABD-WT structure is not obvious from the figure layout. The cavity rather looks as a surface representation of the V795 side chain from the α N-ABD-WT structure. Maybe the authors could display a surface representation of the α E-ABD-WT structure, that would indicate the cavity. BTW: In the crystal packing of the protein, is this site occupied by any moiety of a symmetry-related molecule or are there any water molecules observed in this cavity? Please comment on this.

AR4-8) The cavity shown in Fig. 5b is indeed from the molecular surface representation of the α E-catenin, and not from the surface representation of α N-catenin. In effort to address the concerns raised by the reviewer, we have prepared a new supplementary figure (Sup. Fig. 11c) showing the entire molecular surface representation of α Ecat-ABD-WT. A cavity near V796 is present in both α Ecat-ABD-WT chains in the asymmetric unit: the cavity in chain A (panel i and Fig. 5b) is closed off and contains four water molecules, whereas the cavity in chain B (panel ii) does not contain electron density for solvents or symmetry-related molecule and is connected to the outer surface. Comparison of the surface representations of α Ecat-ABD-WT and α Ncat-ABD-WT suggests that the appearance of the cavity in α Ecat-ABD-WT involves a localized conformational change of the α 5- β 1 linkage containing V796. As the reviewer noted, the close resemblance of the cavity and a surface representation of V795 in the cryptic state strongly suggest that the residue in this position likely assumes dynamic conformational states contributing to dynamic F-actin binding.

RC4-9) Also, please provide the PDB accession numbers in the figure legend. As the L785 residue shows such severe effect in the F-actin binding assay in panel C, it would also be good to include this side chain in panel b, right inlet, and not only in the supplementary figure.

AR4-9) We have deposited all three crystal structure to the PDB, as noted in the methods section as: “**Data availability:** Crystal structure coordinates and structure factors of α Ncat-ABD-H1 form A, α Ncat-ABD-H1 form B and α Ecat-ABD-WT are deposited in the Protein Data Bank under accession codes: 6DUW, 6DUY and 6DV1, respectively.” We have now included L785 in the revised Fig. 5b.

RC4-10) Typos:

Page 16, middle: ..., and a corresponding *Drosophila* construct ...

AR4-10) We have corrected the mistake in the revised text. Thank you.

References

1. Moseley, H.N.B., Sahota, G. & Montelione, G.T. Assignment validation software suite for the evaluation and presentation of protein resonance assignment data. *J. Biomol. NMR* **28**, 341-355 (2004).
2. Rangarajan, E.S. & Izard, T. Dimer asymmetry defines α -catenin interactions. *Nat. Struct. Mol. Biol.* **20**, 188-193 (2013).
3. Ishiyama, N. *et al.* An autoinhibited structure of α -catenin and its implications for vinculin recruitment to adherens junctions. *J. Biol. Chem.* **288**, 15913-15925 (2013).
4. Yonemura, S., Wada, Y., Watanabe, T., Nagafuchi, A. & Shibata, M. α -Catenin as a tension transducer that induces adherens junction development. *Nat. Cell Biol.* **12**, 533-542 (2010).
5. Yonemura, S., Itoh, M., Nagafuchi, A. & Tsukita, S. Cell-to-cell adherens junction formation and actin filament organization: similarities and differences between non-polarized fibroblasts and polarized epithelial cells. *Journal of Cell Science* **108 (Pt 1)**, 127-142 (1995).
6. Barrick, S. *et al.* Salt bridges gate α -catenin activation at intercellular junctions. *Molecular Biology of the Cell* **29**, 111-122 (2018).
7. Yao, M. *et al.* Force-dependent conformational switch of α -catenin controls vinculin binding. *Nat. Comm.* **5**, 4525 (2014).
8. Desai, R. *et al.* Monomeric α -catenin links cadherin to the actin cytoskeleton. *Nat. Cell Biol.* **15**, 261-273 (2013).
9. Pokutta, S., Drees, F., Takai, Y., Nelson, W.J. & Weis, W.I. Biochemical and structural definition of the 1-afadin- and actin-binding sites of α -catenin. *J. Biol. Chem.* **277**, 18868-18874 (2002).
10. Thompson, P.M. *et al.* Identification of an actin binding surface on vinculin that mediates mechanical cell and focal adhesion properties. *Structure* **22**, 697-706 (2014).

Reviewers' comments:

Reviewer #1 (Remarks to the Author):

Criteria for publication:

1) Provides strong evidence for its conclusions.

Generally yes, but there is a major exception discussed below.

2) Novel (we do not consider meeting report abstracts and preprints on community servers to compromise novelty).

Yes

3) Of extreme importance to scientists in the specific field.

Yes

4) Ideally, interesting to researchers in other related disciplines.

Yes

Recommendation:

Invite the authors to revise their manuscript to address specific concerns before a final decision is reached

Detailed comments:

I had two major concerns with the first manuscript, and several minor concerns. The authors have expended significant effort to address all points.

FIRST POINT:

""""

RC1-4) Major issues:

Major concern 1)

Page 7) "These results further support that the unfolded α 1-helix propagates acat-ABD to weakly dimerize through the β H-dependent interface."

Page 8) "Collectively, these results support the view that α -catenin facilitates actin bundling through ABD homodimerization."

The data establishing that the H1 helix disruption promotes ABD dimer formation was unconvincing. This is presented as one of the central points to their mechanism, and thus is important to establish clearly. They do have some limited data in support of this idea:

AR1-4) acat-ABD dimerization is one of the major focal points of our manuscript, yet it is difficult to study due to its relatively weak interaction affinity in solution. Hence, we have employed numerous biophysical and biochemical methodologies to provide experimental evidence to support the weak dimerization of ABD, including newly added size-exclusion chromatography, multi-angle light-scattering (SEC-MALS) data in the revised manuscript.

""""

The authors present new SEC-MALS data to address this concern. In doing so, they find evidence of a low abundance population of dimers in the SEC chromatograms, as assessed by a mass estimate using SEC-MALS. By integrating the peak area and plotting relative to the presumed input concentration, they find a roughly linear relationship between input concentration and observed dimer (Fig 2d). Moreover, they find that the increase in dimer area grows faster for H1-ABD than for wt-ABD, although only by a small margin.

I still have a concern with the claim that the H1-ABD truncation supports dimer formation.

Unfortunately, this new data does not convince me that H1-ABD dimers are more stable than wt-ABD dimers. There are three reasons for this.

First, the effect is modest, even at the extreme concentrations used here. Given that their max concentration is ~ 1mM (30 mg/ml for ~28 kDa protein) and the NMR peaks are about 50% shifted at 0.8 mM, I expected more than 1% dimer by this concentration. The magnitude of such effects in SEC can be difficult to predict though, so this is a weak criticism. I worry this may be a contaminant of a fortuitous size as no gels are provided to show the dimer peak contains ABD.

Second, the appearance of a resolved dimer peak is unusual for weak monomer-dimer equilibria. More typically, weak monomer-dimer equilibria show up as a broadened but single dominant peak, which shifts to early elution volumes with increasing concentration. The exact peak shape and behavior will depend on the interaction model, concentrations used and rate constants, so the detailed shape is difficult to predict. Here, the authors observe resolved dimers instead.

Third, the concentration dependence of the dimer peak seems incorrect for a weak affinity monomer-dimer equilibrium. The authors report that the dimer peak area concentration scales roughly linearly with input concentration (Fig 2d).

If we assume that this area is reporting on the dimer concentration, which seems reasonable, this means the concentration of dimer scales roughly linearly with total concentration.

Does that make sense? Given the very low dimer concentration observed, it is safe to estimate the concentration of ABD monomers, [ABD], as the total ABD concentration. If the dimerization is governed by a monomer-dimer equilibrium, then we can estimate the solution dimer concentration as:

$$[\text{ABD dimer}] * K_{\text{dimer}} = [\text{ABD}]_{\text{total}}^2$$

Thus, this general behavior is inconsistent with the presented data. The concentration dependence appears linear, not parabolic. For example, the ratio of dimer areas for the 10 mg/ml samples and the 30 mg/ml samples is roughly 1:3, and not 1:9. I am not convinced this linear dependence is reporting on a dynamic monomer-dimer equilibrium.

Fourth, the dimer area is used directly to assert that H1 supports dimer formation. This makes sense if the loaded concentrations and volumes are exact, but this is rarely the case in reality. Given the low fraction of dimer, the monomer areas should give an idea of how well matched the total loading was for each sample. The monomer areas appear to differ on the order of the differences between the H1-ABD and wt-ABD dimer areas. Based on max absorbance level in the provided examples, the H1-ABD monomer peaks appear to be of greater absorbance than the wt-ABD monomer peaks, particularly at those concentrations flagged as being statistically increased in dimer content. In other words, there is a very real chance that the modest difference in dimer accumulation may be due to modest differences in stock concentrations, intrinsic absorbance, or some sort of sample application issue.

While I applaud the authors' substantial efforts here, the end result just does not convince me that the H1-ABD mutation stabilizes the dimer over the wt-ABD dimer, and I still worry that this claim is incorrect. I still see potential here, but am not satisfied with the current proof of their thesis.

.....

AR1-8) Based on our NMR results and the new SEC-MALS data, we also believe that acat-ABD has a very weak affinity for homodimerization in solution. We have directly addressed this by adding the following statement in the Discussion section (page 17) of the main text: "Although acat-ABD appears

to dimerize with a very weak affinity in solution (Fig. 2c,d), AJ-localized α -catenin cooperatively binding to F-actin would likely increase the propensity of α cat-ABD to dimerize and promote F-actin bundling.”

.....

This response is reasonable. I still disagree that they have shown a weak affinity homodimer definitively. That said, the conclusion that AJ F-actin binding could hugely affect apparent affinity is completely reasonable.

Another approach to supporting the dimerization idea is to disrupt the dimer interface and look for bundling effects (which is reasonably cast as the dimer sensitive assay). The Δ BH/H1 mutation disrupts bundling (Fig. 2e), but the Δ BH mutation does not, although slightly less Δ BH co-sediments with actin in the bundling assay. Both Δ BH and Δ BH/H1 disrupt actin binding (right side of Fig 2e). In the bundling assay, despite the disrupted direct binding of actin, the Δ BH material still bundles actin. Given the model, if Δ BH disrupted the dimer, as might be predicted, then it isn't clear how Δ BH can result in bundling.

This all raises a real concern on my part, is the dimer 'real'? While the NMR line broadening and observation of a limited quantity of dimer in SEC-MALS suggests a dimer may exist, it doesn't actually mean that a dimer is relevant for the rest of the biology and biochemistry discussed. How much of the story is lost if the H1 deletion just shifts an allosteric conformational equilibrium that affects actin binding? I think the newly provided BLI experiments actually speak to the conformational equilibrium nicely (Supplemental Fig 13b). Focusing on an allosteric equilibrium keeps the strongest parts of the story (in this reviewer's mind at least) and removes what is proving to be a persistently weak element. I think the paper would still warrant publication in Nat Comm minus the 'H1 disruption leads to dimerization' point, as the depth of information on the allosteric mechanism is greatly interesting on its own.

SECOND POINT:

.....

RC1-10) Major concern 2)

Page 12) “Our site-directed mutagenesis and actin cosedimentation assays with the α cat-ABD variants support critical roles of L785, I792 and V796 in F-actin-binding: Ala substitutions of these hydrophobic residues, individually or together as 3A, led to 40~90% reduction in the amount of ABD cosedimenting with F-actin compared to α cat-ABD-WT (Fig. 5c), whereas mutating K795, preceding V796, resulted in less reduction (~30%) (Fig. 5c). ...

AR1-10) ...

.....

The mutagenesis of hydrophobic residue data is much more compelling. The V714 mutation is a much better control. The added BLI data with these mutations is also convincing and alleviates concerns about quantitation with the spin down assays.

Minor response points.

1) New p. 13: “Also, none of the above mutations appear to interfere with the F-actin bundling activity of α cat- ABD (Supplementary Fig. 12).”

Examining Supplementary Fig. 12, it is not at all clear how this statement connects with the data. L785A and 3A have clear effects in the bundling assays, V796A probably has an effect, though more modest. In the H1 context, all of them have a significant effect. Given that the point here was about direct actin binding, I am not sure this discrepancy affects the argument. Please correct.

2) The H1 context is barely mentioned in the results section at all. Seems worth a passing comment at

least, particularly given the more pronounced effects shown. Is it puzzling that the bundling is less effective in the H1-hydrophobic mutant experiments?

3) I am not sure the actin cosedimentation experiments are sufficiently described in the methods. What are the concentrations of actin and ABD in the spin down assays?

4) My gut instinct is that the R_{max} based $K1/K2$ (%) numbers may be telling you something about the allosteric conformational bias. The H1 and ΔBH mutations seem particularly provocative (my comment on ΔBH is inferred from the time course as it does not appear in the table).

Responses to RC1-12 through RC1-20:
Satisfactory changes all around.

One minor point there, the 'N' subscript nomenclature helps, but it used inconsistently in Supplementary Fig 5, where superscripts and 'E' are used.

Reviewer #2 (Remarks to the Author):

The authors have satisfactorily addressed all my comments and concerns. The revised manuscript contains a more clear presentation of some data, information, and figures, which should help the potential readers. This was very important given the complexity of this multidisciplinary work. The manuscript has also been improved by the inclusion of new data (e.g. supplemental figures 4e, 7c, 13, and 14); a more rigorous analysis of some data (e.g. statistical analysis) also helps to better sustain the conclusions of the work.

Several of my concerns to the original manuscript were related to the three crystal structures. The atomic structures and the diffraction data have now been deposited in the PDB, proper validation reports have been submitted, and the crystallographic data in the manuscript (Supplementary Table 2) correspond to the deposited structures.

In summary, the results are more clearly and solid presented in the revised manuscript and, in my opinion, they sustain the conclusions on the novel mechanism underlying the mechanical response of alpha-catenin in AJs.

Reviewer #3 (Remarks to the Author):

I am satisfied with the response of the authors to my comments. I feel that their response to all the comments raised by the other reviewers strengthen very much this piece of work.

Reviewer #4 (Remarks to the Author):

The authors have performed a very thoroughly revision of the manuscript. I am pleased with all changes that have been made upon the revision. The broad range and depth of biochemical and biophysical techniques applied in this study is truly exceptional.

Authors' responses (AR) to Reviewer #1's comments (RC)

Reviewer #1 (Remarks to the Author):

RC5-1)

Criteria for publication:

1) Provides strong evidence for its conclusions.

Generally yes, but there is a major exception discussed below.

2) Novel (we do not consider meeting report abstracts and preprints on community servers to compromise novelty).

Yes

3) Of extreme importance to scientists in the specific field.

Yes

4) Ideally, interesting to researchers in other related disciplines.

Yes

Recommendation:

Invite the authors to revise their manuscript to address specific concerns before a final decision is reached

Detailed comments:

I had two major concerns with the first manuscript, and several minor concerns. The authors have expended significant effort to address all points.

AR5-1) Thank you for the opportunity to further revise our manuscript to address all remaining concerns raised by Reviewer #1 before reaching a final decision.

RC5-2) FIRST POINT:

""""

RC1-4) Major issues:

Major concern 1)

Page 7) "These results further support that the unfolded α 1-helix propagates α cat-ABD to weakly dimerize through the β H-dependent interface."

Page 8) "Collectively, these results support the view that α -catenin facilitates actin bundling through ABD homodimerization."

The data establishing that the H1 helix disruption promotes ABD dimer formation was unconvincing. This is presented as one of the central points to their mechanism, and thus is important to establish clearly. They do have some limited data in support of this idea:

AR1-4) α cat-ABD dimerization is one of the major focal points of our manuscript, yet it is difficult to study due to its relatively weak interaction affinity in solution. Hence, we have employed numerous biophysical and biochemical methodologies to provide experimental evidence to support the weak dimerization of ABD, including newly added size-exclusion chromatography, multi-angle light-scattering (SEC-MALS) data in the revised manuscript.

""""

The authors present new SEC-MALS data to address this concern. In doing so, they find evidence of a low abundance population of dimers in the SEC chromatograms, as assessed by a mass estimate using SEC-MALS. By integrating the peak area and plotting relative to the presumed input concentration, they find a roughly linear relationship between input concentration and observed dimer (Fig 2d). Moreover, they find that the increase in dimer area grows faster for H1-ABD than for wt-ABD, although only by a small margin.

I still have a concern with the claim that the H1-ABD truncation supports dimer formation. Unfortunately, this new data does not convince me that H1-ABD dimers are more stable than wt-ABD dimers. There are three reasons for this.

AR5-2) While *in vitro* characterization of weak protein-protein interaction remains technically challenging, your previous suggestion “to definitively demonstrate a dimer using a well-established hydrodynamic method” has tremendously helped us to “push the envelope” in further analyzing this very weak dimerization of α cat-ABD in solution. We believe that our SEC-MALS data, new SDS-PAGE results of the dimer peaks, editorial changes and our responses to your concerns have sufficiently addressed the remaining issues raised by the reviewer. We would like to respectfully point out that the H1 mutation is a set of missense mutations designed to unfold α 1-helix (ABD-H1; RAIM670-673GSGS), and does not contain a truncation mutation.

RC5-3) First, the effect is modest, even at the extreme concentrations used here. Given that their max concentration is ~ 1mM (30 mg/ml for ~28 kDa protein) and the NMR peaks are about 50% shifted at 0.8 mM, I expected more than 1% dimer by this concentration. The magnitude of such effects in SEC can be difficult to predict though, so this is a weak criticism. I worry this may be a contaminant of a fortuitous size as no gels are provided to show the dimer peak contains ABD.

AR5-3) As suggested by the reviewer, it is difficult to directly compare the concentration-dependent changes observed in SEC and NMR. First, diffusion of the injected sample passing through the SEC column will likely result in sample dilution. Hence, dimers that are present in the original sample but can dissociate upon dilution are less likely to be properly quantified¹. Second, our SEC analyses were performed at 4 °C, a considerably lower temperature than the NMR experiments (30 °C), which likely contributed to a higher proportion of monomer compared to the NMR samples. Third, we respectfully disagree with the assumption made by the reviewer that “the NMR peaks are about 50% shifted at 0.8 mM”, since it remains unclear whether the NMR peaks are 100% shifted or ABD dimerization was saturated at 1.6 mM. Based on these points, less than expected dimer formation observed during our SEC analyses compared to our NMR results could be attributed to the inevitable sample dilution during SEC, different experimental temperatures and the limited ranges of sample concentrations associated with these techniques.

To further support our claim that the α cat-ABD weakly dimerizes in solution, we have now added the coomassie-stained SDS-PAGE gel images of the dimer peak fractions from the 30 gm/mL SEC runs (Supplementary Fig. 7c). These results clearly show that the dimer peaks of both α cat-ABD-WT and -H1 predominantly consist of 28-kDa α cat-ABD molecules, without any detectable amount of “contaminant of a fortuitous size”. These findings are consistent with the fact that the samples used in our *in vitro* experiments, including SEC-MALS, were prepared

from the previously purified monomer fraction of α cat-ABD separated by prep-grade SEC. We believe that our SEC-MALS results accompanied by the SDS-PAGE analyses of the dimer peaks provide sufficient evidence to support our claim that the α cat-ABD weakly dimerizes in solution.

RC5-4) Second, the appearance of a resolved dimer peak is unusual for weak monomer-dimer equilibria. More typically, weak monomer-dimer equilibria show up as a broadened but single dominant peak, which shifts to early elution volumes with increasing concentration. The exact peak shape and behavior will depend on the interaction model, concentrations used and rate constants, so the detailed shape is difficult to predict. Here, the authors observe resolved dimers instead.

AR5-4) While “the appearance of a resolved dimer peak is unusual for weak monomer-dimer equilibria”, it does depend on several factors as mentioned by the reviewer. Here, one possible explanation for weakly dimerizing populations of α cat-ABD-WT and -H1 resulting in resolved dimer peaks is that ABD dimers are kinetically slow to dissociate, hence leading to “slow exchange”-like phenomenon on SEC. Slow dissociation of ABD dimers may not be too surprising, considering that crystal structures of the α Ncat-ABD-H1 dimer is formed through an extensive dimer interface, which occludes over 3100 Å² of solvent accessible surface. Interestingly, a weak monomer-dimer equilibrium resulting in a resolved dimer peak has been also documented for weak homodimerization of full-length α -catenin through the N-terminal dimerization domain. Dissociation of full-length α -catenin dimers appears to be kinetically blocked^{2,3}, despite its relatively weak dimer affinity ($K_d = \sim 80 \mu\text{M}$)⁴.

RC5-5) Third, the concentration dependence of the dimer peak seems incorrect for a weak affinity monomer-dimer equilibrium. The authors report that the dimer peak area concentration scales roughly linearly with input concentration (Fig 2d).

If we assume that this area is reporting on the dimer concentration, which seems reasonable, this means the concentration of dimer scales roughly linearly with total concentration.

Does that make sense? Given the very low dimer concentration observed, it is safe to estimate the concentration of ABD monomers, [ABD], as the total ABD concentration. If the dimerization is governed by a monomer-dimer equilibrium, then we can estimate the solution dimer concentration as:

$$[\text{ABD dimer}] * K_{\text{dimer}} = [\text{ABD}]_{\text{total}}^2$$

Thus, this general behavior is inconsistent with the presented data. The concentration dependence appears linear, not parabolic. For example, the ratio of dimer areas for the 10 mg/ml samples and the 30 mg/ml samples is roughly 1:3, and not 1:9. I am not convinced this linear dependence is reporting on a dynamic monomer-dimer equilibrium.

AR5-5) As we discussed in AR5-3), there are several technical issues with trying to compare the concentration-dependent changes observed in our SEC and NMR data. First, a very weak dimer affinity of α cat-ABD and the inevitable sample dilution associated with SEC (100 μL of sample was applied to a 24 mL-bed-volume column to collect a dimer peak over ~ 1 mL elution volume) have likely resulted in our SEC data representing the achievable range of $[\text{ABD}]_{\text{total}}$ that is

relatively narrow and well below the presumably high K_D (Fig. AR5-5). Second, the measurement of the dimer peaks near the detection limit of the UV absorbance detector may have resulted in underestimation of the total amount of ABD dimer (Supplementary Fig. 7c). We believe that these factors likely contributed to the linear-like increase of dimer peak areas. Our NMR data is also limited to the concentration range of ABD that is too narrow to determine a peak position for the fully saturated state (Fig. AR5-5). As suggested by the reviewer, we expect to observe the parabolic behaviour of a dynamic monomer-dimer equilibrium with additional SEC data points at high enough $[ABD]_{Total}$ to reach saturation of dimer formation (Fig. AR5-5). However, the deformed monomer peak shapes at 30 mg/mL of ABD (Supplementary Fig. 7c) suggest that injecting samples with higher $[ABD]$ may be beyond the practical limits of our SEC-MALS experimental setup for collecting reliable data. For these reasons, we respectfully disagree with the reviewer, and feel that the presented data are consistent with the expected behaviour of a dynamic monomer-dimer equilibrium under the experimental conditions tested.

Figure AR5-5) Theoretical plots of Dimer Peak Area vs. $[ABD]_{Total}$ (top) and Fraction of Dimer vs. $[ABD]_{Total}$. A presumed K_D based on very weak ABD dimerization and the estimated $[ABD]_{Total}$ ranges of our SEC and NMR data are depicted.

RC5-6) Fourth, the dimer area is used directly to assert that H1 supports dimer formation. This makes sense if the loaded concentrations and volumes are exact, but this is rarely the case in reality. Given the low fraction of dimer, the monomer areas should give an idea of how well matched the total loading was for each sample. The monomer areas appear to differ on the order of the differences between the H1-ABD and wt-ABD dimer areas. Based on max absorbance level in the provided examples, the H1-ABD monomer peaks appear to be of greater absorbance than the wt-ABD monomer peaks, particularly at those concentrations flagged as being statistically increased in dimer content. In other words, there is a very real chance that the modest difference in dimer accumulation may be due to modest differences in stock concentrations, intrinsic absorbance, or some sort of sample application issue.

AR5-6) We share the same view that detection of weak dimers is technically challenging, yet it is important to show that weak/transient protein-protein interactions play critical roles in nature. When performing the SEC-MALS analyses of α cat-ABD-WT and -H1, we carefully prepared our samples by making dilutions from the same WT and H1 stocks, and by measuring the concentration of each sample immediately before injection. To ensure that our findings were not affected by any slight deviations in protein concentrations and injection volumes, we performed triplicates for each concentration of both WT and H1 to show that the differences observed between WT and H1 at the 20-30 mg/mL concentration range were statistically significant. We had considered using the monomer peak areas to deduce the dimer population, however performing further analysis of the monomer peaks was hampered by the altered monomer peak shape, likely due to overloading of the sample on to the column. Also, differences in the intrinsic absorbance based on the primary sequence is negligible between WT and H1 (0.7% difference) to contribute towards much larger absorbance difference we observed between WT and H1 (>~30%). For these reasons, we believe that the observed differences in dimer formation between WT and H1 are due to the H1-induced effect on ABD dimerization, and unlikely resulted from mishandling or intrinsic absorbance differences of samples during the SEC-MALS analyses.

RC5-7) While I applaud the authors' substantial efforts here, the end result just does not convince me that the H1-ABD mutation stabilizes the dimer over the wt-ABD dimer, and I still worry that this claim is incorrect. I still see potential here, but am not satisfied with the current proof of their thesis.

AR5-7) We hope you will find that the further revised manuscript, which now contains the SEC-MALS data accompanied by the new SDS-PAGE results of the dimer peaks, NMR studies, actin bundling assay results, and the crystallographically observed dimer structures, is substantially improved compared to our previous submissions, and makes a very strong case to support our model that ABD dimerization plays a role in α -catenin's function.

RC5-8)

AR1-8) Based on our NMR results and the new SEC-MALS data, we also believe that α cat-ABD has a very weak affinity for homodimerization in solution. We have directly addressed this by adding the following statement in the Discussion section (page 17) of the main text: "Although α cat-ABD appears to dimerize with a very weak affinity in solution (Fig. 2c,d), AJ-localized α -catenin cooperatively binding to F-actin would likely increase the propensity of α cat-ABD to dimerize and promote F-actin bundling."

This response is reasonable. I still disagree that they have shown a weak affinity homodimer definitively. That said, the conclusion that AJ F-actin binding could hugely affect apparent affinity is completely reasonable.

AR5-8) We appreciate your positive feedback on our assertion that α -catenin facilitating the cadherin-actin linkage will likely engage in F-actin-bundling through ABD dimerization.

RC5-9) Another approach to supporting the dimerization idea is to disrupt the dimer interface and look for bundling effects (which is reasonably cast as the dimer sensitive assay). The Δ BH/H1 mutation disrupts bundling (Fig. 2e), but the Δ BH mutation does not, although slightly less Δ BH co-sediments with actin in the bundling assay. Both Δ BH and Δ BH/H1 disrupt actin binding (right side of Fig 2e). In the bundling assay, despite the disrupted direct binding of actin, the Δ BH material still bundles actin. Given the model, if Δ BH disrupted the dimer, as might be predicted, then it isn't clear how Δ BH can result in bundling.

AR5-9) We agree that the actin bundling assay provides a nice read-out for detecting weak dimerization of actin-binding proteins, and indeed our results clearly show that α cat-ABD can facilitate F-actin bundling, hence supporting the idea of α cat-ABD homodimerization. Our interpretation of the actin bundling results showing that actin bundling can occur with $\Delta\beta$ H and H1 individually, but not with H1 $\Delta\beta$ H, is stated on page 8 as: "These results suggest that actin bundling can be facilitated by ABD dimerization through the β H-dependent interface, as well as through an unknown mechanism involving the α 1-helix in our assays." Considering the inherent flexibility of α 1-helix, one possibility is that the α cat-ABD- Δ BH could form a dimer by exchanging α 1-helices between two protomers.

RC5-10) This all raises a real concern on my part, is the dimer 'real'? While the NMR line broadening and observation of a limited quantity of dimer in SEC-MALS suggests a dimer may exist, it doesn't actually mean that a dimer is relevant for the rest of the biology and biochemistry discussed. How much of the story is lost if the H1 deletion just shifts an allosteric conformational equilibrium that affects actin binding? I think the newly provided BLI experiments actually speak to the conformational equilibrium nicely (Supplemental Fig 13b). Focusing on an allosteric equilibrium keeps the strongest parts of the story (in this reviewer's mind at least) and removes what is proving to be a persistently weak element. I think the paper would still warrant publication in Nat Comm minus the 'H1 disruption leads to dimerization' point, as the depth of information on the allosteric mechanism is greatly interesting on its own.

AR5-10) We would like to thank the reviewer for carefully considering all the merits of our paper, and suggesting that it is important to discuss all possibilities, including the H1 mutation "just shifts an allosteric conformational equilibrium that affects actin binding" without promoting ABD dimerization. We agree with the reviewer's suggestion and we have modified the following sentence in the Discussion section on page 17: "Considering the very weak α cat-ABD dimerization (Fig. 2c and Supplementary Fig. 7c), which is marginally increased by the H1 mutation in solution (Fig. 2d), it is possible that tension-induced unfolding of α 1-helix allosterically changes the conformational dynamics of the actin-binding site without affecting dimerization." We believe that these changes have addressed Reviewer #1's main concern, which helped us to improve the manuscript significantly with more unbiased and robust discussion of our results in the manuscript.

RC5-11) SECOND POINT:

""""

RC1-10) Major concern 2)

Page 12) “Our site-directed mutagenesis and actin cosedimentation assays with the α Ecat-ABD variants support critical roles of L785, I792 and V796 in F-actin-binding: Ala substitutions of these hydrophobic residues, individually or together as 3A, led to 40~90% reduction in the amount of ABD cosedimenting with F-actin compared to α Ecat-ABD-WT (Fig. 5c), whereas mutating K795, preceding V796, resulted in less reduction (~30%) (Fig. 5c). ...

AR1-10) ...

The mutagenesis of hydrophobic residue data is much more compelling. The V714 mutation is a much better control. The added BLI data with these mutations is also convincing and alleviates concerns about quantitation with the spin down assays.

AR5-11) We would like to thank the reviewer for suggesting to include these new data to further strengthen our manuscript.

RC5-12)

Minor response points.

1) New p. 13: “Also, none of the above mutations appear to interfere with the F-actin bundling activity of α Ecat- ABD (Supplementary Fig. 12).”

Examining Supplementary Fig. 12, it is not at all clear how this statement connects with the data. L785A and 3A have clear effects in the bundling assays, V796A probably has an effect, though more modest. In the H1 context, all of them have a significant effect. Given that the point here was about direct actin binding, I am not sure this discrepancy affects the argument. Please correct.

AR5-12) The amount of cosedimented actin resulting from the actin bundling assay depends on both the actin-binding affinity and the ability of ABD to dimerize, hence we cannot infer the level of actin-bundling activity of α cat-ABD or its dimerization by comparing the supernatant/pellet ratio of actin. Nevertheless, all the actin-binding site mutants cosedimented with F-actin during actin bundling assays. To convey that none of the actin-binding site residue mutations did not appear to interfere with the ability of ABD to bundle F-actin through homodimerization, we have modified the above statement by replacing “activity” with “ability” to reflect the data.

RC5-13) 2) The H1 context is barely mentioned in the results section at all. Seems worth a passing comment at least, particularly given the more pronounced effects shown. Is it puzzling that the bundling is less effective in the H1-hydrophobic mutant experiments?

AR5-13) Thank you for the suggestion. We have now added the following text in the results section (page 13): “The effects of I792A and V796A were greater in the H1 background (reduction of 70% and 73%, respectively), confirming that alterations of these residues significantly reduce F-actin binding by α Ecat-ABD-H1 (Fig. 5c).”

As we discussed in AR5-12, the sedimented amount of actin in the actin bundling assay is affected by the direct actin-binding affinity of α cat-ABD as well as their ability to bundle actin. Hence, the reduced amounts of bundled actin observed with H1I792A compared to that of I792A

are consistent with the apparent actin binding affinity of H1I792A being lower than I792A based on Fig. 5c.

RC5-14) 3) I am not sure the actin cosedimentation experiments are sufficiently described in the methods. What are the concentrations of actin and ABD in the spin down assays?

AR5-14) The final concentrations of actin and ABD in the actin cosedimentation experiments are both 5 μ M. We have added this information in the Methods section as: “Samples of F-actin and ABD were mixed (the protein mixture contains 5 μ M ABD and 5 μ M actin in 50 μ L) in Ultra-Clear Centrifuge Tubes (Beckman Coulter) and incubated for 1 h at RT.”

RC5-15) 4) My gut instinct is that the R_{max} based $K1/K2$ (%) numbers may be telling you something about the allosteric conformational bias. The H1 and $\Delta\beta$ H mutations seem particularly provocative (my comment on $\Delta\beta$ H is inferred from the time course as it does not appear in the table).

AR5-15) Yes, it is intriguing that the largest changes in the $KD1/KD2$ (%) ratio were observed with the mutants containing V796A, which plays a central role in the allosteric conformational change of α cat-ABD. A good fit of H1 data to 1:1 binding is consistent with the increased propensity of V796 to be exposed, as shown in our MD and NMR-based modelling results (Fig. 6d,e). Although we cannot comment on the $KD1/KD2$ (%) number for $\Delta\beta$ H based on a single-concentration binding curve shown in Supplementary Fig. 14b, the reduced R_{max} of $\Delta\beta$ H compared to WT suggests that this deletion mutation may have inadvertently affected the conformational state of V796.

RC5-16) Responses to RC1-12 through RC1-20:
Satisfactory changes all around.

AR5-16) Thank you for your feedback.

RC5-17) One minor point there, the ‘N’ subscript nomenclature helps, but it used inconsistently

AR5-17) We have corrected this inconsistency in Supplementary Fig. 5.

References

1. Wang, W. & Roberts, C.J. *Aggregation of Therapeutic Proteins*. (John Wiley & Sons, Inc., Hoboken, New Jersey; 2010).
2. Pokutta, S., Choi, H.-J., Ahlsen, G., Hansen, S.D. & Weis, W.I. Structural and thermodynamic characterization of cadherin- β -catenin- α -catenin complex formation. *J. Biol. Chem.* **289**, 13589-13601 (2014).
3. Desai, R. *et al.* Monomeric α -catenin links cadherin to the actin cytoskeleton. *Nat. Cell Biol.* **15**, 261-273 (2013).
4. Ishiyama, N. *et al.* An autoinhibited structure of α -catenin and its implications for vinculin recruitment to adherens junctions. *J. Biol. Chem.* **288**, 15913-15925 (2013).